***Nat Immunol.* Author manuscript; available in PMC 2022 June 10.**

# Hypoxia shapes the immune landscape in lung injury and promotes the persistence of inflammation

**Ananda S. Mirchandani[1],\*, Stephen J. Jenkins[#1], Calum C. Bain[#1], Manuel A. Sanchez-Garcia[1], Hannah Lawson[2], Patricia Coelho[1], Fiona Murphy[1], David Griffith[1], Ailiang Zhang[1], Tyler Morrison[1], Tony Ly[3], Simone Arienti[1], Pranvera Sadiku[1], Emily R. Watts[1], Rebecca. S. Dickinson[1], Leila Reyes[1], George Cooper[1], Sarah Clark[4], David Lewis[3], Van Kelly[3], Christos Spanos[3], Kathryn M. Musgrave[5,6], Liam Delaney[1], Isla Harper[1], Jonathan Scott[5], Nicholas J. Parkinson[7], Anthony J. Rostron[5], J. Kenneth Baillie[7,3], Sara Clohisey[7], Clare Pridans[1], Lara Campana[8], Philip Starkey Lewis[8], A John Simpson[5], David H. Dockrell[1], Jurgen Schwarze[1], Nikhil Hirani[1], Peter J. Ratcliffe[9,10,11], Christopher W. Pugh[9], Kamil Kranc[2], Stuart J. Forbes[1,8], Moira K. Whyte[1], Sarah R. Walmsley[1]**

[1]University of Edinburgh Centre for Inflammation Research, Queen's Medical Research Institute, University of Edinburgh, Edinburgh, United Kingdom

[2]Barts Cancer Institute, Queen Mary University of London, London, United Kingdom

[3]Wellcome Centre for Cell Biology, School of Biological Sciences, University of Edinburgh, Edinburgh

[4]Intensive Care Unit, Royal Infirmary of Edinburgh, NHS Lothian

[5]Translational and Clinical Research Institute, Newcastle University, Newcastle upon Tyne, UK

[6]Department of Respiratory Medicine, The Newcastle upon Tyne Hospitals NHS Foundation Trust, Newcastle upon Tyne, UK

[7]Roslin Institute, University of Edinburgh, Edinburgh, UK

[8]Centre for Regenerative Medicine, University of Edinburgh, Edinburgh, United Kingdom

[9]Nuffield Department of Medicine Research Building, Nuffield Department of Medicine, University of Oxford, Oxford, United Kingdom

[10]Ludwig Institute for Cancer Research, Nuffield Department of Medicine, University of Oxford, Oxford, United Kingdom

\*Correspondence to: Ananda Mirchandani (Ananda.Mirchandani@ed.ac.uk).

**Author contributions:** A.S.M, S.J.J, C.C.B, H.L, K.K, S.J.J, M.K.W, and S.R.W. conceived and 20 designed the experiments, A.S.M, C.C.B, H.L, P.C, F.M, M.A. S-G, LR, TM, SA, R.D, E.R.W, G.C, L.D, D.G, I.H, J.S, A.Z, L.C, D.L, V.K, C.S, N.J.P and P.S-L performed the experiments. A.S.M, C.C.B, M.A. S-G, H.L, T.L, G.C analysed the data. A.S.M, C.C.B, S.J.J, H.L, K.K, S.J.F, C.P, P.J.R. M.K.W and S.R.W. interpreted the data. S.C, K.M.M, J.S, A.J.R, A.J.S, D.H.D. facilitated obtaining patient samples. A.S.M S.J.J. N.H. M.K.W. and S.R.W. helped obtain funding. J.K.B and C.L provided reagents. J.S. provided *Ifnar1* KO mice. A.S.M, S.J.J, C.C.B, H.L, K.K, C.W.P, S.J.F, M.K.W and S.R.W. wrote the manuscript.

**Declaration of interests**
A.S.M, S.R.W, M.K.W, S.J.F, S.J.J have filed a patent for the use of CSF1 as a therapy in ARDS with the University of Edinburgh (PCT/GB/2020/051184). A.J.S is a National Institute for Health Research (NIHR) Senior Investigator. The views expressed in this article are those of the authors and not necessarily those of the NIHR, or the Department of Health and Social Care.

[11]The Francis Crick Institute, London, United Kingdom

[#] These authors contributed equally to this work.

## Abstract

Hypoxemia is a defining feature of acute respiratory distress syndrome (ARDS), an often-fatal complication of pulmonary or systemic inflammation, yet the resulting tissue hypoxia, and its impact on immune responses, is often neglected. Here we showed that ARDS patients were hypoxemic and monocytopenic within the first 48 hours of ventilation. Monocytopenia was also observed in mouse models of hypoxic acute lung injury, in which hypoxia drove the suppression of type I interferon signalling in the bone marrow. This impaired monopoiesis, resulted in reduced accumulation of monocyte-derived macrophages and enhanced neutrophil-mediated inflammation in the lung. Administration of CSF1 in mice with hypoxic lung injury rescued the monocytopenia, altered the phenotype of circulating monocytes, increased monocyte-derived macrophages in the lung and limited injury. Thus, tissue hypoxia altered the dynamics of the immune response to the detriment of the host and interventions to address the aberrant response offer new therapeutic strategies for ARDS.

---

ARDS, a clinical syndrome defined by bilateral opacites on chest imaging denoting the presence of lung inflammation, blood hypoxemia with tissue hypoxia, and the requirement for positive pressure ventilation has a mortality rate of up to 40%[1,2]. Despite decades of research, effective therapies for ARDS remain elusive with much of the treatment research focusing on modulating the inflammatory injury, as its persistence is a poor prognostic indicator[3,4]. However, the effects of blood hypoxemia and the ensuing tissue hypoxia on the persistence of inflammation in ARDS have not been fully investigated.

Macrophages have a key role in driving tissue inflammation resolution. In mice, lung macrophages can be subdivided into SiglecF[+] alveolar macrophages (AM) and SiglecF[−] interstitial macrophages (IM), which inhabit distinct anatomical niches. Under homeostatic conditions, AM largely self-renew[5,6] whilst IM require recruitment of circulating monocytes[7,8]. Evidence is emerging for key roles of IM in inflammation regulation[9,10] and repair[11].

Here we investigated whether hypoxemia associated with ARDS, including severe COVID-19 disease, affected the accumulation of lung monocyte-derived macrophages (MDM) and if this in turn impacted the resolution of inflammation. We found that patients with ARDS had clinical evidence of persistent hypoxemia despite ventilatory support and were profoundly monocytopenic during the first 48 hours of ventilation, an observation replicated in mouse models of hypoxic acute lung injury (ALI). We further showed that systemic responses to tissue hypoxia suppressed type 1 interferon (IFN) signalling and fundamentally altered bone marrow hematopoiesis, with ensuing consequences for the phenotype and number of monocytes in the blood, and accumulation of MDM in the lung during ALI. This, in turn, led to the persistence of inflammation. Critically, targeting this pathway with the monocyte and macrophage growth factor CSF1 corrected these hypoxia-mediated changes, and drove inflammation resolution.

# Results

## ARDS is characterised by monocytopenia and an altered immune phenotype

Despite the heterogeneity of the etiologies leading to ARDS, one defining feature is blood hypoxemia and tissue hypoxia. We first characterised the arterial oxygen partial pressure ($PaO_2$) in patients with moderate to severe ARDS. In accordance with the Berlin criteria, patients were sampled within 1 week of a known insult or new or worsening respiratory symptoms, had bilateral opacities on chest radiograph and evidence of respiratory failure with a $PaO_2/FiO_2$ (fraction inspired $O_2$) < 200 mmHg and positive end-expiratory pressure (PEEP) > 5cm $H_2O$. Patients with ARDS had a range of etiologies and associated pathogens (Table 1), and showed clinically significant (Fig. 1a) and sustained (Fig. 1b) blood hypoxemia over a 24 hour period despite supplementary oxygen therapy (Fig. 1c) and ventilatory support. In 13 of the 22 patients this was associated with an elevated circulating lactate (Fig. 1d), indicating ongoing tissue hypoxia. To characterize the kinetics of circulating leucocyte populations, we sampled blood from ventilated patients <48 hours from diagnosis of ARDS and commencement of positive pressure ventilation (hereafter early ARDS) or 48 hours-7 days from diagnosis (late ARDS). Healthy donors were used as controls because tissue hypoxia is a common feature of critically unwell patients. Early ARDS patients had elevated circulating leucocyte counts (Fig. 1e), but significantly lower proportions and numbers of circulating monocytes (Fig. 1e) compared to controls. In late ARDS patients, circulating leucocyte numbers remained elevated compared to controls (Fig. 1f), but monocyte frequency and counts were equivalent (Fig. 1f). In early ARDS, we detected an increase in the proportion of CD14[+]CD16[+] intermediate monocytes at the expense of classical CD14[++]CD16[-] monocytes (Extended data Fig. 1a-c)[12]. ARDS monocytes, irrespective of timepoint, had lower expression of the major histocompatibility class II (MHCII) marker HLADR and higher expression of CD11b compared to healthy controls (Fig. 1g). Proteomic analysis of sorted blood CD14[++]CD16[-] ARDS monocytes indicated changes in the abundance of proteins whose transcripts have been reported to be sensitive to hypoxic culture in human monocytes[13] when compared to healthy controls (Fig. 1h). We also observed a significant increase in secretory granule content in ARDS CD14[++]CD16[-] monocytes (Fig. 1i,j) which was associated with altered expression of hypoxia-regulated proteins, including SLC2A3[14], IGFR2[15], PSMD4[16] and FTL[17]. Nanostring analysis identified a specific transcriptional signature in ARDS monocytes (Fig. 1k), with 41 genes differentially expressed compared to healthy controls (Fig. 1l). Notably, 4 MHC complex genes *(HLA-DMA, HLA-DMB, HLA-DQA1, HLA-DRB3),* important for antigen-presenting function, as well as genes associated with monocyte adhesion and extravasation (SELL[18]), trans-endothelial migration (*CD99*[19]) and LPS-signaling (*MAP2K4*[20], *MAP3K14*[21]) were significantly downregulated in ARDS monocytes (Fig. 1l). Thus, ARDS affected both the transcriptomic and protein signature of blood monocytes.

## Experimental ARDS reproduces monocytopenia and phenotypic alterations

We next investigated whether the circulating monocyte profile alterations observed in ARDS patients were replicated by the induction of hypoxemia in a mouse model of ALI (Extended data fig. 2a). Mice exposed to 10% $FiO_2$ demonstrated equivalent levels of hypoxemia to

patients with ARDS (Extended data Fig. 2b). Administration of nebulised LPS immediately prior to induction of hypoxia or maintenance in normoxia induced an increase in circulating leucocytes and $CD115^+CD11b^+$ monocytes (Fig. 2a) in mice housed in normoxia, relative to naïve mice after 24 hours, yet this increase was absent in LPS-challenged mice housed in hypoxia (10% $FiO_2$). A 10 selective loss of non-classical $CD115^+CD11b^+Ly6C^{lo}$ monocytes in LPS-challenged hypoxic mice (Fig. 2a) relative to normoxic LPS-challenged mice was observed. Blood $CD115^+CD11b^+Ly6C^{hi}$ classical monocytes from hypoxic LPS-challenged mice had decreased expression of the adhesion molecules ICAM1 and CD11a and increased expression of the CCR2 chemokine receptor, compared to normoxic counterparts (Fig. 2b). The absolute number of circulating leucocytes, $CD115^+CD11b^+$ monocytes (Fig. 2c) and $Ly6G^+CD11b^+$ neutrophils (Extended data Fig. 2c) in the blood of normoxic or hypoxic LPS-treated mice normalised at day 5, although the proportion of $Ly6C^{lo}$ monocytes remained contracted in hypoxic LPS-challenged mice compared to normoxic LPS-challenged controls (Fig. 2c). We also observed persistent alterations in the transcriptome of $Ly6C^{hi}$ monocytes in hypoxic LPS-challenged mice (Fig. 2d), including decreased expression of the chemokine receptor *Ccr5* and the scavenger receptor *Cd36*, which are markers of monocyte maturity[22] and increased expression of *Il1b*, an inflammatory cytokine associated with poor outcomes in ARDS[23]. Thus, in mice with ALI, hypoxia drives alterations in monocyte numbers and phenotype that parallel those described in ARDS patients.

**Tissue hypoxia prevents accumulation of lung MDM**

Next we explored the effect of hypoxemia on the LPS-induced inflammation in the lung. Consistent with tissue hypoxia, HIF-1α protein was most highly expressed in the lungs of hypoxic LPS-treated mice compared to normoxic-LPS controls (Fig. 3a). LPS challenge significantly increased lung $Ly6G^+CD11b^+$ neutrophil numbers in hypoxia- or normoxia-housed mice compared to naive controls (Fig. 3b), with equivalent numbers of $CD3^+CD19^-MHCII^-$ T cells and $CD3^-CD19^+MHCII^+$ B cells in the lungs of these mice (Extended data Fig. 3a and b). LPS challenge led to a significant expansion of the $Lin^-CD45^+CD64^{hi}$ macrophage compartment in normoxia-, but not hypoxia-housed mice (Fig. 3c, d). AM are $CD64^{hi}SiglecF^+CD11c^+$, while $CD64^{hi}SiglecF^-$ macrophages are almost exclusively comprised of interstitial macrophages (IM) in health, and include inflammation-elicited, parenchymal or alveoli-localized, $CD64^{hi}SiglecF^-y6C^+$ monocyte-derived macrophages (MDM) after injury. The number of $CD64^{hi}SiglecF^+CD11c^+$ macrophages was equivalent in LPS-challenged mice housed in hypoxia or normoxia (Fig. 3c, d). Hypoxia significantly blunted the LPS-mediated expansion of the $CD64^{hi}SiglecF^-$ macrophage compartment observed in normoxic mice (Fig. 3c, d), an effect that appeared to be entirely attributable to the absence of $CD64^{hi}SiglecF^-Ly6C^+$ MDMs in LPS-treated hypoxic mice (Fig. 3c, d). The number of $CD64^{hi}SiglecF^-Ly6C^-MHCII^+$ macrophages was similar in all treatments (Fig. 3c, d). The reduction in the number of $CD64^{hi}SiglecF^-Ly6C^+$ MDMs occurred despite elevated amounts of the monocyte chemoattractant CCL2 in the alveoli of LPS-treated hypoxic mice (Fig. 3e) and similar numbers of $CD64^{lo}CD11b^+Ly6C^+$ monocytes in the lung of hypoxic and normoxic LPS-treated mice (Extended data Fig. 3c). These observations suggested that monocytes recruited to the lung following LPS-treatment did not convert to

CD64[hi]SiglecF[−]Ly6C[+] MDMs in hypoxia. The contribution from intravascular cells to lung cell counts could not be completely excluded in these data.

To examine the effects of hypoxia on monocyte-macrophage dynamics in a model of severe streptococcal pneumonia, C57/BL6J mice were inoculated with *D39 Streptococcus pneumoniae* or vehicle and housed in either hypoxia or normoxia following a 4hour recovery period. Reduced blood leukocyte and monocyte counts were detected in infected mice housed in hypoxia compared to normoxia (Extended data Fig. 3d,e). Whilst the number of lung neutrophils was equivalent in hypoxic and normoxic *S.pneumoniae-infected* mice (Fig. 3f), the accumulation of CD64[hi]SiglecF[−] macrophages was reduced (Fig. 3f), with a particular absence of CD64[hi]SiglecF[−]Ly6C[+] MDMs (Fig. 3 f, g).

To determine if hypoxia altered the monocyte-macrophage lung compartment directly, we returned LPS-challenged hypoxic mice to normoxic conditions after 24hours of hypoxia for a further 24hours. Mice returned to normoxia showed a significant increase in the proportion and the number of CD64[hi]SiglecF[−] macrophages and BAL MDM, compared to mice that remained in hypoxia (Fig. 3h). Together, these data indicated that hypoxia directly induced sustained changes in the lung macrophage compartment during various inflammatory challenges.

## Hypoxia directly alters hematopoiesis

To determine the mechanism by which hypoxia regulated the number of circulating monocytes, we measured bone marrow (BM) output by pulsing naïve or LPS-challenged mice, housed in hypoxia or normoxia, with bromodeoxyuridine (BrdU) 12hours post-LPS[12,24]. Hypoxemic LPS-challenged mice, had a 80% reduction in the proportion of BrdU[+] CD115[+]CD 11b[+]Ly6C[hi]monocytes compared to normoxic LPS-challenged counterparts (Fig. 4a). LPS equally reduced the proportion of blood BrdU[+] neutrophils in hypoxic and normoxic mice compared to naive controls (Extended data Fig. 4a), with similar frequency of BrdU[+] lymphocytes in all samples (Extended data Fig. 4b). Examination of the BM stem cell compartment (Extended data Fig. 4c) 24hours post-hypoxic exposure indicated a reduction in absolute numbers of Lin[−]Sca-1[+]Kit[+] (LSK) cells in hypoxic compared to normoxic mice independent of LPS treatment (Figure 4b), with a specific reduction in the CD48[+]CD150[-] HPC-1 and CD48[+]CD150[+] HPC-2 hematopoietic progenitor cells (Fig. 4b), which have restricted multipotency[25]. Irrespective of oxygenation, LPS-treatment reduced the absolute number of Lin[-]cKit[+]Sca1[-]CD127[-]CD16/32[-]CD34[+] common myeloid progenitor cells (CMP) compared to naïve control mice (Fig. 4b).

Because erythrocytes and monocytes originate from CMP[26] (Extended data Fig.4d), we investigated whether hypoxia shifted hematopoiesis in favour of red blood cell production by measuring the effect of hypoxia on CMP progeny (Lin[-]cKit[+]CD41[-]CD32/16[-]) 24 hours after LPS-challenge[26]. Hypoxia did not affect the number of CD150[-]CD105[-] pre-granulocyte/monocyte precursors (pre-GM) (Fig. 4c), but significantly increased the proportion of CD150[+]CD105[-]megakaryocyte-erythroid precursor (pre-MegE) cells (Fig. 4c), and led to an increase in downstream CD150[+]CD105[+] pre-colony-forming-unit-erythroid precursor (pre-CFU-E) cells (Fig. 4c). Whilst hematocrit values were equivalent at this 24 hour time-point (Extended data Fig. 4e), BM CFU-E proportions were reduced in LPS-treated

hypoxic mice compared to normoxic counterparts (Fig. 4c) suggesting increased red cell egress. The hematocrit of LPS-treated hypoxic mice was increased compared to normoxic controls at day 5 (Extended data Fig. 4f). UMAP analysis of Lin⁻cKit⁺CD41⁻CD32/16⁻ cells within the bone marrow compartment post-LPS indicated that skewing of the CMP towards erythropoiesis was achieved by day 5 (Fig. 4d, e). Taken together, these data demonstrated that hypoxia altered hematopoiesis, reducing monocyte BM output.

## Hypoxia suppresses type I interferon signalling

Next, we investigated the mechanism by which hypoxia suppressed monopoiesis in mice with ALI. Erythropoietin (EPO), a red-blood cell growth factor, was significantly increased at 24 hours (Fig. 5a) and day 5 post-LPS in hypoxia-housed compared to normoxia-housed mice (Fig. 5b). IL-11, a hypoxia responsive megakaryocyte and hematopoietic growth factor[27,28] was also increased at day 5 post-LPS in hypoxic compared to normoxic mice (Extended data Fig. 5a). Type I interferon (IFN-α and IFN-β) and type II IFN (IFN-γ) are known drivers of emergency monopoeisis[29,30]. While IFN-β and IFN-γ were equivalent in hypoxic and normoxic LPS-treated mice, IFN-α was markedly reduced in hypoxic mice 24 hours post-LPS challenge compared to normoxic counterparts (Fig. 5c and Extended data Fig. 5b and 5c). *Ifnar1*⁻/⁻ mice, which lack the type I IFN receptor, had a contraction of the LSK compartment (Fig. 5d) and increased proportion of the MEP/preCFUe/CFUe erythroid progenitors 24 h after LPS-treatment during normoxia (Fig. 5e-g). Five days post-LPS challenge, normoxic *Ifnar1*–/– mice had enhanced numbers of circulating red blood cell and were significantly monocytopenic compared to wild-type controls (Fig. 5h). In addition, normoxic *Ifnar1*⁻/⁻ mice had similar numbers of neutrophils and CD64^hiSiglecF⁺ 20 macrophages (Fig. 5i), but reduced numbers of CD64^hiSiglecF⁻ macrophages and MDM (Fig. 5i) in the lung 24 hours post-LPS, compared with wild-type controls.

Increased HIF-1α stabilisation was observed in hypoxic LPS-treated mice BM compared to normoxic counterparts (Fig. 5j), in-keeping with tissue hypoxia. Because hypoxia can alter type I IFN signalling in cancer[31], we next investigated IFNAR expression in the BM. LSK cells from hypoxic mice 24 hours post-LPS treatment did not upregulate IFNAR expression, compared to normoxic mice (Figure 5k). IFNAR expression was also reduced in blood CD115⁺CD11b⁺Ly6C^hi monocytes from hypoxic LPS-challenged mice compared to normoxic counterparts (Extended data Fig. 5d). BM cells from naïve mice cultured *in vitro,* in hypoxia (1% O₂) showed significant blunting of type I IFN-mediated *Irf8, Irf1* and *Ccr5* expression (Fig. 5l). Taken together, these findings suggested that hypoxia directly altered the immune response through local and systemic mechanisms.

## Loss of monocyte recruitment associates with persistence of inflammation

We subsequently investigated the longer-term effects of systemic hypoxia on the myeloid compartment and inflammation resolution. Infiltrating neutrophils promote vascular injury, protein leak and alveolar epithelial damage in ARDS, and drive deleterious inflammatory responses in murine models of hypoxic ALI[32,33]. 5 days after LPS challenge, normoxic mice had very few neutrophils within the bronchoalveolar space while hypoxic mice showed evidence of ongoing inflammation with significant bronchoalveolar neutrophilia (Fig. 6a). Hypoxic LPS-treated mice had a reduction in the number of bronchoalveolar

CD64$^{hi}$SiglecF$^-$MDMs compared to normoxic counterparts (Fig. 6b). The total number of lung Ly6G$^+$CD11b$^+$ neutrophils was greater in hypoxic compared to normoxic mice at day 5 post-LPS (Fig. 6c), and while the number of CD64$^{hi}$SiglecF$^+$CD11c$^+$ macrophages had returned to baseline (Fig. 6c), the non-AM CD64$^{hi}$SiglecF$^-$ macrophages remained contracted (Fig. 6c), largely as a consequence of fewer CD64$^{hi}$SiglecF$^-$Ly6C$^+$ MDMs (Fig. 6c). In addition, the bronchoalveolar lavage (BAL) from hypoxic mice at this timepoint had higher CXCL1 and IL-6 (Fig. 6d), parameters reported to be elevated in ARDS non-survivors[23]. Moreover, hypoxic mice showed more sustained weight loss at day 5 post-LPS, compared to the normoxia-housed controls (Fig. 6e). Collectively, these data indicated that hypoxia-induced monocytopenia was associated with persistent lung inflammation.

## CSF1 accelerates the resolution of lung inflammation in hypoxia

Finally, we tested whether increasing the number of monocytes during hypoxia facilitated inflammation resolution. LPS-challenged hypoxic mice with absent (24 h) or low (5 days) baseline levels of M-CSF (Extended data Fig. 6a, b) were treated with 4 daily injections of CSF1-Fc fusion protein[34] or PBS. CSF1-Fc markedly increased the number of CD115$^+$CD11b$^+$ monocytes and moderately increased the number of Ly6G$^+$CD11b$^+$ neutrophils in the blood at day 5 compared to PBS (Fig. 7a). CSF1-Fc-treated hypoxic LPS-challenged mice also had increased numbers of CD64$^{lo}$CD11b$^+$Ly6C$^+$ monocytes and CD64$^{hi}$SiglecF$^-$ macrophages in the lung compared to mice receiving PBS (Fig. 7b). The number of CD64$^{hi}$SiglecF$^+$CD11c$^+$ macrophages were not affected (Fig. 7b). Importantly, the absolute number of Ly6G$^+$CD11b$^+$ neutrophils in the lung tissue (Fig. 7b) and BAL (Fig. 7c) were reduced in CSF1-Fc-treated hypoxic LPS-challenged mice at day 5, despite equivalent levels of CXCL1 in the BAL between CSF1-Fc treated and PBS-treated mice (Extended data Fig. 6c). Hypoxic LPS-challenged mice treated with CSF1-Fc had reduced weight loss (Extended data Fig. 6d) and reduced IgM levels in the BAL fluid at day 5 (Fig. 7c) compared to PBS counterparts, suggesting reduced alveolar inflammation and vascular leak. To test these observations in a model of virally-induced epithelial injury, C57/BL6J mice were inoculated with influenza A virus (PR8) and placed in hypoxia immediately after. CSF1-Fc or PBS was administered 12h and 36h post-PR8 challenge. Hypoxic PR8-infected mice treated with CSF1-Fc had increased numbers of CD64$^{lo}$CD11b$^+$Ly6C$^{hi}$ monocytes in the lung (Extended data Fig. 6e), and improved physiological outcomes (Extended data Fig. 6f) compared to mice receiving PBS. This was associated with a significant reduction in BAL protein levels, a marker of lung injury (Extended data Fig. 6g) and lactate dehydrogenase activity, as an indicator of cellular damage (Extended data Fig. 6h).

To dissect the mechanism by which CSF1-Fc accelerated inflammation resolution, we tracked the ontogeny of CD64$^{hi}$SiglecF$^-$ macrophages in CD45.1$^+$CD45.2$^+$ C57BL6 mice reconstituted with CD45.2$^+$ BM cells following lung-protected single-dose irradiation. 8 weeks after BM reconstitution (Extended data Fig. 6i), mice were challenged with LPS, housed in normoxia or hypoxia for 5 days and treated with PBS or CSF1-Fc daily. In keeping with increased recruitment from the blood, the proportion of CD64$^{hi}$SiglecF$^-$Ly6C$^+$ MDMs derived from donor CD45.2$^+$ BM cells in LPS-treated mice paralleled that seen in the blood (Fig. 7d, e). In the absence of LPS challenge, the proportion of CD45.2$^+$CD64$^{hi}$SiglecF$^-$Ly6C$^-$ macrophages relative to CD45.2$^+$CD115$^+$CD11b$^+$ blood

monocyte pool was ~20%, indicating that maintenance of this subset was dependent on blood monocytes (Fig. 7e)[35]. In LPS-treated mice housed in normoxia, the proportion of CD45.2+CD64hiSiglecF-Ly6C- macrophages relative to CD45.2+CD115+CD11b+ blood monocytes was ~80%, suggesting the expansion of this population during inflammation was predominantly through recruitment of blood monocytes (Fig. 7e). The chimerism of this population was ~60 % in hypoxic LPS-challenged PBS-treated mice and ~80 % in hypoxic LPS-challenged mice treated with CSF1-Fc (Fig. 7e), indicating that CSF1-Fc replenished the number of CD64hiSiglecF− lung macrophages in LPS-treated hypoxic mice predominantly through increased recruitment of circulating CD115+CD11b+ monocytes.

In LPS-treated hypoxic mice, CSF1-Fc treatment also normalised the hypoxic-suppression of the type I IFN-associated gene *Ccr5,* without enhancing expression of *Il1b* (Fig. 7f). CSF1-Fc significantly induced the expression of a number of genes that were suppressed in the ARDS patient samples, such as *Itga5, Cd99, Sell* and *Anxa*[36] (Fig. 7g, Supplementary Table 1). Proteomic survey showed reduced abundance of secretory-granule proteins in circulating Ly6Chi monocytes of LPS-challenged hypoxic CSF1-treated compared with PBS-treated counterparts (Fig. 7h), suggesting that CSF1 altered their phenotype towards a less inflammatory profile. Increased numbers of lung MHCII- Lyve-1+ macrophages, a subset of repair-associated macrophages reported in bleomycin-induced lung injury[11], were observed in CSF1-Fc-treated hypoxic LPS-challenged mice compared to PBS counterparts (Fig. 7i and Extended data Fig. 6j) with increased chimerism noted in the CSF1-Fc-treated mice in the lung protected chimera model (Fig 7j)

To explore how CSF-1-Fc-mediated expansion of CD64hiSiglecF- macrophages facilitated neutrophil clearance, we investigated the expression of known mediators of efferocytosis in the blood and lung[37]. IL-10 was elevated in the serum of CSF1-Fc-treated hypoxic LPS-challenged mice compared to the PBS-treated counterparts (Fig. 7k). Immunofluorescence staining showed increased numbers of IL-10-expressing interstitial F4/80+ cells (Fig. 7l and Extended data Fig. 7a), and F4/80+Lyve1+ macrophages (Fig. 7l and Extended data Fig. 7b) in the lung of hypoxic LPS-challenged CSF1-Fc-treated mice compared to PBS-treated counterparts. Sorted CD64hiSiglecF-MHCII-Lyve1+ macrophages from these CSF1-Fc-treated mice expressed the *Il-10* transcript (Extended data Fig. 8a). Transcriptional profiling indicated that CD64hiSiglecF-MHCII-Lyve1+ macrophages from CSF1-Fc-treated LPS-challenged hypoxic mice had lower expression of archetypal inflammatory genes *(Il1b, Il6, Tnf* and *Il18* and *S100a11)* and genes associated with lung fibrosis (Mmp8, *Mmp12, Col14a1, Fpr1, Pdgfa, Pdgfb)* relative to CD64hiSiglecF-MHCII- from PBS-treated counterparts. Furthermore, genes reported to be enriched in severe SARS-CoV2 infection BAL monocyte or macrophages, such as *CD14, CCL3, S100A8*[38] were supressed in the CD64hiSiglecF-MHCII-Lyve1+ macrophages from the CSF1-Fc-treated hypoxic mice compared to PBS-treated counterparts (Fig. 7m), indicating that CSF1 expanded the number of IL-10-producing macrophages and enabled resolution of hypoxia-driven inflammation.

Finally, we asked if CSF1 was sufficient to overcome the loss of CD64hiSiglecF−Ly6C+ MDMs and CD64hiSiglecF- macrophages observed in the absence of type I IFN signalling. Normoxic LPS-challenged *Ifnar1* -/- mice treated with CSF1-Fc (4 daily injections) had increased numbers of CD115+CD11b+ monocytes in the blood, Ly6Chi monocytes

(Extended data Fig. 8b and 8c) and CD64[hi]SiglecF[-] macrophages in the lung (Fig. 7n) compared to those receiving PBS. In addition, these mice had increased lung CD64[hi]SiglecF[-]Lyve1[+] macrophage (Fig. 7n) and BAL-recovered CD64[hi]SiglecF[-] MDM numbers (Fig. 7o) with enhanced return to baseline bodyweight at day 5 post-LPS challenge (Fig. 7p) compared to LPS-challenged PBS-treated *Ifnar1*[-/-] mice.

## Discussion

Here we showed that monocyte recruitment and conversion into lung macrophages is required to drive inflammation resolution in hypoxic ALI. Hypoxemic mice with ALI demonstrated an increase in erythropoiesis, with an associated reduction in monopoiesis, monocytopenia and failure to expand the MDM and non-AM CD64[hi]SiglecF[-]macrophage compartment in the lung. In the context of prioritising the preservation of tissue oxygen-delivery, increased erythropoiesis makes physiological sense, such as in adaptation to altitude, where monocytopenia has been reported as early as 1969[39]. However, when engagement of an effective innate immune response is also required, our data demonstrated that hypoxia-induced immune changes observed in early disease have long-term consequences for inflammation resolution, such as persistence of neutrophilic inflammation, a well-known poor prognostic feature of ARDS[3].

Monocytes are professional phagocytes and are key mediators of restoration of homeostasis. There is increasing appreciation that the phenotype of circulating monocytes can be pre-determined by systemic cues affecting their BM progenitors[40]. The presence of a specific phenotypic profile in ARDS patients' circulating monocytes, irrespective of the sampling timepoint, would be in keeping with alterations within their progenitors. It will be important, in future work, to explore whether the absence of systemic IFN-α, and suppressed IFNAR expression on BM LSK and circulating monocytes, as observed in our mouse model of hypoxic ALI, and in hypoxemic patients with severe SARS-Cov2 infection[41,42], may be sufficient to drive the phenotypic and functional changes observed in the circulating monocytes of hypoxic ARDS cohort.

Expanded numbers of airway monocytes and macrophages in the BAL of patients with severe SARS-Cov2 infection have been reported[38], although it is unclear whether this expansion is promoting or limiting disease pathogenesis. An important limitation of our current study is the inability to sample the lung macrophage compartment in patients with ARDS. Transcriptomic survey of the airway monocytes and macrophages in patients with severe SARS-Cov2 infection identified an enrichment of markers of immaturity, inflammatory proteins and cytokines[38]. We show that treatment with CSF1 increased the number of CD64[hi]SiglecF[-] macrophages and monocyte differentiation towards CD64[hi]SiglecF[-]MHC[-]Lyve1[+] macrophages, a cell-type that has tissue repair roles in various disease contexts[43], including in the lung[11]. Additionally, CSF1-Fc suppressed the expression of several genes reported to be enriched in the BAL of patients with severe SARS-Cov2 infection[38]. These findings, compounded with the effect of CSF1-Fc on accelerating inflammation resolution and reducing lung injury, underscored the therapeutic potential of CSF1 in ARDS. The use of other growth factors has been trialled in ARDS without success, with systemic-[44] or lung-delivered[45] GM-CSF failing to improve ARDS mortality. While

GM-CSF plays a key role in AM homeostasis, its pleotropic nature means it can also act as a neutrophil chemoattractant and growth factor[46], with neutrophils pathogenic in the context of ARDS, and hypoxia promoting neutrophil survival and pro-inflammatory function[32,33]. On the other hand, monocytes and MDMs can directly inhibit neutrophil-mediated damage to the host[47]. In our system, CSF1-Fc treatment expanded the CD64$^{hi}$SiglecF$^-$ macrophage compartment and led to an increase in lung IL-10$^+$ macrophages, with a concomitant increase in systemic IL-10 and reduced numbers of neutrophils in the airspace and the lung. Macrophages are known to release IL-10 upon efferocytosis, which drives the resolution of inflammation[37]. Furthermore, IL-10-producing lung IM are key regulators of both allergic[8,9] and endotoxin-mediated lung injury[10]. These findings strengthen the case of the therapeutic potential of CSF1 in human ARDS.

## Methods

### Resource Availability

**Lead contact**—Further information and requests for resources and reagents should be directed to and will be fulfilled by the Lead Contact, Ananda Mirchandani (Ananda.Mirchandani@ed.ac.uk)

**Human healthy control blood donors**—Patients with ARDS were recruited and informed consent obtained directly or by proxy under the "META-CYTE" study (17/SS/0136/AM01) and "ARDS-NEUT" study (20/SS/0002) as approved by the Scotland A Research Ethics Committee. Samples were also obtained under the "Effects of Critical Illness on the Innate Immune System" study as approved by Health Research Authority (REC number 18/NE/0036).

All healthy participants gave written informed consent in accordance with the Declaration of Helsinki principles, with AMREC approval for the study of healthy human volunteers through the MRC / University of Edinburgh Centre for Inflammation Research blood resource (15-HV-013). Up to 20-40mls of whole blood was collected into citrate tubes and up to 10 million cells were stained for flow cytometry assessment and sorting. Briefly, the whole blood was treated with red cell lysis buffer (Invitrogen) and cells counted prior to staining for flow cytometry. Cells were incubated with anti-CD16/32 Fc-block (2:50) for 30 minutes followed by staining for 30 minutes with antibodies (see Table 2) followed by a wash with FACS buffer (PBS+2% Fetal calf serum (FCS)). Dapi (1:1000) was added prior to flow cytometry to determine live cells. Monocytes were identified as Singles Dapi$^-$ CD45$^+$non-granulocyte Lin(CD3/CD56/CD19+/-CD66b)$^-$ HLADR$^+$ CD14$^+$ and/or CD16$^+$ cells.

Samples obtained from April 2020 were fixed prior to acquisition given the potential for SARS-Cov2 dissemination. Briefly, 1uL of zombie Aqua fixable viability dye (stock 1:20 dilution) was added to 100uL of whole blood for 15 minutes at room temperature in the dark. 2uL of FcBlock was added for a further 30 minutes, on ice. Samples were then stained as above and fixed/ lysed using BD FACS Lyse for 10 minutes at room temperature. The sample was then resuspended in 300uL of FACS buffer and 50uL of Countbright beads added (Thermofisher) prior to acquisition.

**Mice**—Male C57/BL6J mice aged 8-15 weeks were purchased from Envigo or Charles River. *Ifnar1*[-/-] (*ifnar* [tm/agt]) mice were obtained from J.S. who purchased them originally from the Jackson laboratory. Animal experiments were conducted in accordance with the UK Home Office Animals (Scientific Procedures) Act of 1986 with local ethical approval.

**Mouse LPS ALI Model**—Mice were treated with nebulized LPS (3mg), and were then housed in normoxia or hypoxia (10% O2) immediately thereafter for up to 5 days. Mice were treated with Colony Stimulating Factor (CSF)-1-Fc (kind gift from Prof D Hume) by subcutaneous injection (0.75mg/g/mouse) from day 1 to 4 post-LPS, prior to cull on day 5.

**D39 Strep. Pneumoniae infection**—Mice were anesthetized and $10^7$ colony forming units (or vehicle) were delivered in 50uL PBS via intratracheal intubation. Following reversal of anesthetic and a period of recovery, mice remained in normoxia, or were placed in hypoxia.

**Influenza A (PR8) virally-induced ALI model**—Mice were lightly anesthetized using isofluorane and 20 plaque-forming units (p.f.u.) of PR8 Influenza A virus in Dulbecco's Modified Eagle's Media were inoculated intranasally. After 1 hour of recovery time, mice were placed in hypoxia for 48 hours. Subcutaneous PBS or CSF1-Fc injections (as above) at 12 hours and 36 hours were administered. Sickness scores were determined using methods as described previously[32].

**Lung and alveolar cell sampling**—Mice were culled with an overdose of intraperitoneal anesthetic (Euthetal) followed by blood collection from the inferior vena cava. Alveolar leukocytes were collected by bronchoalveolar lavage (BAL), then mice were perfused lightly with PBS through the heart, prior to harvest of lung tissue. On occasion, lower limbs were harvested for bone marrow leukocyte assessment (see below).

Tissue leukocytes were extracted from surgically dissociated lung tissue by enzymatic digestion with 2ml enzyme mix (RPMI with 0.625 mg/ml collagenase D (Roche), 0.85 mg/ml collagenase V (Sigma-Aldrich), 1 mg/ml dispase (Gibco, Invitrogen) and 30 U/ml DNase (Roche Diagnostics GmbH) for 45 minutes at 37°C in a shaking incubator. The digest material was passed through a 100μM cell strainer with the addition of FACS buffer (PBS with 0.5% BSA/2% fetal calf serum and 0.02mM EDTA). Cell pellets were treated with red cell lysis buffer (Sigma) and washed in FACS buffer. The resulting cell suspension was subsequently passed through a 40μm strainer before cell counting using a Casey TT counter (Roche). Single cell suspensions (5 million cells/sample) were then stained for flow cytometry. BAL samples were counted prior to staining for flow cytometry.

**Blood and bone marrow sampling**—Mouse blood and bone marrow were treated with red blood cell lysis buffer (Biolegend) prior to counting and staining for flow cytometry (see Table 2).

Hematopoietic cell assessment was performed using both hind legs that were crushed using a pestle and mortar until a homogenous cell suspension was achieved or flushed though using a 32G needle. Cells were collected in cold FACS buffer and filtered through a 70 um

nylon strainer (BD Falcon, 352340). Cells were treated with RBC lysis buffer (Biolegend) prior to staining

**Tissue protected chimeras**—6–8-week old C57BL/6J CD45.1$^+$CD45.2$^+$ mice were anesthetized and were irradiated with a single dose of 9.5 Gy $\gamma$-irradiation while all but the hind legs and lower abdomen were protected by a 2 inch lead shield. The following day, the mice received 2-5 x 10$^6$ BM cells from CD45.2$^+$ C57BL/6J by iv injection. The chimerism of blood monocytes (proportion of donor cells) was determined by flow cytometry in each individual mouse at day 5 and the chimerism in the lung macrophage populations (as described in the figures) was divided by this reference value, thereby determining the proportion of the cells that were of blood ontogeny.

**Flow Cytometry**—Mouse cells were treated with $\alpha$-CD16/32 Fc block (e-bioscience) (1:100) prior to staining with antibodies (see Table 3). Relevant fluorescence minus one (FMO) samples were used as controls. Zombie Aqua fixable viability dye (Biolegend) was used prior to Fc block to exclude dead cells from digest samples or Dapi for single cell suspensions.

Cells were acquired on the LSRFortessa (Becton Dickinson) or sorted on an Aria II or Fusion machine (Becton Dickinson). Compensation was performed using BD FACSDiva software and data analyzed in FlowJo version 10 or FCS Express 7 for tsne analysis.

**Gating strategies**—Human monocytes: Singles Dapi$^-$CD45$^+$non-granulocyte Lin(CD3/CD56/CD19/CD66b)$^-$HLADR$^+$ CD14$^+$ and/or CD16$^+$ cells

Mouse blood monocytes: Singles Dapi$^-$CD45$^+$ Lin(CD3/CD19/ Ly6G)$^-$CD115$^+$CD11b$^+$Ly6C$^{hi}$, Ly6C$^{int}$ or Ly6C$^-$

Mouse blood neutrophils: Singles, Dapi$^-$CD45$^+$Ly6G$^+$CD11b$^+$Ly6C$^{int}$

Mouse lung/ BAL alveolar macrophages: Singles, Zombie Aqua$^-$CD45$^+$Lin (CD3/CD19/Ly6G)$^-$CD64$^{hi}$SiglecF$^+$CD 11c$^+$

Mouse lung interstitial/BAL inflammatory macrophages: Singles, Zombie Aqua$^-$CD45$^+$Lin (CD3/CD19/Ly6G)$^-$CD64$^{hi}$SiglecF$^-$CD11c$^{+/-}$ then Ly6C$^{+/-}$ MHCII$^{+/-}$

Lung classical monocytes: Singles, Zombie Aqua$^-$SinglesCD45$^+$Lin (CD3/CD19/Ly6G)$^-$CD64$^{lo}$ CD11b$^+$ Ly6C$^+$

Lung/ BAL neutrophils: Singles Aqua or Dapi$^-$, CD45$^+$CD11b$^+$Ly6G$^+$

Lung cDC1 Zombie Aqua-Singles CD45$^+$CD11c$^{hi}$,CD103$^+$, CD64$^-$MHCII$^+$

BM HSPC SLAM analysis Alive, Singles LK (Lin-cKit$^+$) and LSK (Lin-cKit$^+$Sca-1$^+$) cells. LSK cells were further sub-gated on hematopoietic stem cells (HSCs: LSK CD48-CD150$^+$), multipotent progenitors (MPPs: LSK CD48$^-$CD150$^-$), hematopoietic progenitor cells-1 (HPC-1: LSK CD48$^+$CD150$^-$) and hematopoietic progenitor cells-2 (HPC-2: LSK CD48$^+$CD150$^+$)

BM erythroid progenitors based on Pronk analysis[26] Singles, Dapi or Aqua[-], Lin[-], CD11b[-], cKit[+], Sca1[-], CD32/16[-], CD41[-], CD105[+] or CD150[+] (Pre-MegE CD150[+]CD105[-], Pre-CFUE CD150[+]CD105[+], CFUE CD150[-]CD105[+])

Further gating strategy information can be made available upon request.

**BAL/serum Cytokine/ Chemokine quantification—**BAL and serum supernatants were collected and stored at -80°C until use. Cytokine and chemokine levels were measured using an MSD V-plex plate as per manufacturer's instructions.

**Lung injury measurements—**IgM BAL levels were measured using Ab133047 Abcam kit as per manufacturer's instructions.

BAL LDH activity (measured as colorimetric reduction of NAD to NADH) was performed using Ab102526 (Abcam) as per manufacturer's instructions.

BAL total protein was measured using Pierce BCA Assay (Thermofisher) as per manufacturer's instructions

**In vitro bone marrow culture—**Naïve wild-type C57BL/6 bone marrow was obtained flushing the femoral and tibial bones and RBC were lysed. Cell were cultured in hypoxia (FiO$_2$ 1%) or normoxia (FiO$_2$ 21%) with conditioned DMEM for 1 hour prior to addition of IFNβ 10ng/ml (RnD 8234-MB-010) for a further 3 hours. Cell pellets were collected and Qiagen RLT buffer added (containing 10uL/ml ß-mercaptoethanol). Pellets were snap frozen and stored at -80 for RNA extraction.

**RNA isolation and relative quantification—**RNA was isolated from bone marrow cells using the gDNA eliminator solution for purification of total RNA (RNeasy Plus Mini Kit, Qiagen). cDNA was synthesized by using AMV reverse transcriptase with random primers (Promega). TaqMan gene expression assays (Applied Biosystems, Thermo Fisher) and PrimeTime qPCR Probe Assays (IDT) were used for relative quantification of cDNA using SDS 2.4 (Thermo Fisher) and normalized to ACTB expression.

**Immunohistochemistry—**Murine paraffin-embedded blocks were prepared from lungs fixed via the trachea with 10% buffered formalin. The lung sections were stained with anti-IL10 (ab189392, Abcam), anti-F4/80 (ab6640, Abcam) or isotype control after deparaffinization and antigen retrieval. Antigen retrieval was performed by microwave heating in citric acid-based antigen unmasking solution (Vector, cat. H-3300-250). The following were used TSA plus system amplification (NEL744B001KT, Perkin Elmer) and autofluorescence quenching with TrueView (Vector, SP-8400). The nuclei were stained with DAPI (422801, Sigma-Aldrich). Images were obtained using EVOS FL Auto 2 (Invitrogen). All image acquisition and processing steps were performed using the same settings for both sample groups.

The lung sections were stained with anti-mouse LYVE-1 (103-PA50AG, ReliaTech GmbH), anti-mouse F4/80 (ab6640, Abcam) overnight at 4°C after deparaffinization and antigen retrieval. Antigen retrieval was performed by microwave heating in citric acid-based

antigen unmasking solution (Vector, cat. H-3300-250). The following were used TSA plus system amplification (NEL744B001KT, Perkin Elmer) and autofluorescence quenching with TrueView (Vector, SP-8400) according manufacturer's instructions. The nuclei were stained with DAPI (422801, Sigma-Aldrich). Images were acquired using a EVOS FL Auto 2 (Invitrogen).

All image acquisition and processing steps were performed using the same settings for both sample groups.

**nCounter NanoString platform analysis**—For human monocytes 5000 cells were sorted using the aforementioned human monocyte gating strategy directly into 2ul RLT buffer using a BD Fusion Sorter (Patients 4-8 were sampled). 5000 mouse classical monocytes were sorted from mice treated with LPS and housed in normoxia, hypoxia and hypoxia+CSF1 gating on single $DAPI^-CD45^+Lin^-CD115^+Ly6C^{hi}$ cells into 2ul RLT. Cell pellets were vortexed and centrifuged prior to immediate freezing until ready for processing. Human and mouse myeloid inflammation NanoString gene expression plates were run as per manufacturer's instructions at the University of Edinburgh HTPU Centre within the MRC Institute of Genetics and Molecular Medicine/Cancer Research UK Edinburgh Centre.

**Proteomic analysis**—Sorted monocytes were processed for proteomics using the 'in cell digest", as described by Kelly 25 et al. bioRxiv 2020[48], resuspended in digestion buffer (0.1 M triethylammonium bicarbonate + 1mM $MgCl_2$) and digested with benzonase (>99%, Millipore) for 30 min at 37 °C, followed by trypsin (Thermo Fisher Scientific, 1:50 w/w protein) overnight at 37 °C. A second aliquot of trypsin (1:50) was subsequently added and incubated at 37 °C for 4 hours. A minimum of 25 ng of trypsin was added. Digests were acidified and desalted using StageTips[49] and either subjected to tip-based fractionation or direct analysis by LC-MS/MS.

Following digestion, and in order to generate the reference spectral library, peptides were subjected to reverse phase high pH tip fractionation following the general guidelines as described by Rappsilber et al.[49] In brief, tips for fractionation were made using three SDB-XC disks (Merck, UK) per tip. The tip was cleaned and conditioned using sequentially methanol, 80% acetonitrile (MeCN) (Thermo Fisher Scientific, UK) in 0.1% $NH_4OH$ (v/v) and 0.1% $NH_4OH$ (52mM) (v/v). Peptides, resuspended also 0.1% $NH_4OH$ (pH=10), were spun through the SDB-XC disks and the flow-through was collected, acidified and concentrated on C-18 stage-Tips before subjected to MS analysis. Fractionation was then achieved by sequential elution with 7%, 14%, 21%, 28%, 35%, 55%, and 80% MeCN in 0.1% $NH_4OH$. Fractions were then dried at ambient temperature (Concentrator 5301, Eppendorf, UK) and prepared for MS analysis by resuspension in 6uL of 0.1% TFA.

Data-dependent acquisition (DDA) LC-MS analyses were performed on an Orbitrap Fusion™ Lumos™ Tribrid™ Mass Spectrometer (Thermo Fisher Scientific, UK) coupled on-line, to an Ultimate 3000 HPLC (Dionex, Thermo Fisher Scientific, UK). Peptides were separated on a 50 cm (2 μm particle size) EASY-Spray column (Thermo Scientific, UK), which was assembled on an EASY-Spray source (Thermo Scientific, UK) and operated constantly at 50°C. Mobile phase A consisted of 0.1% formic acid in LC-MS grade water

and mobile phase B consisted of 80% acetonitrile and 0.1% formic acid. Peptides were loaded onto the column at a flow rate of 0.3 μL min$^{-1}$ and eluted at a flow rate of 0.25 μL min$^{-1}$ according to the following gradient: 2 to 40% mobile phase B in 120 min and then to 95% in 11 min. Mobile phase B was retained at 95% for 5 min and returned back to 2% a minute after until the end of the run (160 min in total).

The spray voltage was set at 2.2kV and the ion capillary temperature at 280°C. Survey scans were recorded at 60,000 resolution (scan range 400-1600 m/z) with an ion target of 1.0E6, and injection time of 50ms. MS2 was performed in the orbitrap (resolution at 15,000), with ion target of 5.0E4 and HCD fragmentation[50] with normalized collision energy of 27. The isolation window in the quadrupole was 1.4 Thomson. Only ions with charge between 2 and 6 were selected for MS2. Dynamic exclusion was set at 60 s. The cycle time was set at 3 seconds.

Samples subjected to Data Independent Acquisition (DIA), were prepared for MS analysis by resuspension in 0.1% TFA. MS Analyses were performed on an Orbitrap Fusion™ Lumos™ Tribrid™ Mass Spectrometer (Thermo Fisher Scientific, UK). LC conditions (instrumentation, column, and gradient) were the same as described above.

Survey scans were performed at 15,000 resolution, with scan range at 350-1500 m/z, maximum injection time at 50ms and AGC target at 4.5E5. MS/MS DIA was performed in the orbitrap at 30,000 resolution with a scan range of 200-2000 m/z. The mass range was set to "normal" the maximum injection time to 54ms and the AGC target to 2.0E5. The inclusion mass list with the correspondent isolation windows are shown in the table below. Data for both survey and MS/MS scans were acquired in profile mode. A blank sample (0.1% TFA, 80% MeCN 1:1 v/v) was run in between of each sample to avoid carryover.

| m/z | z | t start (min) | t stop (min) | Isolation Window (m/z) |
|---|---|---|---|---|
| 410 | 3 | 0 | 155 | 20 |
| 430 | 3 | 0 | 155 | 20 |
| 450 | 3 | 0 | 155 | 20 |
| 470 | 3 | 0 | 155 | 20 |
| 490 | 3 | 0 | 155 | 20 |
| 510 | 3 | 0 | 155 | 20 |
| 530 | 3 | 0 | 155 | 20 |
| 550 | 3 | 0 | 155 | 20 |
| 570 | 3 | 0 | 155 | 20 |
| 590 | 3 | 0 | 155 | 20 |
| 610 | 3 | 0 | 155 | 20 |
| 630 | 3 | 0 | 155 | 20 |
| 650 | 3 | 0 | 155 | 20 |
| 670 | 3 | 0 | 155 | 20 |
| 690 | 3 | 0 | 155 | 20 |

| m/z | z | t start (min) | t stop (min) | Isolation Window (m/z) |
|---|---|---|---|---|
| 710 | 3 | 0 | 155 | 20 |
| 730 | 3 | 0 | 155 | 20 |
| 750 | 3 | 0 | 155 | 20 |
| 770 | 3 | 0 | 155 | 20 |
| 790 | 3 | 0 | 155 | 20 |
| 820 | 3 | 0 | 155 | 40 |
| 860 | 3 | 0 | 155 | 40 |
| 910 | 3 | 0 | 155 | 60 |
| 970 | 3 | 0 | 155 | 60 |

MS raw data files were processed using Spectronaught v14.7.201007.47784 using either a human or mouse reference FASTA sequence from UniProt using default search parameters. The resulting protein-level data were analysed using R v3.5.0. Protein parts-per-million (ppm) intensities were calculated by dividing the mean ppm intensities between conditions (e.g. for the human monocyte samples, ARDS and healthy control), P-values were calculated using a t-test on log-transformed ppm intensities. Proteins were designated as significantly changing if they showed p-values <0.05 and fold changes exceeding 1.96 standard deviations away from the mean (i.e. z-score >1.96). Only proteins that were quantitated in all samples are shown in the volcano plot.

**Gene expression analysis—**Normalization of data was carried out using the geNorm selection of housekeeping genes function on NanoString nCounter analysis software. Resulting Log2 normalized values were used in subsequent analyses. Differential genes ("DE genes) were defined as genes with log2 FC>1, P-value <0.05 across sample groups. Hierarchical clustering of sets of DE genes was carried out using Euclidian and Ward methods based on Pearson correlation values across transcriptional scores. Z-score scalar normalization of data was applied to the data prior to plotting as heatmaps. Analyses, including the drawing of heatmaps and volcano plots were carried out in R using the package ggplot2 (https://cran.r-project.org/web/packages/ggplot2/index.html). Analysis of datasets was carried out by Thomson Bioinformatics, Edinburgh, UK.

**Quantification, statistical analysis and reproducibility—**Statistical tests were performed using Prism 8.00 and 9.0.2 software (GraphPad Software Inc) (specific tests detailed in figure legends). Significance was defined as a p value of <0.05 (after correction for multiple comparisons where applicable). Sample sizes (with each n number representing a different blood donor for human cells or an individual mouse for animal experiments) are shown in each figure.

Mirchandani et al. Page 17

## Extended Data

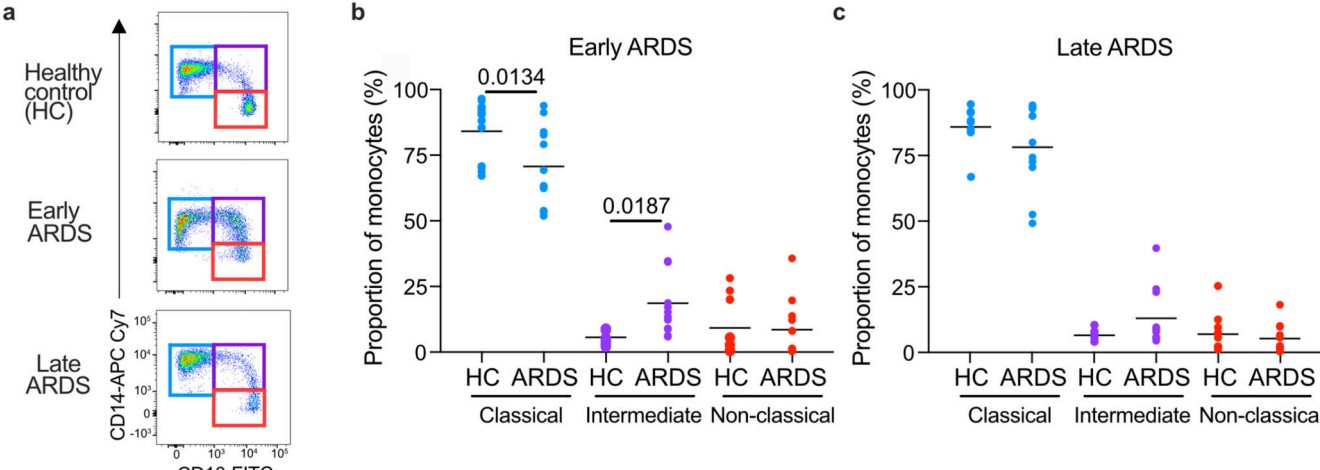

**Extended Data Fig. 1. Monocyte sub-populations are altered early in ARDS**
(**a**) Representative plots and proportions of monocyte sub-populations based on CD14 5 and
CD16 expression early and (**b**) late (**c**). Each data point = one individual patient/ healthy
donor control (HC), **b, c** Data+mean, one-way ANOVA with Holm-Sidak post-test.

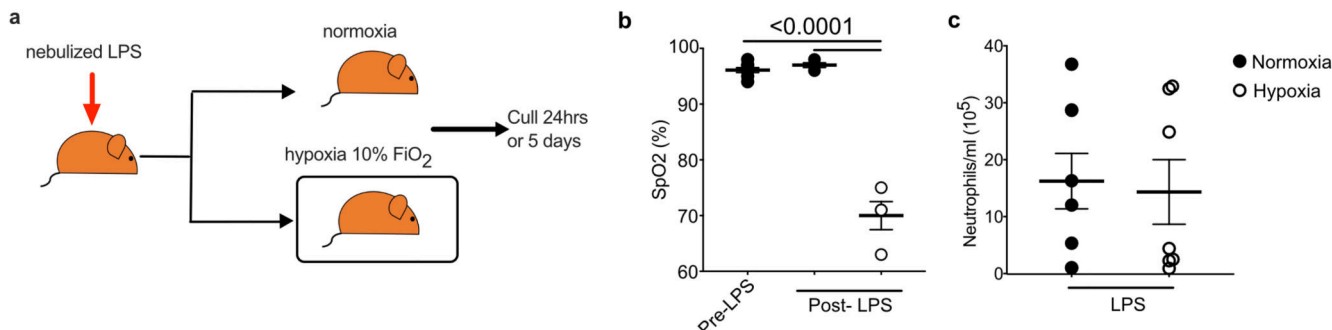

**Extended Data Fig. 2. A hypoxic environment induces hypoxaemia with equivalent circulating
neutrophil**
(**a**) Schematic of normoxic and hypoxic LPS-induced ALI. (**b**) Oxygen saturations in mice
were measured in mice at baseline pre-LPS nebulisation (pre-LPS) and 6 hours post-LPS (N
LPS-mice housed in normoxia post-LPS, H LPS-mice housed in hypoxia post-LPS). (Pre-
LPS n=6, N LPS n=3, H LPS n=3). (**c**) Blood was collected from mice treated with LPS and
placed in normoxia (N LPS) or hypoxia 10% (H LPS) for 5 days and circulating neutrophils
quantified by flow cytometry (Live Singles CD45[+]Ly6G[+]CD11b[+]). **b, c** Each data point
represents an individual mouse. Data shown as mean+SEM. **c** two pooled independent
experiments. Statistical testing performed using one-way ANOVA with Tukey's multiple
comparisons test.

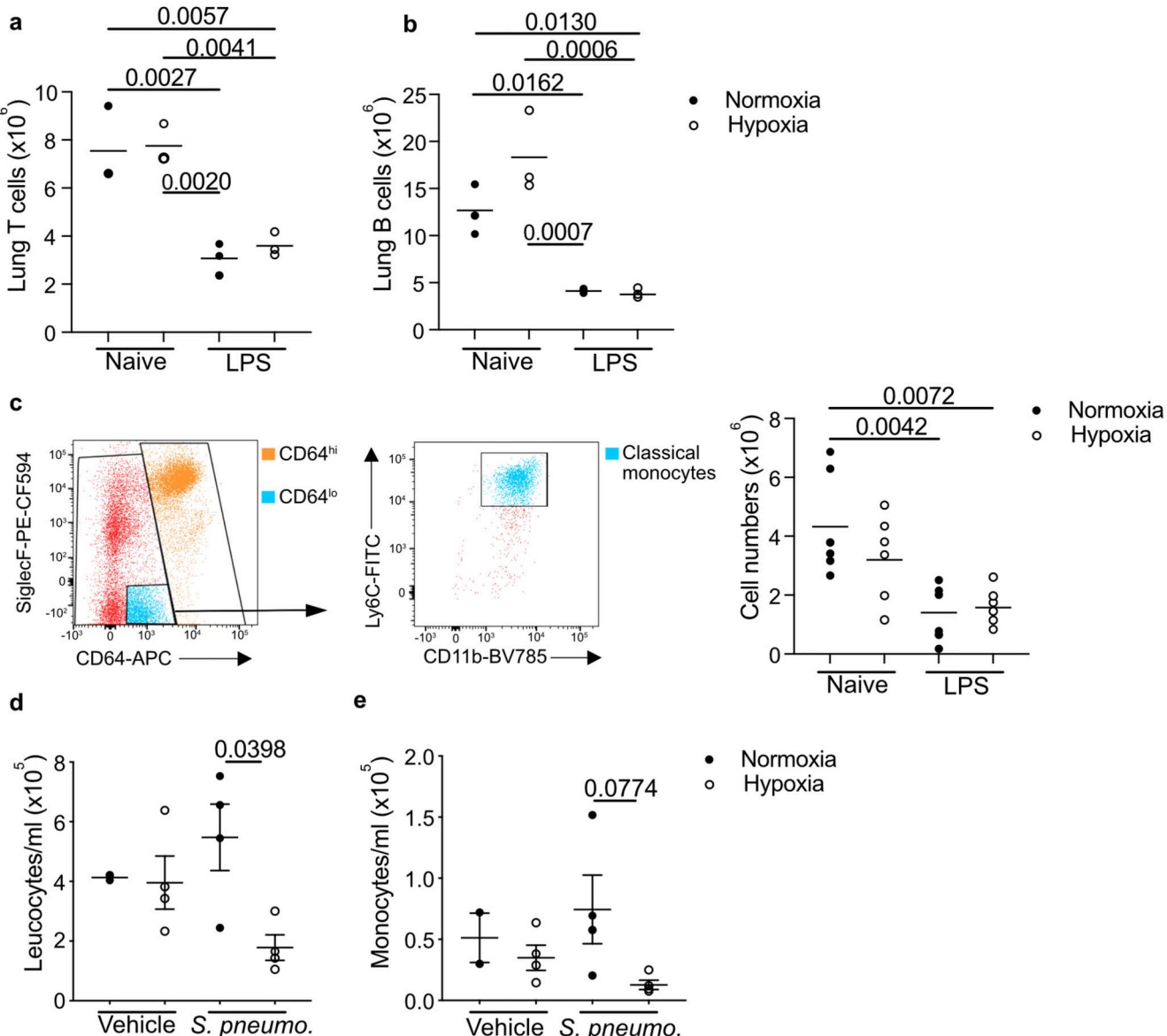

**Extended Data Fig. 3. T cell and B cells are equivalent post-LPS and *Streptococcus pneumoniae* infection in hypoxia leads to leukopenia and monocytopenia.**

T cells (**a**) (Live, singles, CD45[+], Lin[+] (CD3/CD19/Ly6G) MHCII[-] CD11b[-]) and B cells (**b**) (Live, singles, CD45[+], Lin[+] (CD3/CD19/Ly6G) MHCII[+] CD11b[-]) were quantified in lung digests from mice housed in normoxia (N) or 10% $FiO_2$ hypoxia (H) for 24 hours, left naïve or nebulised with LPS. Mice were inoculated with *Streptococcus pneumoniae* (Strep) or vehicle (Veh) intratracheally (i.t.) and housed in normoxia (N) or hypoxia (H) until 24 hours post-i.t. (**c**) Representative dot plots of gating strategy for classical monocytes in the lung gated on Singles Live CD45[+]Lin-lung cells and associated counts in the lung 24 hours post-LPS challenge and housed in normoxia (N) and hypoxia (H). Blood cell counts and (**c**) monocyte counts mice challenged with vehicle or *strep pneumoniae* and housed in normoxia (N) or hypoxia (H) for 24 hours. Each point represents and individual mouse. Data shown

as mean+SEM. Statistical testing performed using one-way ANOVA with Tukey's multiple comparisons test.

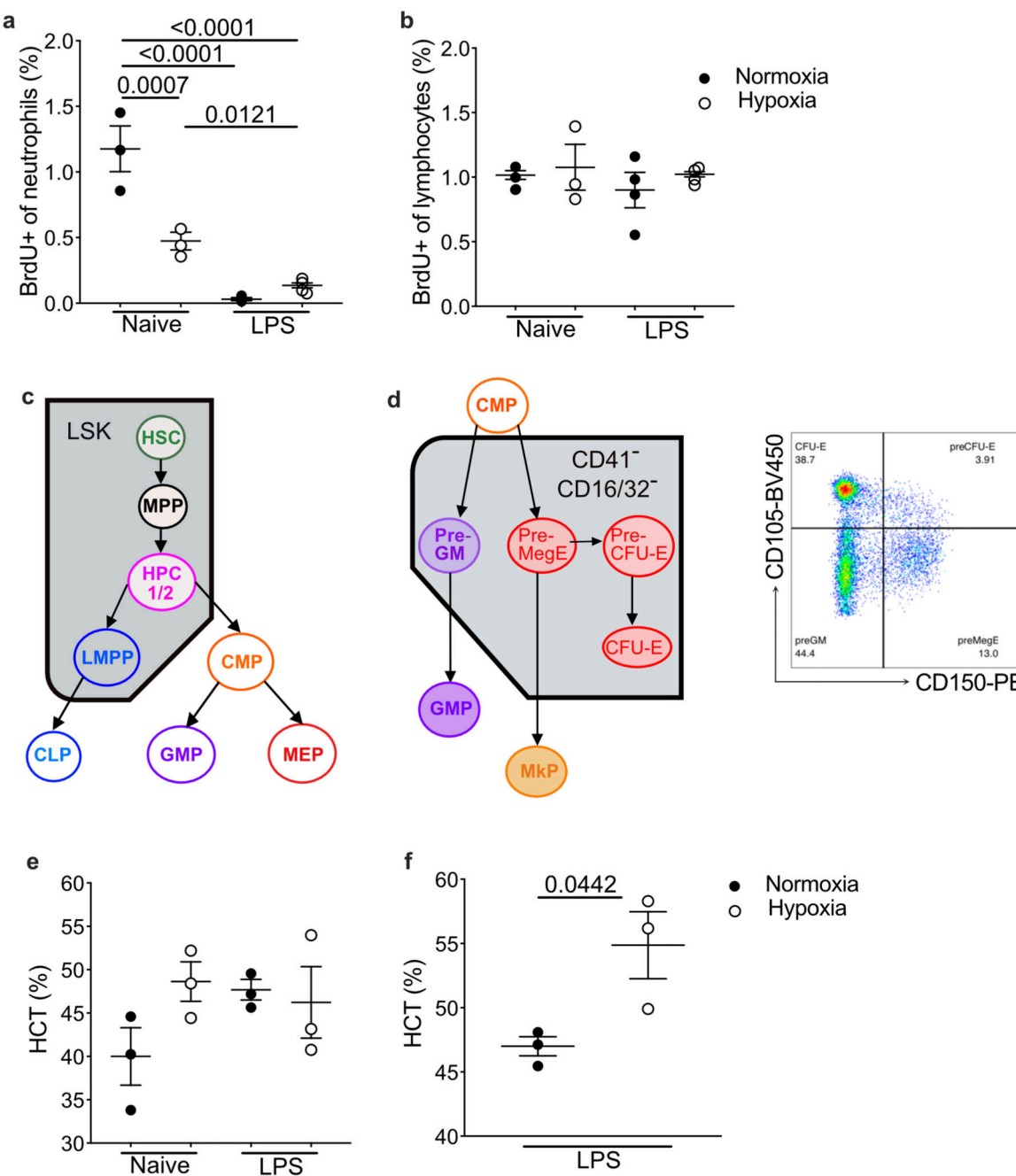

**Extended Data Fig. 4. Impact of hypoxia on bone marrow cell egress and composition**
(**a**) BrdU+ blood lymphocytes (CD3 and CD19+) and (**b**) BrdU+ blood neutrophil proportion in mice treated with LPS and housed in normoxia (N) and hypoxia (H) for 24 hours (naïve N and H n=3, LPS N and H n=4). Data representative of 2 experiments (**c**) Schematic showing hematopoietic hierarchy with Lin-Sca-C-Kit+ (LSK) compartment and

progenitors (HSC-hematopoietic stem cell, MPP-multipotent progenitor, HPC-hematopoietic progenitors, CMP-common myeloid progenitor, LMPP-lymphoid-primed multipotent progenitor, CLP-common lymphoid progenitor, GMP-granulocyte/monocyte progenitor, MEP-Megakaryocyte/erythrocyte progenitor). (**d**) Schematic showing erythrocytosis and monopoiesis (Pre-GM-pre-granulocyte/monocyte precursor, Pre-MegE - megakaryocyte-erythrocyte precursor, Pre-CFU-E - pre-colony forming unit erythroid, CFU-E - Colony forming unit erythroid, GMP - granulocyte-monocyte precursor, MkP- megakaryocyte precursor). gating strategy for bone marrow common 5 myeloid progenitor progeny on CD41⁻CD16/32⁻ cells. (**e**) Blood hematocrit at 24 hours (n=3/ group, data representative of 2 experiments) or (**f**) 5 days in mice treated with LPS and housed in normoxia (N) and hypoxia (H) (n=3/group, data representative of 2 experiments) was measured. **a, b** Mean+SD. Statistical testing performed using one-way ANOVA with Tukey's multiple comparisons test, **e, f** Mean+SD. **f,** statistical testing 10 performed using unpaired two-sided t-test.

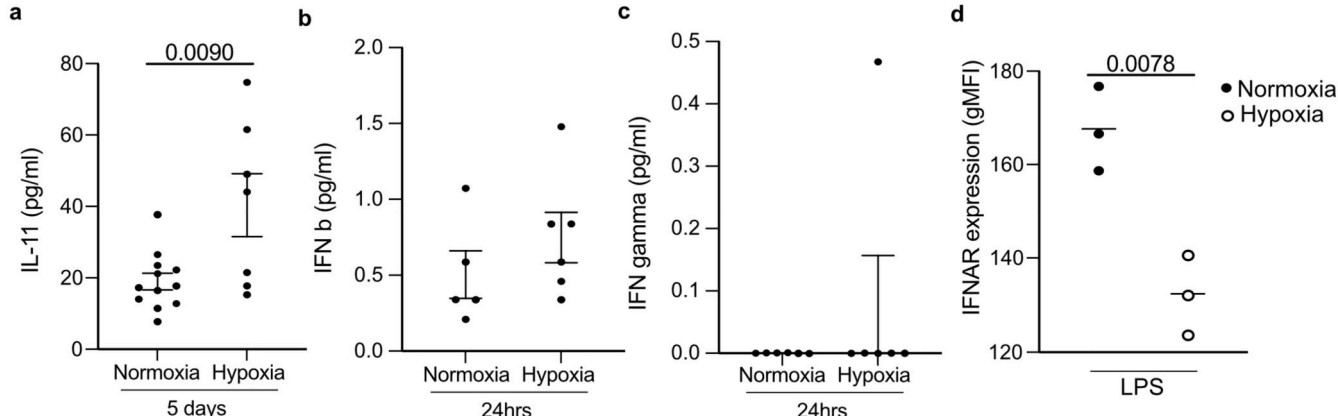

**Extended Data Fig. 5. Hypoxia elevates circulating IL-11 levels without altering IFN beta and IFN gamma levels and reducing IFNAR expression in circulating classical monocytes.**
(**a**) Serum IL-11, (**b**) IFN beta and (**c**) IFN gamma level mice treated with nebulised LPS and housed in normoxia (N) or hypoxia (H), for 24 hours or 5 days (as indicated on figure), were measured by ELISA as per manufacturers' instructions. Data mean±SEM. Each data point represents an individual mouse. (**d**) IFNAR expression was measured by flow cytometry in classical blood monocytes in mice treated with nebulized LPS and housed in either normoxia (N LPS) or hypoxia (H LPS) for 24 hours. **a, d,** Statistical testing unpaired two-sided t-test.

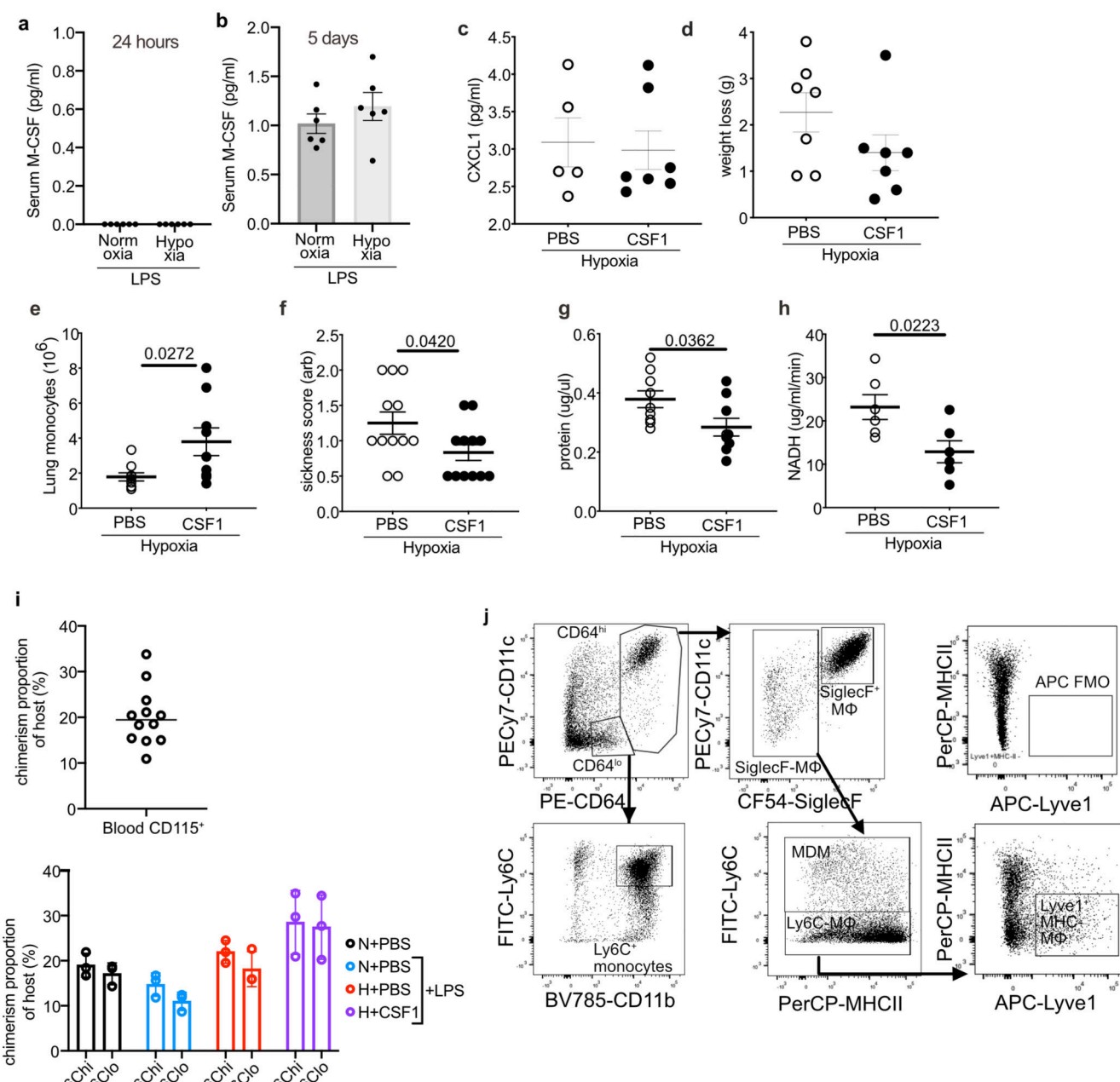

**Extended Data Fig. 6. Circulating CSF1 is unchanged in hypoxic ALI and exogenous CSF1 improves injury outcomes altering the CD64^hiSiglecF^- Mφ phenotype**

Serum MCSF (CSF1) from LPS-challenged mice housed in normoxia (N) or hypoxia (H) for 24 hours (**a**) or 5 days (**b**) was measured. (**c**) BAL CXCL1 measured in LPS-challenged mice housed in hypoxia for 5 days and treated with PBS or CSF1-Fc (H CSF1). (**d**) weight loss from baseline in hypoxic LPS-induced ALI treated with PBS (H PBS) or CSF1-Fc (H CSF1). (**e**) Lung monocyte numbers, (**f**) arbitrary sickness scores, (**g**) BAL protein and (**h**) LDH activity (as measured by NADH) measured at 48 hours in mice with virally-induced ALI housed in hypoxia and treated with PBS or CSF1-Fc (H 10 CSF1).

(**i**) Baseline blood chimerism (proportion of donor cells relative to host) of circulating monocytes in lung-protected chimeras prior to ALI induction and chimerism of monocytes based on Ly6C expression post-LPS. (**j**) Lung cDC1 (gated on Alive CD45$^+$Lin-CD64$^-$ CD11c$^+$Cd103$^+$) chimerism and counts and (**k**) cDC2 (gated on Alive CD45$^+$Lin-CD64$^-$ CD11c$^+$Cd103$^-$ CD11b$^+$) chimerism and counts. Chimerism relative to 15 blood monocyte chimerism. (**l**) *Il10* expression was measured by NanoString platform analysis in MHC-lung macrophages from LPS-challenged mice housed in hypoxia for 5 days and treated with PBS and compared to Lyve1$^+$MHCII- of LPS-challenged mice, housed in hypoxia and treated with CSF1-Fc. (**m**) Representative dot plots of lung digests showing gating strategy for identification of the different monocyte and macrophage 20 populations in the lung gated on Live Singles CD45$^+$Ly6G$^-$ cells, and including Lyve1$^+$MHC-CD64$^{hi}$SiglecF$^-$M$\phi$, and associated APC FMO control including Lyve1$^+$MHC- CD64$^{hi}$SiglecF$^-$M$\phi$, and associated APC FMO control. Ifnar$^{/-}$ (KO) mice were nebulised with LPS and treated with PBS (KO PBS) or CSF1-Fc (KO CSF1). Mice were sacrificed on day 5 and (**n**) blood monocyte and (**o**) lung Ly6C$^+$ monocytes were quantified by flow cytometry. **c, d** Mean+SD, **e-h** mean±SEM. **c-h, j, k** 2 pooled experiments, **l** representative of 3 experiments, **n, o** representative of 2 experiments. Statistical testing **e-h, n, o** unpaired two-sided t-test, **k,** one-way ANOVA with Tukey's multiple comparisons test.

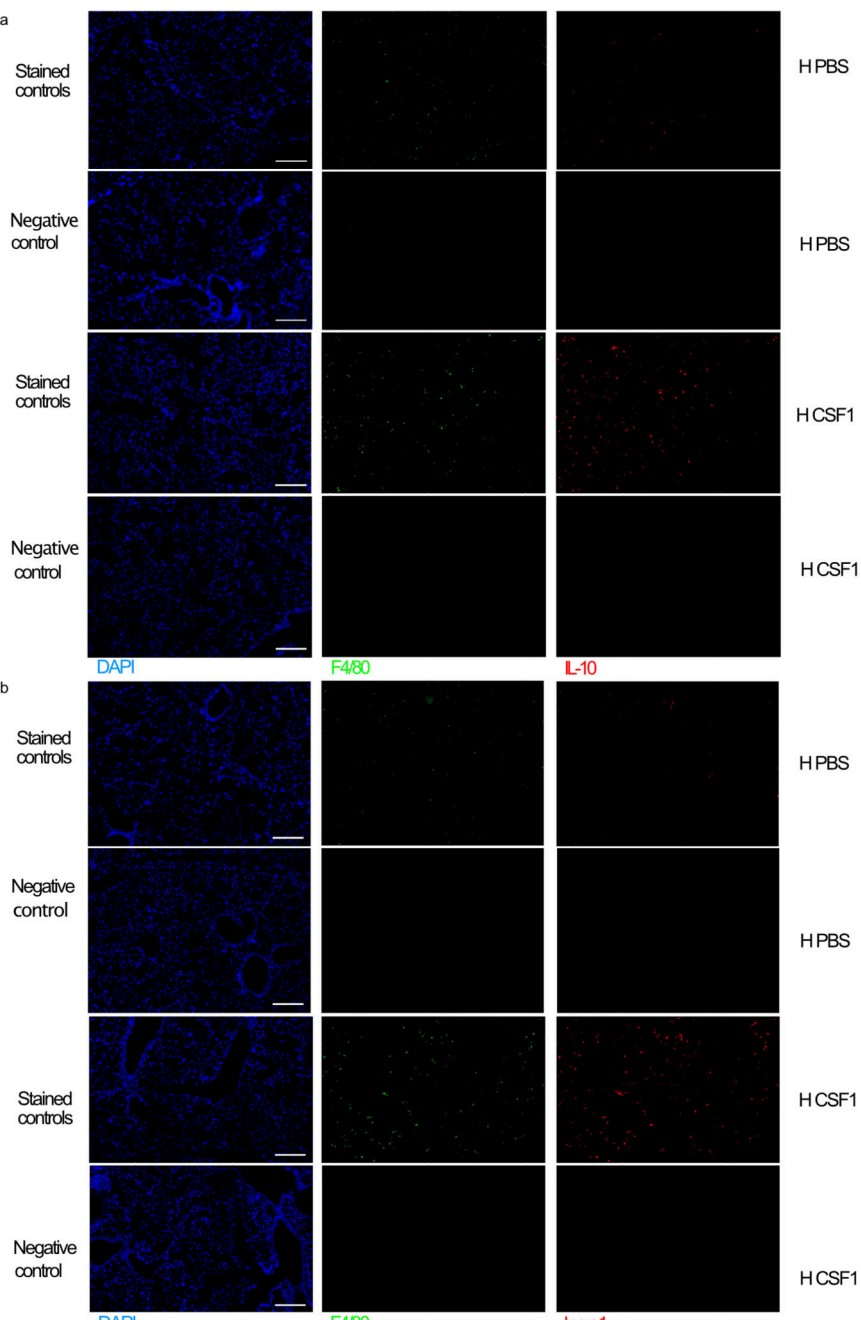

**Extended Data Fig. 7. Lung immunohistochemistry tiled separate-channel images with negative controls.**

**(a, b)** Representative images of separate channels for stained tissue and negative controls (no primary antibody) for the indicated antibodies in lung sections from mice 5 housed in hypoxia post-LPS and treated with either PBS or CSF1 as indicated (n=3/ group). Scale bar represents 200μm.

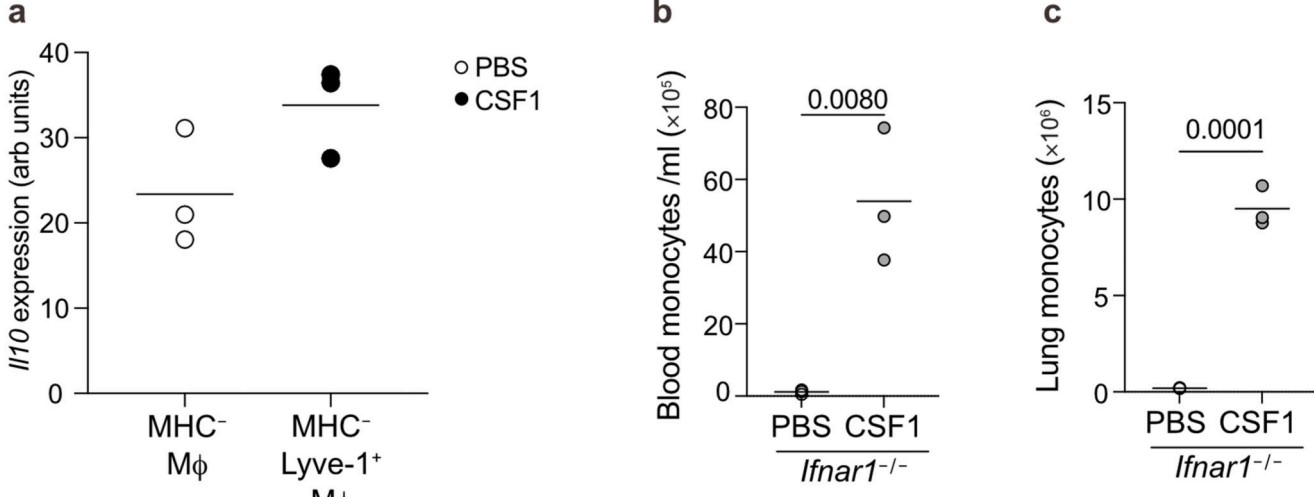

**Extended Data Fig. 8. Treatments with CSF1-Fc elevates Il10 expression in Lyve1[+]MHCII[-] macrophages and increases blood and lung monocyte numbers in Ifnar1[-/-] mice**

(**a**) Il10 expression was measured by NanoString platform analysis in MHC[-] lung macrophages from LPS-challenges mice, housed in hypoxia for 5 days and treated with PBS and compared to Lyve1[+]MHCII[-] of LPS-challenged mice, housed in hypoxia and treated with CSF1-Fc. (**b,c**) Ifnar1[-/-] mice were nebulised with LPS and treated with PBS or CSF1-Fc. Mice were sacrificed on day 5 and (**b**) blood monocyte and (**c**) lung Ly6C[+] monocytes were quantified by flow cytometry. (**a-c**) representative of 2 experiments. Statistical testing (**a-c**) unpaired two-sided t-test.

## Supplementary Material

Refer to Web version on PubMed Central for supplementary material.

## Acknowledgments

We thank D Hume for providing the CSF1-Fc used in these experiments. We thank Thomson Bioinformatics, 27 Strathalmond Road, Edinburgh, United Kingdom, for analysing the NanoString data. Flow cytometry data were generated with support from the QMRI Flow Cytometry and Cell Sorting Facility, University of Edinburgh. We thank the Royal Infirmary of Edinburgh Critical Care Research Team for their assistance in recruiting, consenting and obtaining samples from patients with ARDS. This work was funded by a Wellcome Trust Senior Clinical fellowship awarded to S.R.W. (098516 and 209220), Wellcome Trust Post-doctoral Training Clinical Fellowship (110086) and a Wellcome Trust iTPA grant (PIII052) awarded to A.S.M. and was partly funded by UKRI/NIHR through the UK Coronavirus Immunology Consortium (UK-CIC). C.C.B holds a Sir Henry Dale Fellowship jointly funded by the Wellcome Trust and the Royal Society (Grant Number 206234/Z/17/Z). K.K.'s laboratory is supported by grants from Cancer Research UK (C29967A/14633 and C29967/A26787, Medical Research Council, The Barts Charity, The Kay Kendall Leukaemia Fund and Blood Cancer UK. J.S. is a NIHR Senior Investigator. The views expressed in this article are those of the author(s) and not necessarily those of the NIHR, or the Department of Health and Social Care. For the purpose of open access, the author has applied a CC BY public copyright license to any Author Accepted manuscript version from this submission.

## Materials Availability

This study did not generate new unique reagents.

All nanostring data shown in this manuscript has been deposited in Geo. Accession codes: GSE200429, GSE200549, GSE200558

All proteomic data generated in this project has been deposited on Pride. Accession codes: PXD033151

Analyses, including the drawing of heatmaps and volcano plots were carried out in R using the package ggplot2 (https://cran.r-project.org/web/packages/ggplot2/index.html). Analysis of datasets was carried out by Thomson Bioinformatics, Edinburgh, UK.

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

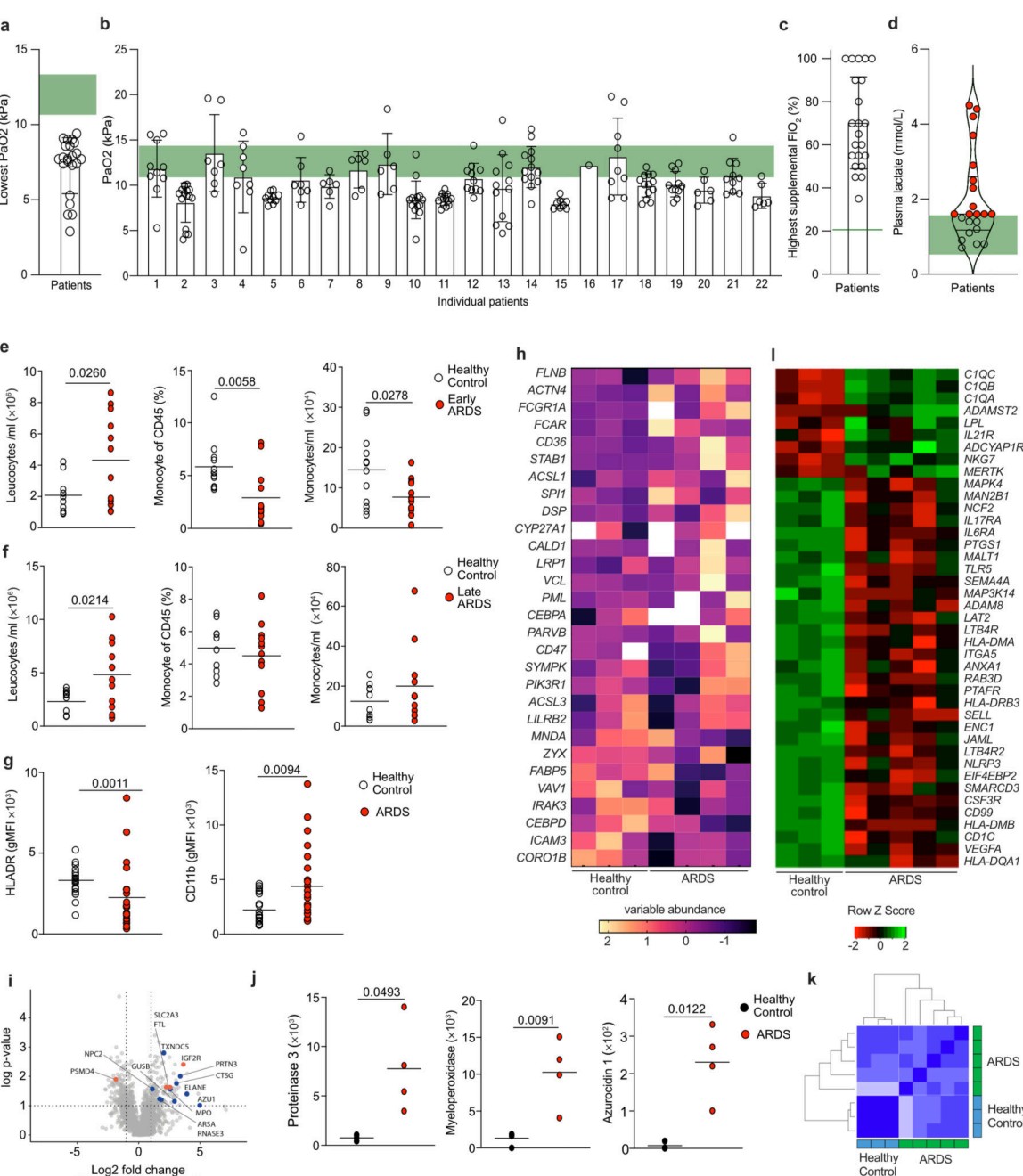

**Figure 1. Patients with ARDS are monocytopenic early in the disease with phenotypically distinct circulating monocytes.**

**a,b,** Lowest (**a**) and all (**b**) partial pressures of oxygen (PaO$_2$) from clinical arterial blood samples from ARDS patients, 24 hours preceding research blood sampling, (green: normal range). **c, d,** Highest recorded fractional inspired oxygen (FiO$_2$) (**c**) and highest recorded arterial plasma lactate level within 24hours of research sampling (**d**) in ARDS patients, (red samples: lactate  upper limit of normal, green: normal local reference (0.5-1.6mmol/ L)). **e,** Blood leucocyte counts, monocyte proportions and monocyte counts from ARDS

patients, collected within 48 hours of diagnosis (early ARDS) and a healthy volunteer cohort (HC). **f,** Blood leucocyte count, monocyte proportions and monocyte counts from ARDS patients collected between 48hours and 7days (late ARDS) and HC. **g,** Monocyte HLADR and CD11b expression in HC and ARDS patients. **h,** CD14$^{++}$CD16$^-$ classical monocytes proteomic data from ARDS patients, relative to HC, for proteins associated with a human monocyte *in vitro* hypoxic gene signature[13]. **i,** Classical (CD14$^{++}$CD16$^-$) monocytes proteome volcano plot from HC and ARDS patients. Significantly-upregulated granule-associated proteins in ARDS patients versus HC (blue), a sample of known hypoxia-regulated proteins (orange). **j,** classical monocytes proteinase 3, myeloperoxidase and azurocidin 1 copy numbers in HC and ARDS patients. **k,** Pearson correlation and **l,** heatmap of differentially expressed genes from HC and ARDS patient blood monocytes Data in **a-c** mean±SD, **e,f,** expressed as median, **g, j,** shown as mean. **a, c-g, j** each data point represents one patient/ HC, **b** each data point represents one independent clinical sample. Statistical testing: **e-f, j** unpaired two-tailed Students t-test, **g** Mann-Whitney test.

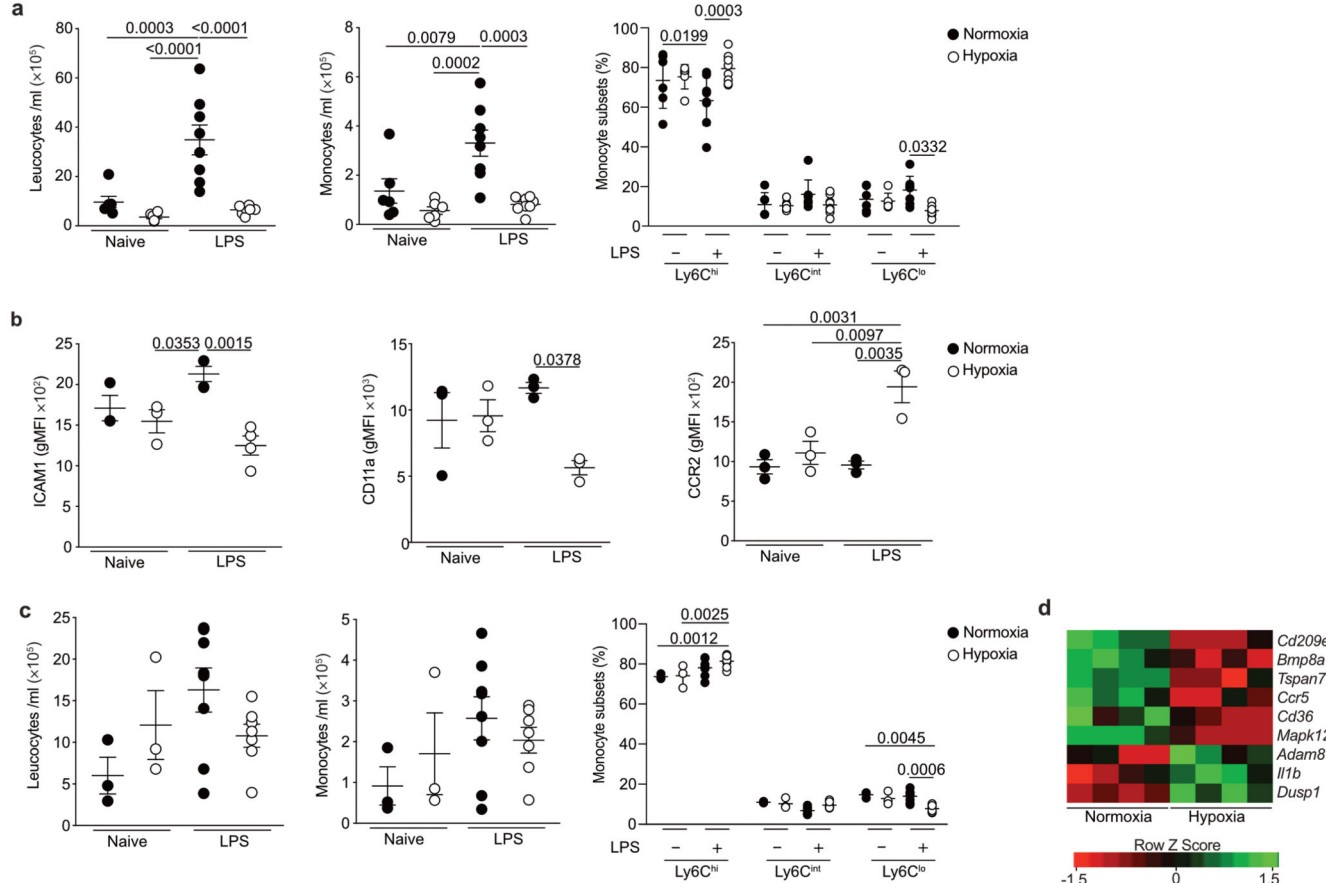

**Figure 2. Hypoxic acute lung injury replicates early monocytopenia in mice and alters the circulating monocyte phenotype.**

**a,** Blood leucocyte counts, monocyte counts and proportion of blood monocyte subgroups in naïve or LPS-treated mice housed in normoxia or hypoxia for 24 hours. **b,** Classical monocyte (CD115+CD11b+Ly6C^hi) surface expression of ICAM, CD11a and CCR2 at 24hours post-LPS. **c,** Blood leucocyte counts, monocyte counts and proportions of monocyte sub-populations in naïve or LPS-treated mice housed in normoxia or hypoxia for 5 days post-LPS. **d,** Differentially expressed genes in circulating classical monocytes from LPS-treated mice housed in normoxia or hypoxia for 5 days. Data: mean±SEM. **a and c** data pooled from 2 independent experiments. Each data point represents an individual mouse. Statistical testing: **a,b** one-way ANOVA with Tukey's multiple comparisons test.

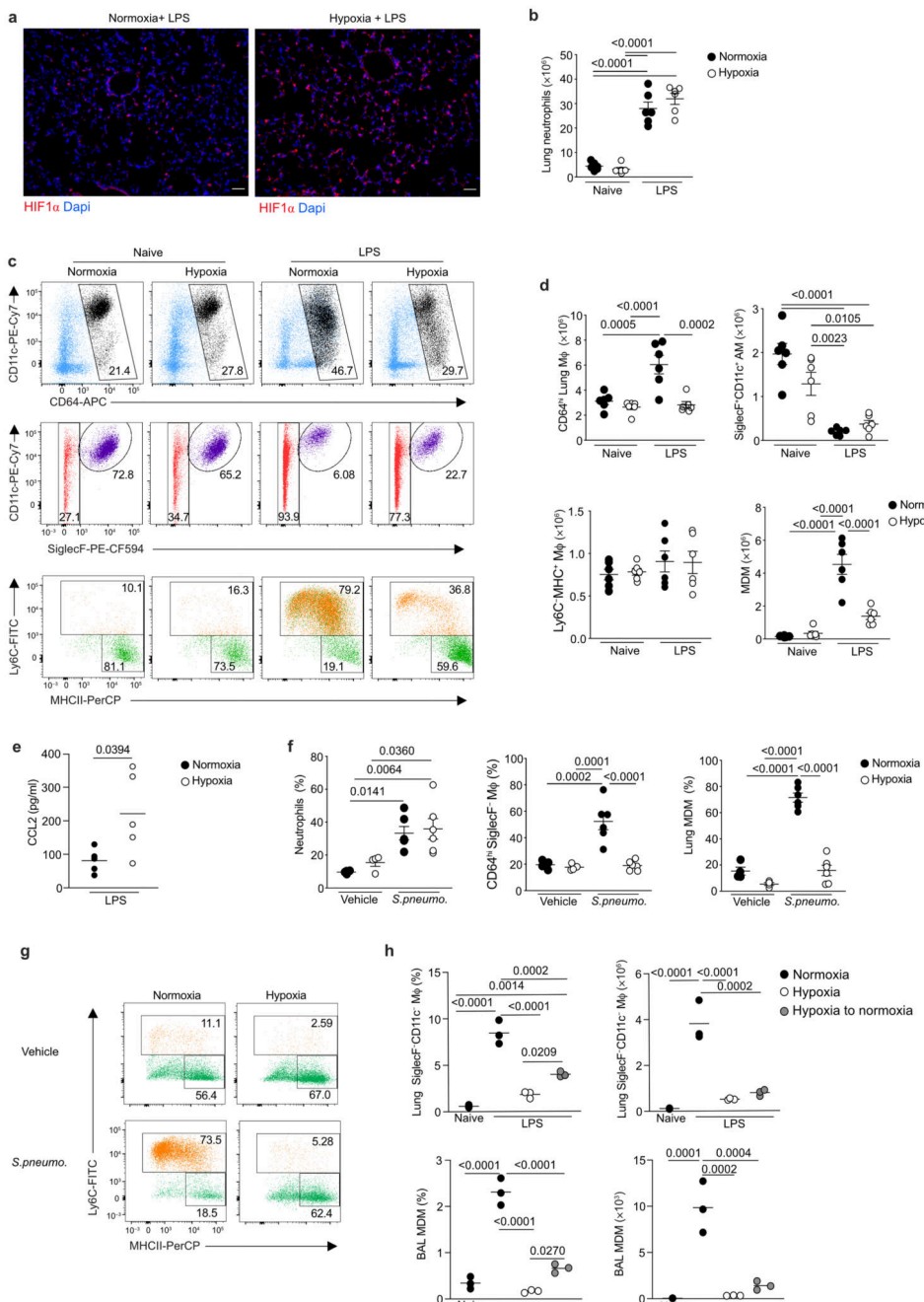

**Figure 3. Systemic hypoxia hampers expansion of the CD64<sup>hi</sup>SiglecF<sup>-</sup> Mϕ niche in ALI and *Streptococcus pneumoniae* infection.**

**a,** Representative lung immunofluorescence HIF1α and DAPI expression from LPS-challenged mice, housed in normoxia or hypoxia for 24 hours. Scale bar represents 50μM. **b-d,** naïve or LPS-challenged mice were housed in normoxia (N) or hypoxia (H) for 24 hours (n=6/group). **b,** Absolute numbers of live lung neutrophils. **c,** Representative dot plots of the CD64<sup>hi</sup> macrophage compartment (top), CD64<sup>hi</sup>SiglecF<sup>+</sup>CD11c<sup>+</sup> AM (middle), and Ly6C and MHCII expression by CD64<sup>hi</sup>SiglecF<sup>-</sup> macrophages (bottom). **d,** Absolute number of

CD64$^{hi}$ macrophages, CD64$^{hi}$SiglecF$^+$CD11c$^+$ AM, Ly6C$^-$MHCII$^+$ lung macrophages and CD64$^{hi}$SiglecF$^-$Ly6C$^+$ MDM. **e,** BAL CCL2 levels from LPS-challenged mice housed in normoxia or hypoxia for 24 hours. **f-g,** *Streptococcus pneumoniae (S.pneumo)-inoculated* mice (n= 6/ group) or vehicle control mice (Veh, n=4/ group) were housed in normoxia or hypoxia until 24 hours post-inoculation. **f,** Neutrophil proportion of total lung leucocytes, CD64$^{hi}$SiglecF$^-$ M$\phi$ proportion of lung CD64$^{hi}$ macrophages and MDM proportion of CD64$^{hi}$SiglecF$^-$ macrophages. **g,** representative dot plots of CD64$^{hi}$SiglecF$^-$ macrophage sup-populations. **h,** Lung CD64$^{hi}$SiglecF$^-$ macrophage proportion of total leucocytes, lung CD64$^{hi}$SiglecF$^-$ macrophage absolute numbers, BAL MDM proportion and BAL MDM absolute numbers**(s)** in naïve or LPS-challenged mice housed in normoxia or hypoxia for 48 hours, or housed for 24 hours in hypoxia, followed by 24 hours of normoxia (hypoxia to normoxia)) (n=3/group). Data: mean±SEM. **a** representative of n=3/ group. **b-d,e** data pooled from 2 independent experiments. **g** representative of 2 independent experiments. Each data point represents an individual mouse. Statistical testing: **b,d,f,h** one-way ANOVA with Tukey's multiple comparisons test, **e** unpaired two-tailed t-test.

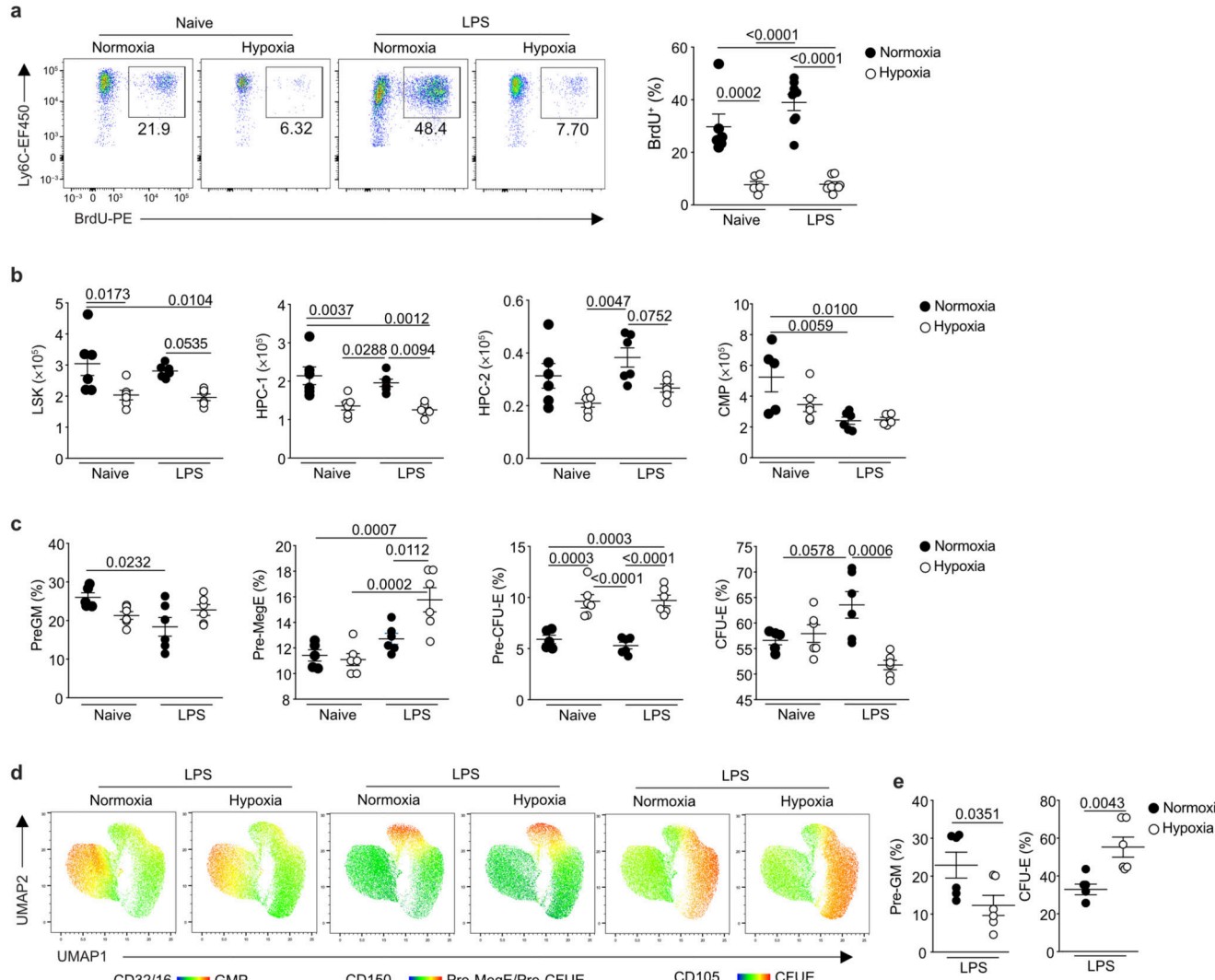

**Figure 4. Systemic hypoxia alters bone marrow hematopoiesis towards increased erythropoiesis.**
**a,** Naïve or LPS-challenged mice were housed in normoxia or hypoxia for 24 hours.
Representative dot plots and proportion of BrdU+ monocytes in mice pulsed with BrdU
for the last 12 hours, gated on live CD45+Lin-(CD3/CD19/Ly6G) CD115+Ly6Chi (naïve,
n=5-6; LPS-treated, n=8 in). **b,** Absolute numbers of bone marrow Lin-Sca-C-Kit+ (LSK),
CD48+CD150- HPC-1 and CD48+CD150+ HPC-2 (n=6/ group) and Lin-cKit+Sca1-CD127-
CD16/32-CD34+ common myeloid precursor (CMP) (n=5-6/ group). **c,** proportion of
bone marrow pre-granulocyte/monocyte 10 precursor (pre-GM), megakaryocyte-erythrocyte
precursor (Pre-MegE), pre-colony forming unit erythroid (pre-CFU-E) and colony forming
unit erythroid (CFU-E) CD41-CD32/16- (n=6/group). **d,e,** Representative UMAP analysis of
bone-marrow cells gated on live CD45+Lineage-Sca1-C-Kit+CD41-CD32/16- cells **(d)** and
summary data of proportions of pre-GM and CFU-E **(e)** measured in the BM of mice treated
with LPS and housed in normoxia (N) or hypoxia (H) for 5 days (n=6/ group). Data shown
as mean ±SEM. Each data point represents an individual mouse. **a-e,** data pooled from 2

independent experiments. Statistical testing **a-c,** by one-way ANOVA with Tukey's multiple comparisons test, **e** un-paired two-tailed Student's t-test.

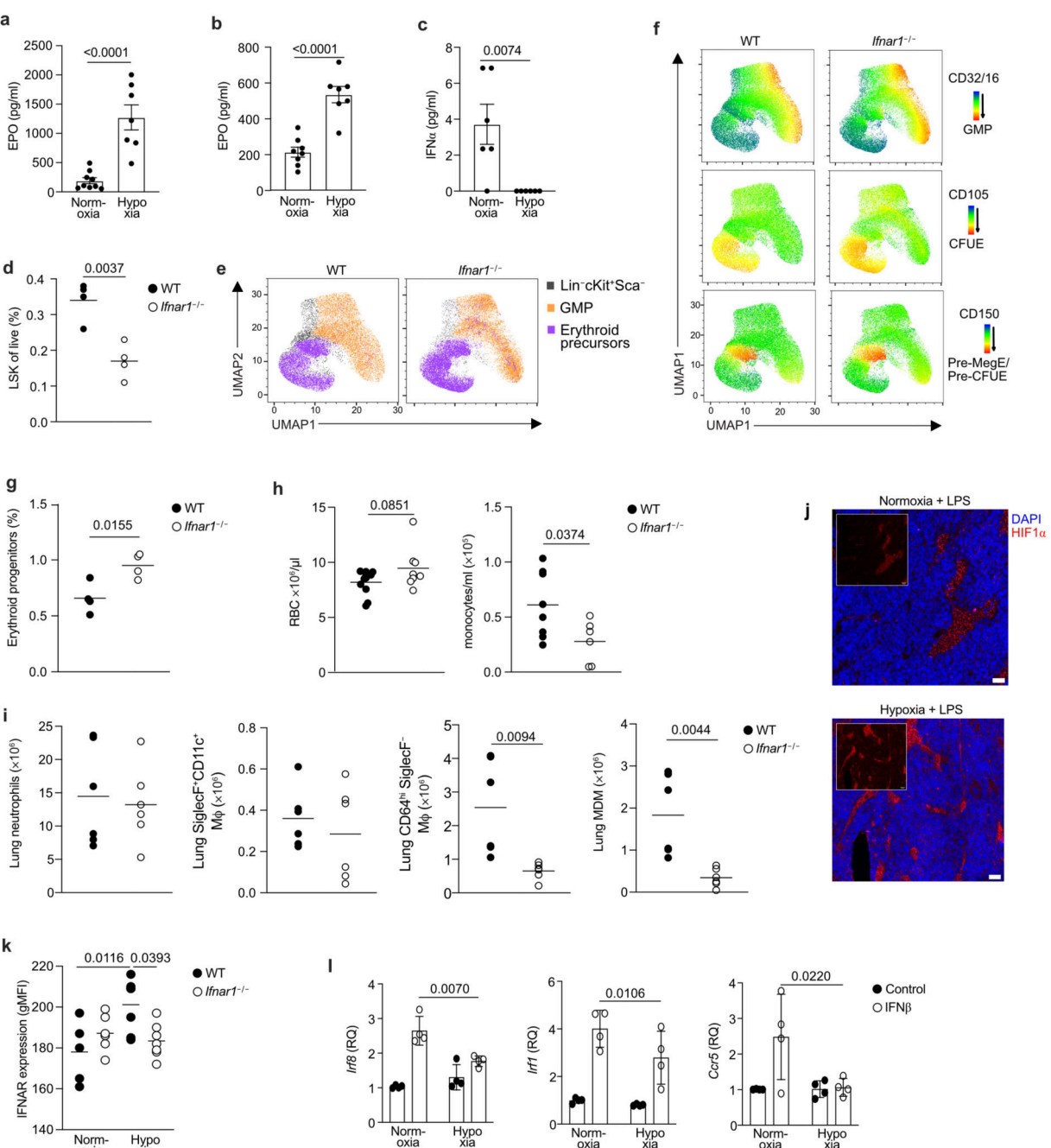

**Figure 5. Hypoxia regulates type I interferon responses hindering lung CD64^hiSiglecF^- Mϕ expansion in response to LPS**

**a, b,** Serum erythropoietin (EPO) levels from LPS-challenged mice and housed in normoxia or hypoxiafor 24 hours **(a)** (n=9 normoxia, n=7 hypoxia) or 5 days **(b)** (n=7/ group). **c,** Serum IFN-α from LPS-challenged mice housed in normoxia or hypoxia for 24 hours (n=6/ group). **d,** Proportion of Lin^-ckit^+Sca^+ (LSK) cells in the BM of wild-type (WT) or *Ifnar1^-/-* mice 24 hours post-LPS-challenge (n=4/ group). **e,** Manual gating of erythroid precursors and GMP in LSK cells displayed on UMAP projection. **f,** Representative

expression of CD32/16 (GMP marker) and CD150/CD105 (erythroid progenitor-associated markers) in LSK cells using Pronk gating strategy[26] displayed on UMAP projection. **g,** Proportion of erythroid progenitor cells (combined MEP, preCFUe and CFUe) in WT and *Ifnar1*$^{-/-}$ BM 24h post-LPS (n=4/ group). **h,** Peripheral red blood cell (RBC) (n=10 WT/ 8 KO) and monocyte counts at day 5 post-LPS in WT and KO mice. **i** Neutrophils, CD64$^{hi}$SiglecF$^+$CD11c$^+$macrophages, CD64$^{hi}$SiglecF$^-$ macrophages and MDM 15 numbers in the lung 24hrs post-LPS in WT and *Ifnar1*$^{-/-}$ mice. **j,** Representative HIF1α and DAPI expression in the femoral BM from LPS-challenged mice housed in normoxia (N LPS) or hypoxia (H LPS) for 24hrs. Scale bar represents 20 μm. **k,** IFNAR expression in the BM LSK in naïve (n=5-6/ group) or LPS-treated mice (n=6/ group) housed in normoxia (N) or hypoxia (H) for 24hrs. **l,** qPCR *Irf8, Irf1* and *Ccr5* expression (normalised to actin-ß) in BM cells from naïve mice cultured in normoxia or hypoxia for 4 hours+/-IFN-ß (n=3/ group). Data:mean±SEM. All data points represent individual mice. Statistical testing: **a-d, g-i** unpaired two-sided Student's t-test, **k** One-way ANOVA Tukey's multiple comparisons test, **l** Two-way ANOVA with Šídák's multiple comparisons post-test. **a-c, d-i** representative of two independent experiments, **j** representative of n=3/ group, **l** two independent pooled experiments.

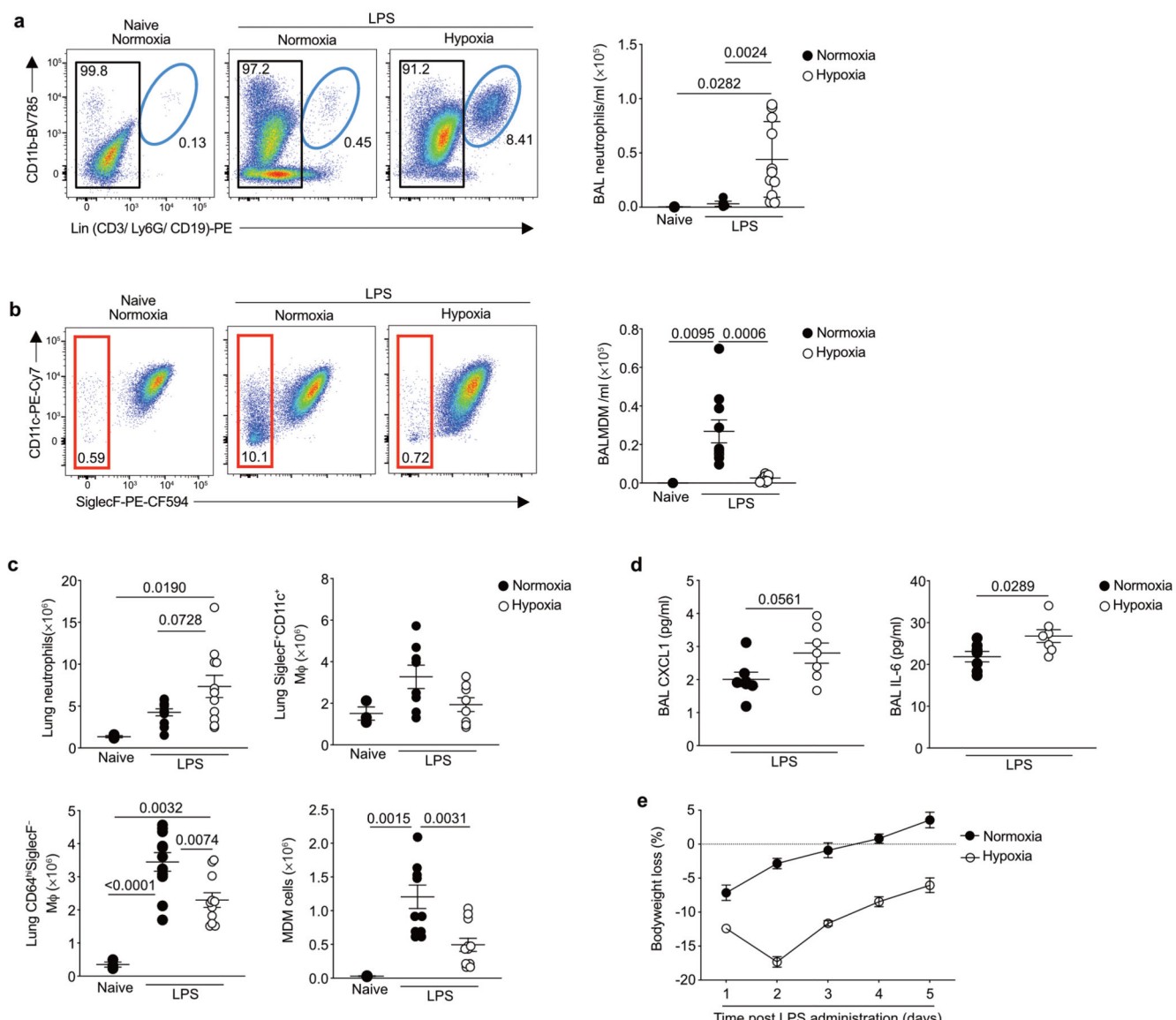

**Figure 6. Ongoing CD64^hiSiglecF^- Mϕ expansion failure is associated with inflammation persistence in hypoxic ALI.**

**a-b,** Representative dot plots and absolute numbers of BAL neutrophils (**a**) and CD45^+Ly6G^- CD64^hiSiglecF^- MDM gated on CD45^hi cells (**b**) of mice treated with LPS and housed in normoxia or hypoxia for 5 days. **c,** Lung neutrophil, CD64^hiSiglecF^+CD11c^+ macrophages, CD64^hiSiglecF^-macrophages and CD64^hiSiglecF^-Ly6C^+ MDM numbers in LPS-challenged mice housed in normoxia or hypoxia for 5 days. **d,** BAL CXCL1 and IL-6 levels in mice treated with LPS and housed in normoxia or hypoxia for 5 days (n=6 N LPS, n=7 H LPS). **e,** Daily weight changes from baseline in LPS-challenged mice housed in normoxia or hypoxia for 5 days (n=4/group). Data shown as mean ±SEM. Each data point represents an individual mouse. Statistical testing: **a-c** one-way ANOVA with Tukey's multiple comparisons test, **d** by unpaired two-tailed Student's t-test. Data in **a-c** pooled from 3 independent experiments, data in **d** pooled from 2 independent experiments.

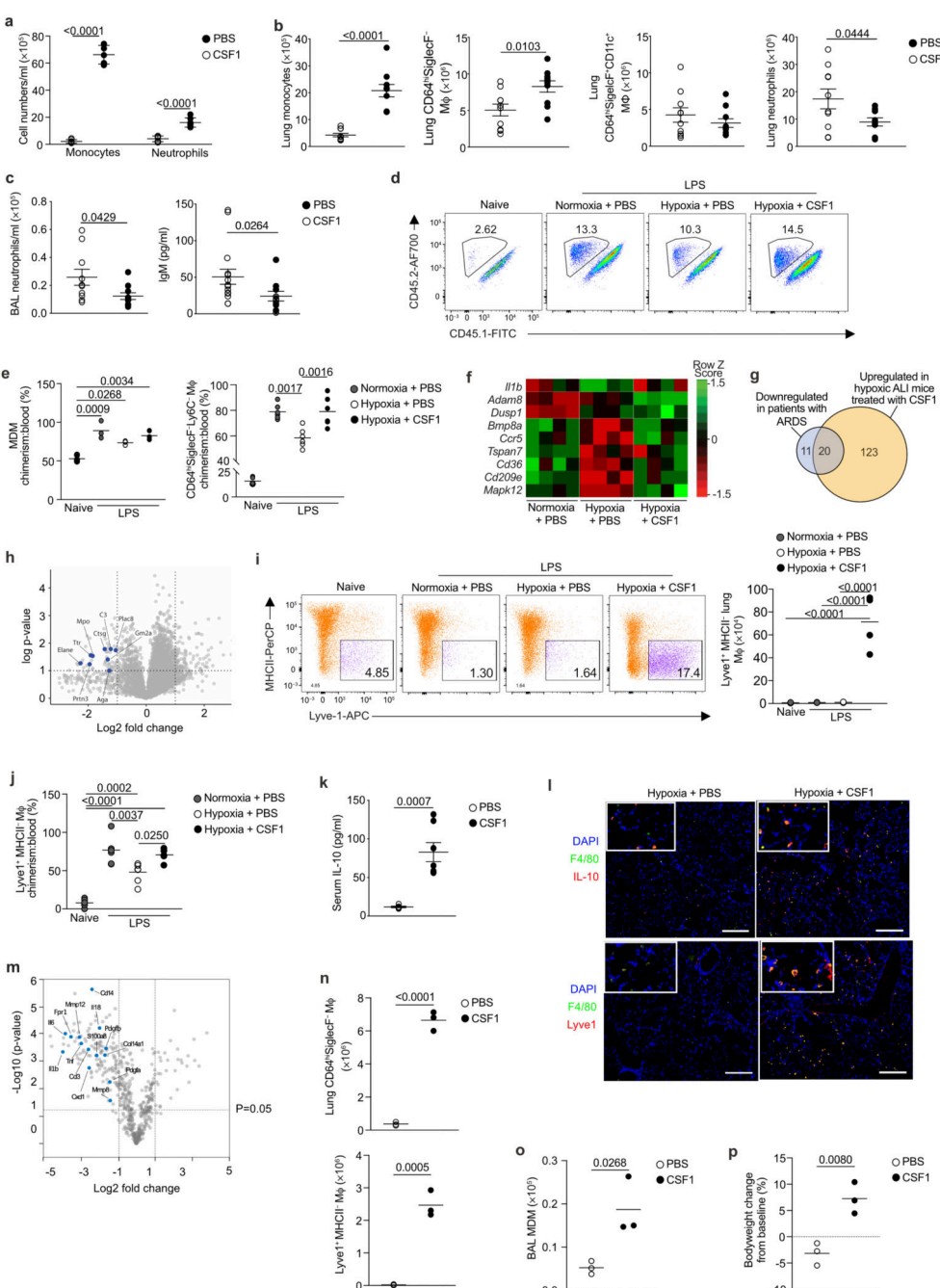

**Figure 7. CSF1 rescues the hypoxic monocytopenia driving inflammation resolution.**
Monocyte and neutrophil blood counts (**a**), lung monocyte, CD64$^{hi}$SiglecF$^-$ macrophage, CD64$^{hi}$SiglecF$^+$ CD11c$^+$ macrophage and neutrophil counts (**b**), and absolute numbers of neutrophils and IgM levels in the BAL (**c**) in hypoxic LPS-challenged mice treated with 4 daily injections of PBS- or CSF1-Fc. **d, e** Representative CD45.2 and CD45.1 expression (**d**) and CD64$^{hi}$SiglecF$^-$Ly6C$^+$ MDM:blood monocyte chimerism proportion and CD64$^{hi}$SiglecF$^-$Ly6C$^-$ macrophages:blood monocyte chimerism proportion (**e**) in lung-protected, naïve or LPS-challenged mice normoxia- or hypoxia-housed and treated with PBS

or CSF1-Fc. **f,** Differentially expressed genes in PBS- or CSF1-Fc treated LPS challenged mice in Ly6C$^{hi}$ blood monocytes at day 5 post-LPS challenge. **g,** Overlap between differentially downregulated genes in ARDS blood monocytes and genes upregulated in CSF1-Fc-treated mice relative to PBS-treated mice. **h,** Comparison of Ly6C$^{hi}$ blood monocyte proteomes from hypoxic LPS-challenged CSF1-Fc-treated mice relative to the PBS-treated counterparts (granule-associated proteins identified). **i,** Representative Lyve1 expression and number of lung CD64$^{hi}$SiglecF$^-$Ly6C$^-$Lyve1$^+$MHCII$^-$ macrophages in naïve or LPS-challenged mice housed in normoxia or hypoxia and treated with PBS or CSF1-Fc for 5 days. **j,** chimerism of CD64$^{hi}$SiglecF$^-$Ly6C$^-$Lyve1$^+$MHCII$^-$ macrophages to blood monocyte in lung-protected, LPS-challenged chimeras housed in normoxia or hypoxia, treated with PBS or CSF1-Fc. **k-l,** Serum IL-10 **(k)** and representative tiled immunofluorescence of lung sections stained for F4/80, IL-10 and DAPI **(l)** and F4/80, Lyve1 and DAPI in LPS-challenged mice housed in hypoxia for 5 days, and treated with CSF1-Fc or PBS. Scale bar represents 200 μm. **(m)** Differentially regulated genes in CD64$^{hi}$SiglecF$^-$Lyve1$^+$MHCII$^-$macrophages from LPS-challenged CSF1-Fc-treated mice relative to CD64$^{hi}$SiglecF$^-$MHCII$^-$macrophages from PBS-treated counterparts, housed in hypoxia for 5 days. **n-p,** Total lung CD64$^{hi}$SiglecF$^-$ macrophages and CD64$^{hi}$SiglecF$^-$Ly6C$^-$Lyve1+MHCII$^-$ macrophages **(n)** BAL MDM **(o)** and bodyweight change (relative to baseline) **(p)** in LPS-challenged *Ifnar1$^{-/-}$* mice treated with PBS (*Ifnar1$^{-/-}$* PBS) or CSF1-Fc (*Ifnar1$^{-/-}$* CSF1) for 5 days.. Data:mean±SEM. Each data point represents an individual mouse. Statistics **b-c, k, n-p** unpaired two-sided Students t-test. **i, j** One-way ANOVA with Tukeys post-test, **e** two-tailed Mann-Whitney following D'Agostino & Pearson normality test. **b-c** data pooled from 3 independent experiments, **f, k, j, n-p** data pooled from 2 independent experiments. **g** all genes fold change >1, except H2-DMa, H2-DMb2, IL-17Ra, Nlrp3 where fold-change >0.5, p<0.05.

**Table 1**

**ARDS patient cohort clinical characteristics and demographics**

Patient cohort and clinical characteristics demonstrates heterogeneity of etiology and evidence of clinically significant ongoing hypoxemia.

| | Early ARDS (n=11) | Late ARDS (n=11) |
|---|---|---|
| Age (years) | 58.8 (±11.5) | 56.9 (±13.6) |
| Proportion females - number (%): | 6 (55) | 5 (45) |
| Body mass index (kg/m$^2$): | 35.8 (±11.6) | 31.1 (±4.2) |
| APACHE2 [*] score: | 20.4 (±7.2) | 18.8 (±8.6) |
| Pulmonary ARDS- number (%): | 9 (82) | 7 (64) |
| *Pulmonary aetiologies[†]- number (%):* | | |
| positive bacterial culture: | 4 (36) | 4 (36) |
| positive viral PCR[‡]: | 2 (18) | 1 (9) |
| positive mycology: | 2 (18) | 2 (18) |
| no positive microbiology samples: | 2 (18) | 1 (9) |
| *Extra-pulmonary aetiologies- number (%):* | | |
| faecal peritonitis: | 1 (9) | |
| mediastinal soft tissue infection: | 1 (9) | |
| bacteraemia: | | 1 (9) |
| biliary Sepsis: | | 1 (9) |
| retropharyngeal Abscess: | | 1 (9) |
| non-infective: | | 1 (9) |
| *Index of tissue hypoxia:* | | |
| *Reference PaO$^2$: 11.1-14.4kPa[§]* | | |
| lowest PaO$^2$ in hospitalisation preceding sampling (kPa): | 4.38 (±1.72) | 6.42 (±2.23) |
| lowest PaO$_2$ 24 within hours before sampling (kPa): | 6.69 (±2.11) | 8.15 (±1.90) |
| *Reference FiO$_2$. 21%* | | |
| highest FiO$_2$ in 24 hours before sampling (%): | 68.1 (±21.2) | 72.3 (±22.3) |
| *Reference arterial lactate: 0.5-1.6mmol/L[§]* | | |
| highest lactate within 24 hours before sampling (mmol/L): | 2.45 (±1.28) | 1.56 (±0.96) |
| *Receipt of organ supportive therapies- number (%):* | | |
| invasive mechanical ventilation: | 7 (64) | 7 (64) |
| vasopressors: | 7 (64) | 6 (55) |
| renal replacement therapy: | 1 (9) | 1 (9) |
| *Receipt of additional medications- number (%):* | | |
| dexamethasone[¶]: | 0 (0) | 0 (0) |
| lopinavir or ritonavir: | 1 (9) | 0 (0) |
| tocilizumab: | 0 (0) | 0 (0) |
| hydroxychloroquine: | 1 (9) | 1 (9) |

Plus-minus data values refer to Mean ± Standard Deviation.

*Acute Physiology and Chronic Health Evaluation Score 2 (APACHE2);

†One patient in each group returned mixed fungal and bacterial cultures, which could not be causatively differentiated;

‡Two patients in the "Early" group and one patient in the "Late" group were SARS-CoV-2 positive;

§Reference ranges as indicated by local health board (NHS Lothian).

¶No other corticosteroids were administered

**Table 2**

**CSF1 upregulates genes that were downregulated in human ARDS monocytes**

Differentially expressed genes found in ARDS patients versus healthy control were compared to differentially upregulated genes found in mouse classical monocytes from hypoxic LPS-challenged CSF1-Fc-treated mice, relative to their PBS controls.

| | | | |
|---|---|---|---|
| Adam8 | Il17Ra | Malt1 | Ptafr |
| Anxa1 | Il6Ra | Man2b1 | Ptgs1 |
| Cd99 | Itga5 | Map2k4 | Rab3d |
| Eifebp2 | Lat2 | Ncf2 | Sema4a |
| H2-DMb2 | Ltb4r1 | Nlrp3 | H2-Dma |

**Table 3**

**Antibodies used**

| Antibody | Clone | Catalogue | Lot | Fluorophore | Source | dilution |
|----------|-------|-----------|-----|-------------|--------|----------|
| CD16 | eBioCD16 | 1-9161-71 | 4304474 | FITC | Ebioscience | 1:20 |
| CD3 | OKT3 | 317308 | B256076 | PE | Biolegend | 1:80 |
| CD56 | HCD56 | 318306 | B252053 | PE | Biolegend | 1:80 |
| CD19 | HIB19 | 302254 | B227178 | PE | Biolegend | 1:200 |
| CCR2 | K036C2 | 357212 | B260108 | PE/Cy7 | Biolegend | 1:80 |
| ICAM | HCD54 | 322718 | B193832 | AF 647 | Biolegend | 1:80 |
| CD45 | 2D1 | 368514 | B248834 | AF 700 | Biolegend | 1:20 |
| CD14 | M5E2 | 301820 | B274258 | APC/Cy7 | Biolegend | 1:20 |
| HLA-DR | L243 | 307624 | B278326 | Pacific Blue | Biolegend | 1:20 |
| CD66b | G10F5 | 305106 | B278603 | PE | Biolegend | 1:20 |
| SiglecF | E50-2440 | 552126 | 7058859 | PE | BD Biosciences | 1:200 |
| CD11b | M1/70 | 101256 | B238075 | PE Dazzle | BD Biosciences | 1:400 |
| CD11b | M1/70 | 101243 | B253527 | BV785 | Biolegend | 1:200 |
| MHCII | M5.114.15.2 | 107624 | B267551 | PerCP | Ebioscience | 1:200 |
| Epcam | G8.8 | 118230 | B251914 | APCCy7 Fire | Biolegend | 1:200 |
| CD3 | 17A2 | 100244 | B198733 | BIOTIN | Biolegend | 1:200 |
| CD3 | 17A2 | 100213 | B261416 | Pacific Blue | Biolegend | 1:200 |
| CD3 | 17A2 | 100229 | B282101 | BV 650 | Biolegend | 1:200 |
| CD3 | 17A2 | 100206 | B210714 | PE | Biolegend | 1:200 |
| CD19 | 6D5 | 115541 | B242632 | BV650 | Biolegend | 1:200 |
| CD19 | 6D5 | 115504 | B244881 | Biotin | Biolegend | 1:200 |
| CD19 | 6D5 | 115526 | B265435 | Pacific Blue | Biolegend | 1:200 |
| CD19 | 6D5 | 115508 | B223615 | PE | Biolegend | 1:200 |
| CD103 | 2E7 | 121433 | | BV605 | Biolegend | 1:400 |
| Ly6G | 1A8 | 127604 | B218526 | BIOTIN | Biolegend | 1:200 |
| Ly6G | 1A8 | 127608 | B221647 | PE | Biolegend | 1:200 |
| Ly6G | 1A8 | 127628 | B280589 | BV 421 | Biolegend | 1:200 |
| Ly6G | 1A8 | 135512 | B213676 | AF 488 | Biolegend | 1:200 |
| Lyve-1 | ALY7 | 50-044382 | 2205461 | eFluor 660 | Ebioscience | 1:200 |
| CD115 | AFS98 | 135510 | B211309 | APC | Biolegend | 1:200 |
| CD115 | AFS98 | 128006 | B217035 | FITC | Biolegend | 1:200 |
| Ly6C | HK1.4 | 128032 | B232012 | BV 421 | Biolegend | 1:200 |
| Pan-CD45 | 30-F11 | 103128 | B274307 | AF 700 | Biolegend | 1:200 |
| CD11c | N418 | 117318 | B222652 | PE/Cy7 | Biolegend | 1:200 |
| CD11c | N418 | 117352 | B218048 | APC/Fire750 | Biolegend | 1:200 |
| CD64 | X54-5/7.1 | 139304 | B191540 | PE | Biolegend | 1:200 |

| Antibody | Clone | Catalogue | Lot | Fluorophore | Source | dilution |
|---|---|---|---|---|---|---|
| CD64 | X54-5/7.1 | 139306 | B207411 | APC | Biolegend | 1:200 |
| CD4 | H129.19 | 553649 | | Biotin | BD Biosciences | 1:1600 |
| CD5 | 53-7.3 | 553019 | | Biotin | BD Biosciences | 1:800 |
| CD5 | 53-7.3 | 100603 | B254317 | Biotin | Biolegend | 1:200 |
| CD8a | 53 -6.7 | 553029 | | Biotin | BD Biosciences | 1:800 |
| CD11b | M1/70 | 101256 | B238075 | Biotin | BD Biosciences | 1:200 |
| CD45R/B220 | RA3-6B2 | 553086 | | Biotin | BD Biosciences | 1:200 |
| Ter119 | TER-119 | 116204 | B295203 | Biotin | Biolegend | 1:200 |
| Ter119 | TER-119 | 553672 | | Biotin | BD Biosciences | 1:50 |
| Gr-1/Ly-6G/C | RB6-8C5 | 553125 | | Biotin | BD Biosciences | 1:100 |
| CD117/cKit | 2B8 | 105811 | B249345 | APC | Biolegend | 1:200 |
| Sca-1/Ly- | E13-161.7 | 122506 | | FITC | Biolegend | 1:200 |
| CD48 | HM48-1 | 103406 | | PE | Biolegend | 1:500 |
| CD150 | 12F12.2 | 115914 | | PECy7 | Biolegend | 1:200 |
| CD71 | RI7217 | 113807 | | PE | Biolegend | 1:500 |
| FcBlock CD16/32 | 93 | 101320 | B295040 | | Biolegend | 1:100 |
| Streptavidin | - | 405232 | B251688 | BV 650 | Biolegend | 1:1000 |
| Streptavidin | - | | | Pacific Blue | BD biosciences | |
| LIVE/DEAD® Fixable Aqua | - | L34957 | 2068285 | UV650 | Life Technologies or Biolegend | 1:501:100 |
| CD45.2 | 104 | 109822 | B252126 | AF700 | Biolegend | 1:200 |
| CD45.1 | A20 | 110741 | B253101 | BV510 | Biolegend | 1:200 |
| Sca.1 | D7 | 108129 | B262926 | BV510 | Biolegend | 1:200 |
| CD150 | TC15- 12F12.2 | 115903 | | PE | Biolegend | 1:200 |
| CD105 | MJ7/18 | 120412 | B245562 | PacBlue | Biolegend | 1:200 |
| CD41 | MWReg30 | 133927 | B268849 | APCCy7 | Biolegend | 1:200 |
| IFNAR | MAR1-5A3 | 127325 | B286788 | PECy7 | Biolegend | 1:200 |
| CD5 | 53-7.3 | 100603 | B254317 | Biotin | Biolegend | 1:200 |
| Ly6G | 1A8 | 127604 | B218529 | Biotin | Biolegend | 1:200 |
| B220 | RA3-6B2 | 103204 | B288658 | Biotin | Biolegend | 1:200 |
| CD11b | M1/70 | 562287 | | CF594 | Biolegend | 1:200 |
| F480 | CI:A3-1 | ab6640 | | Purified | Abcam | 1:100 |
| IL-10 | JES5-2A5 | ab189392 | | Purified | Abcam | 1:100 |
| LYVE-1 | Polyclonal | 103-PA50AG | | Purified | ReliaTech GmbH | 1:200 |
| HIF-1 alpha | Polyclonal | NB100-479 | | Purified | Novus Biotech | 1:00 |

*Nat Immunol*. Author manuscript; available in PMC 2022 June 10.

| Antibody | Clone | Catalogue | Lot | Fluorophore | Source | dilution |
|----------|-------|-----------|-----|-------------|--------|----------|
| CD11b | M1/70 | 101243 | B287244 | BV 785 | Biolegend | |

