## [Peer Review File · Nature immunology]

Peer Review Information

Journal: Nature Immunology

Manuscript Title: Hypoxia shapes the immune landscape in lung injury promoting inflammation persistence

Corresponding author name(s): Ananda Mirchandani

Reviewer Comments & Decisions:

Decision Letter, initial version:
--

Subject: Decision on Nature Immunology submission NI-A31918

Message: 18th May 2021

Dear Dr Mirchandani,

Thank you for your response to the referees comments on your article "Hypoxia shapes the immune landscape in lung injury promoting inflammation persistence". While we find your work of interest, the reviewers have raised substantial concerns that must be addressed. As such, we cannot accept the current version of the manuscript for publication, but would be happy to consider a revised version that addresses these concerns, as long as novelty is not compromised in the interim.

Please revise along the lines specified in your letter. At resubmission, please include a "Response to referees" detailing, point-by-point, how you addressed each referee comment. You can also highlight all changes in the manuscript text file. If no action was taken to address a point, you must provide a compelling argument. This response will be sent back to the referees along with the revised manuscript.

Please include a revised version of any required reporting checklist. It will be available to referees to aid in their evaluation. The Reporting Summary can be found here: <https://www.nature.com/authors/policies/ReportingSummary.pdf>

The Reporting Summary can be found here:
<https://www.nature.com/documents/nr-reporting-summary.pdf>

When submitting the revised version of your manuscript, please pay close attention to our [href="https://www.nature.com/nature-research/editorial-policies/image-integrity"](https://www.nature.com/nature-research/editorial-policies/image-integrity)>Digital Image Integrity Guidelines. and to the following points below:

You may use the link below to submit your revised manuscript and related files:
[REDACTED]

We hope to receive it within 6 months. If you cannot send it within this time, please let us know. We will be happy to consider your revision so long as nothing similar has been accepted for publication at Nature Immunology or published elsewhere.

Nature Immunology is committed to improving transparency in authorship. As part of our efforts in this direction, we are now requesting that all authors identified as 'corresponding author' on published papers create and link their Open Researcher and Contributor Identifier (ORCID) with their account on the Manuscript Tracking System (MTS), prior to acceptance. ORCID helps the scientific community achieve unambiguous attribution of all scholarly contributions. You can create and link your ORCID from the home page of the MTS by clicking on 'Modify my Springer Nature account'. For more information please visit www.springernature.com/orcid.

Thank you for the opportunity to review your work.

Sincerely,

Ioana Visan, Ph.D.
Senior Editor
Nature Immunology

Tel: 212-726-9207
Fax: 212-696-9752
www.nature.com/ni

Reviewers' Comments:

Reviewer #1:

Remarks to the Author:

The manuscript by Mirchandani et al. (Hypoxia shapes the immune landscape in lung injury promoting inflammation persistence) investigates the impact of hypoxia in Acute Respiratory Distress Syndrome (ARDS). The work shows that acute lung injury developed in mice challenged with hypoxia and streptococcus pneumoniae display monocytopenia and failure to expand interstitial macrophage population in the lung, which was paralleled by increased erythropoiesis. Such conditions mimicked the monocytopenia observed in ARDS patients and are associated with unchecked neutrophilic inflammation. The authors propose that the hypoxia-mediated inhibition of lung interstitial macrophages (IM) is, at least partially, consequent to inhibition of type I IFN, a known monocytopenic factor. The authors further observe that administration of CSF-1 rescues the monocytopenia in hypoxic endotoxin-driven acute lung injury model (ALI), enabling lung interstitial macrophages (IM) expansion and acquisition of anti-inflammatory phenotype, through their induction of IL-10.

Comments:

- Even is well expected, the supposed key role of neutrophils in unchecked lung inflammation is not investigated functionally in the mouse models used in the work.
- Fig. 1 shows differential regulation of monocytes subsets by hypoxia. This initial analytic stratification of monocytes subsets is not properly followed, later in the manuscript
- Fig. 1 shows the heatmap of differentially expressed genes measured in blood monocytes, obtained from healthy control and ARDS patients. Accordingly to the work's hypothesis, the authors show that ARDS is associated with increased expression of hypoxia-regulated genes (SLC2A3, etc.). However, ARDS monocytes also display drastic reduction of the prototypic hypoxia-inducible gene VEGFA (Fig. 1j). How to explain this? It would be necessary to measure the expression levels of HIF-1a, as readout of hypoxic conditions (antibodies for FACS analysis are available)
- The proteomic analysis of circulating monocytes from ARDS patients, Fig. 1l, shows increased levels of Proteinase 3, myeloperoxidase and azurocidin, which are known markers of neutrophils, more than monocytes! This is quite surprising.
- There is no mention/evaluation of lymphocytic populations in the manuscript, despite these cells are relevant players in ARDS.
- Fig. 4. The authors show skewing towards increase erythropoiesis in mice exposed or hypoxia, as compared to normoxia. Fig. 4l shows, in particular, elevation of Pre-CFU-E colonies without increase of their precursors Pre-MegaE (according to the scheme of Fig. 4i). Is this dissociation reasonable or something is missing? Does the elevation of erythrocytes correlateds with increased levels of their growth factors (e.g. thrombopoietin, IL-11, erythropoietin; this last is a known hypoxia inducible product)? Can the latter be estimated?
- Fig. 5 shows that bone marrow cells cultured in hypoxia (1% oxygen) do not upregulate IRF8, IRF1 and CCR5 in response to type I IFN. Moreover, analysis of IFNRA expression by LSK cells in vivo showed decreased levels upon mice treatment with LPS and in the contest of hypoxia. A question is whether the authors can estimate the level of hypoxia in the bone marrow of mice and to evaluate whether this is equivalent or compatible with their in vitro conditions. Further, the authors could estimate HIF_1alpha levels by FACS or, alternatively, evaluate hypoxia levels in the bone marrow tissues, by using Pimonidazole. Moreover, since IRF8 is activated by IFN γ , which is also potent inducer of monocytopenia, it would be interesting to measure the effect of hypoxia on IFN type I and II in vivo, in lung mono/macrophages.
- To ascertain the longer-term effects of systemic hypoxia on the myeloid compartment for

inflammation resolution, the authors studied the lung 5 days after LPS challenge (Fig. 6). Mice displayed very few lung neutrophils in normoxia, which significantly increased in hypoxia. Such phenomenon was paralleled by decreased lung macrophages. The increased number of neutrophils in hypoxic mice correlated with increased levels of the chemokine CXCL1 (Fig. 6j). However, in the same condition (hypoxia+LPS), although to in shorter time point (24 h), blood neutrophils were basically abolished (Fig. 4b). The authors should show the levels of blood neutrophils at 5 days, since it is not clear the origin of lung neutrophils at longer time points (from where lung neutrophils come from?).

-In addition, in hypoxia-LPS condition (Fig. 2a) circulating monocytes express higher levels of CCR2. Thus, why increased number of circulating monocytes expressing high CCR2 do not support their recruitment to lungs? Are peripheral monocytes from hypoxia-LPS mice unresponsive to CCL2? Is CCL2 not expressed in the lungs?

-In Fig. 7, the authors treated the LPS-challenged mice with exogenous CSF1, in the form of a CSF1-Fc fusion protein and, as expected, observed increased number of circulating monocytes (only few neutrophils). This was in agreement with increased number of lung IM (Fig. 7e) and with reduction of lung neutrophils. Does CSF1 reduce CXCL1 expression in lungs? The question is related to the fact that Fig. 7b shows increased circulating neutrophils in M-CSF treated mice, which however do not reach the lungs.

-Moreover, considering the striking effects of CSF1 on monocytes mobilization (increased circulating monocytes) and IM accumulation, it would be important to assess as to whether hypoxia reduces M-CSF levels, as well as CSF1R expression by mono/macrophages.

-Fig. 7a-I, the experiments were performed in normoxia and hypoxia. No LPS treatment. Instead, Fig. 7k shows differential gene expression in classical blood monocytes from LPS-treated mice (see legend Fig. 7k). Along the manuscript, remains somehow difficult to discern the effects of LPS from the effects of hypoxia.

-The evaluation of erythropoietic factors (i.e. erythropoietin) and MCSF levels in hypoxia is relevant, since it was shown that inhibition of erythropoiesis by TNF is paralleled by overexpression of PU.1, a key transcription factor driving monocytopoiesis (Grigorakaki C, Morceau F, Chateauvieux S, Dicato M, Diederich M. *Biochem Pharmacol.* 2011

Reviewer #2:

Remarks to the Author:

In this manuscript, the authors describe the impact of hypoxia on lung injury using peripheral blood samples from ARDS patients and also a mouse model that they propose resembles the disease condition in ARDS patients. There are limited treatment options for ARDS patients other than disease management by supportive care. Thus, the study has clinical significance. The rationale for the mouse model they adopted, however, is unclear. Certainly, the patients are hypoxic at the point they are put on a ventilator. However, once intubated, they are under a hyperoxic condition and in fact hyperoxia induces lung injury. Hyperoxic lung injury is a model for ventilator-induced lung injury. It seems the authors are interested in the effect of hypoxia on immune cells, which would be the condition that precedes intubation. Yet, as mentioned in the 2nd line under results, they collected samples after patients were intubated and at 2 time points—at < 48 h after ventilation which they call early ARDS and at > 48 h-7 days, which they refer to as late ARDS. The patients were clearly not hypoxic at any of these time points. There is also insufficient information about the patients from whom samples were obtained.

Specific comments:

1. More granular detail in pulmonary vs extrapulmonary causes of ARDS would be relevant. For extrapulmonary ARDS, are the etiologies all infection/sepsis-related or are other etiologies involved? What were the pathogens for pulmonary causes of ARDS in terms of viral vs bacterial vs unknown vs non-infectious? In Fig 1, how were the 5 ARDS patients selected for monocyte DEG expression. Information should be provided about which patients were selected including their clinical characteristics from Table S1.
2. The human data provided do not support that the lung microenvironment found in the mouse experimental data is representative of human ARDS. Why did the authors not sample peripheral blood before each patient was put on a ventilator to study the actual hypoxic condition? The first figure does not have any human data on the lung environment in ARDS. One cannot assume that interstitial macrophages are also low in the human lung during ARDS. It is possible that the authors do not have an approved protocol for lavaging the lungs of ARDS patients although BAL fluid should not contain many interstitial macrophages (IMs). Monocytopenia in the periphery does not allow one to draw conclusions about the lung immune environment.
3. Mechanistically it appears that in the mouse hypoxia caused bone marrow suppression/altered myeloid differentiation and function in the face of LPS-induced ALI. Is this specific to lung injury or is this altered myelopoiesis due to hypoxia in response to any type of systemic inflammation? It is written as if it is specific to ALI/ARDS, but there is no proof of that comparing the results to systemic LPS administration. While one could rationalize that severe hypoxia elicited by LPS+hypoxia induces the peripheral aberrant immune response, the model is not designed to mimic the condition of a patient on a ventilator when they are not hypoxic any longer. It will be interesting to compare hypoxic mice to those under hyperoxia. This would be more representative of a hypoxic patient with lung injury being rescued with high FiO₂ on a ventilator.
4. The clinical data comparing ARDS to healthy controls is making a comparison between a patient with a critical illness syndrome and a healthy patient. The data would be more compelling if ARDS patients were compared to other critically ill patients without ARDS (and by extension, other critically ill patients without hypoxia).
5. The authors have attempted to explain that their findings of monocytopenia in ARDS is at odds with what multiple studies have described in COVID patients overwhelmingly associating monocytes with more severe disease. I was not able to comprehend their explanation. In non-COVID ARDS as well, a gene signature associated with monocytes identifies a risk score (Jiang et al., JCI insight, 2020).

Minor points:

Many relevant publications have not been cited. IL-10-expressing regulatory interstitial myeloid cells that help in resolution of lung inflammation in models of pneumonia and lung injury are described in the literature. They should cite previous clinical applications of colony stimulating factors in ARDS in their discussion if they are proposing it as a potential therapeutic. IV GM-CSF was trialed in ARDS and failed (<https://www.ncbi.nlm.nih.gov/pmc/articles/PMC3242850/>). Inhaled GM-CSF was tried in a case series (<https://www.atsjournals.org/doi/full/10.1164/rccm.201311-2041LE>).

Reviewer #3:

Remarks to the Author:

In their manuscript "Hypoxia shapes the Immune Landscape in Lung Injury Promoting Inflammation Persistence," Mirchandani and colleagues seek to understand how hypoxia impacts the resolution of inflammation in the injured lung. The investigators begin by showing that fewer monocytes are present in the circulation of patients with ARDS and that monocytes are reduced in the circulation of mice exposed to inflammatory stimuli and low ambient oxygen. Elegant studies of the bone marrow demonstrate lower levels of monocyte precursors in mice subjected to hypoxic conditions and suggest that impaired type 1 interferon signaling is a mechanism. These are followed by experiments in which administration of CSF1-Fc increases lung monocyte numbers while attenuating lung neutrophil accumulation and capillary leak.

The paper focuses on the resolution of lung inflammation in the context of ARDS, and therefore touches on an important area. The paper is well written and the experiments are logically presented. Appropriate controls are used and statistical analysis is sound. Investigation into type 1 interferon signaling and the role of CSF1 in the resolution are novel and represent strengths of the paper. However, there are several limitations to the manuscript.

The main limitations include: a) conceptual issues with hypoxia versus hypoxemia, b) potential issues with identification of IMs and lack of experiments to show their transcriptional profiles, c) incomplete establishment of links between interferon signaling, hypoxia and CSF1. A detailed description of each of these is described below.

Major comments:

1. Hypoxia is a main element of the paper. However, there are issues with how the term is used, whether it applies to the human cohort, and whether it is present in the animal models. These issues are described in detail below. Importantly, the human studies are quite nice, and the animal model is elegant. Careful rewriting of the paper should be able to address the issues.

a. A main concept on which the paper is based is systemic hypoxia. The way this is portrayed in the manuscript tends to cause confusion, since low PaO₂ levels are provided as evidence of hypoxia in the authors' cohort. Importantly, PaO₂ is a measure of hypoxemia (rather than tissue hypoxia). To this point, many patients with ARDS have adequate PaO₂ levels as a result of supplemental oxygen administration (generally with mechanical ventilation). Moreover, when arterial hypoxemia is present it is generally transient. Conversely, hypoxia is common in patients with shock as a result of poor tissue perfusion. Lactate levels would be a more adequate measure of tissue hypoxia.

b. The rationale provided for experiments in Figure 1 is "to determine whether hypoxia is associated with alterations in the blood monocyte compartment." However, the experimental design doesn't adequately enable testing of the association. First, subjects with ARDS will have systemic inflammation that will potently alter monocyte transcriptomes compared to controls. Second, while patients with ARDS can have hypoxia, in most cases it is corrected with supplemental oxygen. Third, while a few of the genes presented in the figure are associated with hypoxia, many quintessential monocyte hypoxic genes (i.e. VEGF) are not demonstrated (PMID: 16849508). The experiment itself is fine – the rationale for the experiment and a link to hypoxia is the problem.

c. Analysis of monocytes from animal experiments show that genes are differentially

expressed. However, it is unclear if hypoxia is the main mechanism. The authors have previously explored HIF-responsive genes (including key metabolic genes). Assessment of these would help support the link between hypoxia and monocyte profiles.

2. Identification of interstitial macrophages in the lungs may be flawed based on the scheme shown in Fig 3.

a. The Ly6C hi cells that are reported to be IMs have the characteristics of monocyte-derived macrophages, many of which will be in the airspaces. Examination of the BAL would likely show these populations. Side-scatter might help separate the IMs (PMID: 23672262). However, use of intratracheal antibodies would be more definitive (PMID: 28850249).

b. The tsne plots shown in figure 7m may help identify IMs. A panel showing non-inflamed mice is essential to provide baseline data. In addition, regions showing resident AMs, monocyte-derived recruited AMs, resident tissue IMs and monocyte derived IMs should be indicated. In the supplement, dot plots with appropriate FMO controls are needed.

c. The authors suggest that Lyve-1+ MHCII negative IMs expand in response to CSF1. These cells should be indicated in the tsne plots in Fig 7m. In addition, flow plots with gating strategies should be shown to confirm this finding and data from Fig 7n. Side scatter should be included to help distinguish IM and AMs.

d. In Figure 7p many of the F4/80 positive cells appear to be alveolar macrophages rather than IMs. Moreover, AMs are notably absent from the H PBS image. The authors should confirm that exposure times are similar between images. Co-staining with Lyve-1 would help confirm the data from Fig 7n.

3. The link between type 1 interferon signaling and hypoxia are incompletely studied. The investigators show that expression of interferon responsive genes is attenuated in BMDMs cultured in 1% oxygen, and that INFAR expression is reduced in LSK cells isolated from LPS treated mice exposed to 10% FiO2. However, attempts to link this to reduced monocyte and macrophage function are missing.

a. Experiments that demonstrate reduced interferon signatures in monocytes from humans and/or mice would provide additional reassurance as would measurement of expression profiles of IMs.

b. Do the investigators think that type 1 interferons are reduced in states of systemic hypoxemia, or are the downstream IFN signaling mechanisms defective? This could be addressed in the discussion

c. If the main detrimental effect of IFNAR deletion is to prevent monocyte production by the bone marrow can this be overcome by administration of CSF1-Fc?

4. Administration of CSF1-Fc impacts monocyte and macrophage numbers. However, it is unclear what roles these expanded cells play.

a. Do the expanded IMs play a role in dampening inflammation?

b. Does the CSF-1 affect AM function to dampen inflammatory responses? In other words, is there a salutary effect of CSF-1 that is independent of monocyte recruitment?

Minor

Table 1 should be expanded. Measures of tissue hypoxia (such as lactate levels) would be helpful, as mentioned above. In addition, whether or not glucocorticoids were administered should be listed. These are now commonly used in ARDS and have been shown to cause monocyte apoptosis.

Subjects with ARDS have reduced levels of classical monocytes. In comparison, classical monocytes are increased in hypoxemic mice exposed to inflammatory stimuli. Since classical monocytes are the precursors to recruited macrophages this seems relevant. How do the authors interpret this finding in the context of their model?

The text describes Figure 6E as showing interstitial neutrophils. Intravascular neutrophils may be included in this assessment. Describing the output as total lung neutrophils (as in the figure) may be more accurate.

Raw data for chimerism experiments should be shown. It will be instructive to see how effective engraftment and shielding are for this set of experiments.

Author Rebuttal to Initial comments

Dear Ioana,

Many thanks for the opportunity to provide a revised manuscript in which we address the issues raised by the reviewers. We note that all reviewers found merit in the experimental approach undertaken and its clinical significance, but raised concerns regarding the concepts of hypoxia and hypoxaemia, how we defined lung monocyte-macrophage populations, and the links between interferon (IFN) signalling / hypoxia and colony stimulating factor (CSF)1. We provide a detailed point by point response to each of the points raised by the reviewers below.

Specifically, we now provide a detailed characterisation of the oxygenation status of the patients sampled, demonstrating evidence of a clinically important period of hypoxaemia, despite supportive oxygen therapy and positive pressure ventilation. We propose that this exposure has long lasting effects within the bone marrow, blood and tissue compartments, and in response to reviewer comments have re-written the manuscript accordingly. Importantly, we provide new data evidencing stabilisation of HIF1 α protein within the lung tissue and bone marrow compartment (mice) in response to inflammation and hypoxemia, with consequent tissue hypoxia, and a hypoxic protein signature in the circulating classical monocytes isolated from the ARDS cohort. Further evidence of activation of physiological responses to systemic hypoxia is also provided in the murine model with quantification of circulating levels of erythropoietin and IL-11. Additional clinical granularity has also been provided pertaining to the aetiology of extra-pulmonary ARDS, and pathogens associated with pulmonary ARDS where the aetiology of the ARDS was infection. We have re-named all lung non-AM CD64⁺SiglecF macrophages “tissue macrophages (tM ϕ)” whilst renaming the Ly6C⁺IM “monocyte-derived macrophages (MDM)” as helpfully suggested by reviewer 3. The remaining CD64⁺SiglecF⁻Ly6C⁻ lung macrophages as “Ly6C⁻ tM ϕ ”. New experimental work has also been undertaken to further define bronchoalveolar lavage-recovered

inflammatory macrophage populations by flow cytometric analysis of side scatter profiles and surface antibody expression, with a proteomic survey of circulating monocyte populations undertaken in parallel. Effects of systemic inflammation using intraperitoneal LPS in mice exposed to 10% FiO₂ have been explored, with a comparable failure to expand intraperitoneal inflammatory macrophage populations in the context of systemic hypoxia. Direct evidence that re-oxygenation is unable to fully rescue the lung tM ϕ expansion following an initial hypoxic insult is now provided, both validating the direct regulation of the immune landscape by systemic hypoxia and providing evidence of long-lasting effects, even when the initial hypoxic challenge is reversed. Finally, links between IFN signalling/hypoxia and CSF1 have been addressed experimentally with transcription profiling and quantification of IFN surface receptor expression of circulating monocytes and IM populations in LPS treated mice exposed to 10% FiO₂ and by the administration of CSF1 to the *Ifnar1*^{-/-} mice. This extensive body of new experimental work, together with the additional clinical granularity and changes to the manuscript text, we believe addresses the concerns raised by each of the reviewers. Thus, we now provide further direct evidence that an initial hypoxic insult is sufficient to drive a sustained change in the immune landscape of the lung during acute lung injury with consequence for resolution of lung inflammation. Furthermore, aberrant pathophysiological pathways induced by hypoxia can be manipulated to improve outcomes. We would be most grateful for you taking all of this into account when considering our manuscript for Nature Immunology.

Reviewer #1

(Remarks to the Author)

The manuscript by Mirchandani et al. (Hypoxia shapes the immune landscape in lung injury promoting inflammation persistence) investigates the impact of hypoxia in Acute Respiratory Distress Syndrome (ARDS). The work shows that acute lung injury developed in mice challenged with hypoxia and streptococcus pneumoniae display monocytopenia and failure to expand interstitial macrophage population in the lung, which was paralleled by increased erythropoiesis. Such conditions mimicked the monocytopenia

observed in ARDS patients and are associated with unchecked neutrophilic inflammation. The authors propose that the hypoxia-mediated inhibition of lung interstitial macrophages (IM) is, at least partially, consequent to inhibition of type I IFN, a known monocytopoietic factor. The authors further observe that administration of CSF-1 rescues the monocytopenia in hypoxic endotoxindriven acute lung injury model (ALI), enabling lung interstitial macrophages (IM) expansion and acquisition of anti-inflammatory phenotype, through their induction of IL-10.

Comments:

-Even is well expected, the supposed key role of neutrophils in unchecked lung inflammation is not investigated functionally in the mouse models used in the work.

We thank the reviewer for their interest in our work. We apologise for failing to reference existing work in this area. Infiltrating neutrophils in the setting of ARDS have previously been reported to increase vascular injury and protein leak (Flick et al. *Circ Res* 1981) and promote alveolar epithelial injury (ARDS network et al., *N Engl J Med* 2000), thus driving a damaging inflammatory response. More recently, we have published work delineating the importance of unchecked neutrophilic inflammation in mice challenged with *Strep. pneumoniae* exposed to 10% FiO₂, using anti-Ly6G neutrophil depletion to explore the neutrophil dependence of the systemic response observed in acute hypoxia (Thompson et al., *Science Immunol* 2017). Moreover, we have observed that systemic hypoxia drives neutrophil protein scavenging, maintaining central carbon metabolism and enabling neutrophils to sustain synthetic and effector function in the tissues (Watts et al., *J Clin Invest* 2021). We have modified the text to include a clearer reference to the role of neutrophils in unchecked lung inflammation in the setting of systemic hypoxia, please page 13, line 24..

- Fig. 1 shows differential regulation of monocytes subsets by hypoxia. This initial analytic stratification of monocytes subsets is not properly followed, later in the manuscript

Human monocyte populations were sub-classified into classical, intermediate or non-classical cells based on their expression of CD14 and CD16 as previously reported (Patel et al. J Exp Med 2017). In mice, classical and non-classical monocytes are best defined by their differential expression of Ly6C (Ingersoll Blood 2010). We have clarified this approach in the manuscript text, please see page 7, line 20

-Fig. 1 shows the heatmap of differentially expressed genes measured in blood monocytes, obtained from healthy control and ARDS patients. Accordingly to the work's hypothesis, the authors show that ARDS is associated with increased expression of hypoxia-regulated genes (SLC2A3, etc..). However, ARDS monocytes also display drastic reduction of the prototypic hypoxia-inducible gene VEGFA (Fig. 1j). How to explain this? It would be necessary to measure the expression levels of HIF-1 α , as readout of hypoxic conditions (antibodies for FACS analysis are available)

We thank the reviewer for raising this important point. We now provide new data detailing the degree of hypoxaemia during the 24 hour period prior to sampling, which persists despite both oxygen therapy and positive airway ventilation (please see rebuttal Figure 1a, b, manuscript Figure 1a, d). To address the protein response of circulating classical monocyte populations to systemic hypoxia, we have now undertaken a proteomic survey of CD14⁺⁺ CD16⁻ cells (please see rebuttal Figure 1c, manuscript Figure 1m), comparing these data to the transcriptional data sets published for human blood monocytes exposed to hypoxic culture in vitro as helpfully suggested by reviewer 3 (Bosco et al. JI, 2006). Of note, HIF-1 α stabilisation is not sustained throughout extended periods of hypoxic cell culture (Bosco et al. JI), underpinning the dynamic nature of hypoxic-responses even in vitro, with a failure to detect HIF1 α by LC-MS also observed. Consequently, we have provided physiological measure of systemic hypoxaemia, rather than measures of HIF stabilisation in the patient cohort.

a

b

c

Figure 1 Patients with ARDS remain hypoxemic despite high supplementary oxygen delivery

(a) Retrospective analysis of the partial pressure of oxygen (PaO_2) from clinical arterial blood gas samples taken from individual ARDS patients in the 24 hours preceding research blood sampling and plotted relative to the normal arterial oxygen range (green shaded area). Each point represents an independent clinical sample measurement. (b) The highest recorded fractional inspired oxygen (FiO_2) provided to the ARDS patients up to 24 hours prior to sampling. (c) Proteomic analysis of classical monocytes ($\text{CD14}^{++}\text{CD16}^{-}$) was performed by LCMS using the monocyte hypoxic signature identified by Bosco et al (Bosco et al. JI), following human in vitro culture of monocytes in hypoxia.

-The proteomic analysis of circulating monocytes from ARDS patients, Fig. 1I, shows increased levels of Proteinase 3, myeloperoxidase and azurocidin, which are known markers of neutrophils, more than monocytes! This is quite surprising.

We agree with the reviewer that this is somewhat unexpected at first. However, there is existing literature demonstrating the expression of granule proteins in monocyte sub-populations in health and following exposure to microbial derivatives (Yanez et al. Immunity 2017). In support of our human data sets, we have also now undertaken confocal imaging of blood Ly6C^{hi} monocytes validating MPO expression. (please see rebuttal Figure 2a). Proteomic survey of the Ly6C^{hi} monocytes from mice challenged with nebulised LPS further confirms the presence of granule proteins, with suppression of granule protein expression in response to CSF1 treatment.

(rebuttal Figure 2b, manuscript Figure 7m).

a

b

Figure 2: Dynamic regulation of circulating Ly6C^{hi} monocyte granule protease expression by CSF1 (a) Ly6C^{hi} mouse monocytes from mice treated with LPS and placed in hypoxia (10% FiO₂) for 24 hours were fixed and sorted by FACS, stained for myeloperoxidase (green) with a dapi cell mask for nuclear staining and imaged by confocal microscopy. (b) The proteome of Ly6C^{hi} monocytes nebulised with LPS, housed in hypoxia (10% FiO₂) and treated with (grey points) or without CSF1 were compared. Granule-associated proteins (blue points) were identified in the volcano plot with abundance relative to the PBS-treated mice plotted.

-There is no mention/evaluation of lymphocytic populations in the manuscript, despite these cells are relevant players in ARDS.

We apologise for omitting these data from the original version of the manuscript. While we were making an effort to keep the data focussed on monocytes/macrophages, we absolutely agree that lymphocytes play an important role in disease pathogenesis. We have now included data on lung B and T cell populations in response to acute challenge (please see rebuttal Figure 3a, b, manuscript Figure S3a, b).

Figure 3: Systemic hypoxia does not affect lung B cell or T cell numbers 24 hours post LPS

T cells **(a)** (Live, singles, CD45⁺, Lin⁺ (CD3/CD19/Ly6G) MHCII⁻ CD11b⁻) and B cells **(b)** (Live, singles, CD45⁺, Lin⁺ (CD3/CD19/Ly6G) MHCII⁺ CD11b⁻) were quantified in lung digests from mice housed in normoxia (N) or 10% FiO₂ hypoxia (H) for 24 hours, left naïve or nebulised with LPS.

-Fig. 4. The authors show skewing towards increase erythropoiesis in mice exposed or hypoxia, as compared to normoxia. Fig.4l shows, in particular, elevation of Pre-CFU-E colonies without increase of their precursors Pre-MegE (according to the scheme of Fig.4i). Is this dissociation reasonable or something is missing? Does the elevation of erythrocytes correlated with increased levels of their growth factors (e.g. thrombopoietin, IL-11, erythropoietin; this last is a known hypoxia inducible product)? Can the latter be estimated?

Thank you for raising this interesting point. We do see an increase of Pre-Meg E in LPS treated mice exposed to 10% FiO₂ (please see manuscript Figure 4k). To address whether the elevation of erythrocytes correlates with increased levels of their growth factors, we have now measured systemic (circulating) levels of erythropoietin, thrombopoietin, and IL-

11, and observe a significant increase in both EPO (24 hours and day 5) and IL-11 (day 5) in the setting of hypoxia (please see rebuttal Figure 4a-d), manuscript Figure 5a, b and Figure 5a).

Figure 4: Serum erythropoietin and IL-11 levels are increased in mice housed in hypoxia

Serum erythropoietin (EPO) (a, b), IL-11(c) and thrombopoietin (d) from mice treated with nebulised LPS and housed in normoxia (N) or hypoxia (H), for 24 hours or 5 days, were measured by ELISA as per manufacturers' instructions.

-Fig. 5 shows that bone marrow cells cultured in hypoxia (1% oxygen) do not upregulate IRF8, IRF1 and CCR5 in response to type I IFN. Moreover, analysis of IFNRA expression by LSK cells in vivo showed decreased levels upon mice treatment with LPS and in the context of hypoxia. A question is whether the

authors can estimate the level of hypoxia in the bone marrow of mice and to evaluate whether this is equivalent or compatible with their in vitro conditions. Further, the authors could estimate HIF_1alpha elevels by FACS or, alternatively, evaluate hypoxia levels in the bone marrow tissues, by using Pimonidazole. Moreover, since IRF8 is activated by IFNg, which is also potent inducer of monocytopoiesis, it would be interesting to measure the effect of hypoxia on IFN type I and II in vivo, in lung mono/macrophages.

This is an important point. We have now stained the bone marrow of mice exposed to nebulised LPS under conditions of both normoxia and hypoxia, and observe an increase in HIF-1 α expression in the setting of hypoxia (please see rebuttal Figure 5a, manuscript Figure 5n). We also, observe a decrease in circulating levels of the type I IFN (IFN α), (rebuttal Figure 5b, c e, f, manuscript Figure 5c and Figure S5b), thus suggesting hypoxic regulation of type I IFN signalling at both local and systemic levels. Within the lung compartment, we do not observe changes in BAL levels of type 1 IFNs (a and b) at either early (24 hours) or late (5 days) timepoints. With respect to the type II IFN (IFN γ), we detect the localised suppression of IFN γ release into the airways at early time points, which reaches equivalence by day 5, and is not associated with changes in circulating levels (rebuttal Figure 5d, g and manuscript Figure S5c).

Figure 5: HIF1 α expression is enhanced in the bone marrow of systemically hypoxic LPS-treated mice and serum interferon alpha is obliterated in mice housed in hypoxia post-LPS

(a) Mice were treated with LPS and placed in normoxia (N LPS) and hypoxia (H LPS) for 24 hours. Mice were sacrificed, femurs collected, fixed and demineralized for 2 weeks. The bone marrows were subsequently sectioned and stained for HIF1 (red) and dapi (blue). The image acquisition and processing steps were performed using the same settings for both sample groups. Serum was collected from mice treated with LPS and placed in normoxia (N LPS) or hypoxia 10% (H LPS) for 24 hours (b-d)) or 5 days (e-g). Type I interferons (IFN α and IFN β) and type II interferon (IFN gamma) were measured by MSD V-plex assay as per manufacturer's instructions.

-To ascertain the longer-term effects of systemic hypoxia on the myeloid compartment for inflammation resolution, the authors studied the lung 5 days after LPS challenge (Fig. 6). Mice displayed very few lung neutrophils in normoxia, which significantly increased in hypoxia. Such phenomenon was paralleled by decreased lung macrophages. The increased number of neutrophils in hypoxic mice correlated with increased levels of the chemokine CXCL1 (Fig. 6j). However, in the same condition (hypoxia+LPS), although to in shorter time point (24 h), blood neutrophils were basically abolished (Fig. 4b). The authors should show the levels of blood neutrophils at 5 days, since is not

clear the origin of lung neutrophils at longer time points (from where lung neutrophils comes from?).

We agree that at 24 hours there is absence of a blood neutrophil BrdU⁺ circulating population (Figure S4b), which would fit with the reported short circulating half-life of a neutrophil with rapid neutrophil recruitment to the lung following LPS challenge. We propose that the persistence of neutrophils at day 5 in hypoxia is in part explained by enhanced neutrophil survival in the setting of hypoxia (Walmsley et al J Exp Med 2005, Walmsley et al., JCI 2011). As requested, we also now include data on levels of blood neutrophils at day 5 (please see rebuttal figure 6, manuscript Figure S2b).

Figure 6: Blood neutrophils are equivalent in mice housed in hypoxia versus mice housed in normoxia 5 days post-LPS

Blood was collected from mice treated with LPS and placed in normoxia (N LPS) or hypoxia 10% (H LPS) for 5

days and circulating neutrophils quantified by flow cytometry (Live Singles CD45⁺Ly6G⁺CD11b⁺).

-In addition, in hypoxia-LPS condition (Fig. 2a) circulating monocytes express higher levels of CCR2. Thus, why increased number of circulating monocytes expressing high CCR2 do not support their recruitment to lungs? Are peripheral monocytes from hypoxia-LPS mice unresponsive to CCL2? Is CCL2 not expressed in the lungs?

The reviewer raises an important point here, specifically that CCL2 levels may be reduced in the context of hypoxia. To address this point directly, we have now measured CCL2 (MCP-1) levels in BAL fluid (please see rebuttal Figure 7, manuscript Figure 3k). This shows that, if anything, CCL2 is elevated by hypoxia. Thus, it is highly unlikely that the reduction in monocytes recruited to the lung is a consequence of impaired CCL2/CCR2 signalling. Instead, we propose that failure to recruit monocytes reflects the fact that the absolute number of circulating monocytes is markedly reduced, as a result of effects on hematopoiesis, leaving very few monocytes to be recruited.

Figure 7: Bronchoalveolar lavage (BAL) CCL2 is elevated in mice housed for 24 hours in hypoxia postLPS

BAL was collected from mice treated with LPS and placed in normoxia (N LPS) or hypoxia 10% (H LPS) for 24 hrs. CCL2 levels were measured by MSD V-plex assay as per manufacturer's instructions.

-In Fig. 7, the authors treated the LPS-challenged mice with exogenous CSF1, in the form of a CSF1-Fc fusion protein and, as expected, observed increased number of circulating monocytes (only few neutrophils). This was in agreement with increased number of lung IM (Fig. 7e) and with reduction of lung neutrophils. Does CSF1 reduce CXCL1 expression in lungs? The question is related to the fact that Fig. 7b shows increased circulating neutrophils in M-CSF treated mice, which however do not reach the lungs.

Again, the reviewer raises a pertinent point. BAL CXCL1 levels are unaffected by CSF1 treatment (please see Figure S5a), ruling out a role for altered neutrophil recruitment consequent upon local changes in CXCL1 expression. Notably, although we do see a moderate effect of CSF1 on circulating neutrophil numbers (please see Figure 7b), the absolute number of neutrophils in the lung tissue and BAL is reduced by CSF1 treatment. We propose that CSF1 treatment augments clearance of neutrophils thereby altering their number. In line with this, efferocytosis can be influenced by IL-10 and we show that CSF1 treatment increases the numbers of IL-10 expressing interstitial F480+ cells in the lung (manuscript Figure 7r)

-Moreover, considering the striking effects of CSF1 on monocytes mobilization (increased circulating monocytes) and IM accumulation, it would be important to assess as to whether hypoxia reduces M-CSF levels, as well as CSF1R expression by mono/macrophages.

We have addressed this point directly by measuring both circulating CSF1 (M-CSF) and CSF1R expression by monocytes. Interestingly, serum CSF1 levels were undetectable at early time points and increased at day 5 post injury, rebuttal Figure 8a, b, manuscript Figure S6a, b. However, importantly, hypoxia had no effect on CSF1 levels. Hypoxia did result in a modest reduction in CSF1R levels on circulating monocytes in both LPS and bacterial lung challenge models (rebuttal Figure 8c, d). Despite this finding, however, CSF1-Fc was able to both stimulate bone marrow monopoiesis increasing circulating monocyte counts (please see manuscript Figure 7b) and alter the granule profile of the classical blood monocyte population (rebuttal Figure 2, manuscript Figure 7m). a b

Blood monocyte Ly6C+ CD115

Blood monocyte Ly6C+ CD115

C^a 0.0131

Figure 8 CD115 (CSF1R) expression is reduced in hypoxic mice with equivalent circulating MCSF levels MCSF was measured in serum from mice nebulised with LPS and housed in normoxia (N) or hypoxia (H) for 24 hours (a) or 5 days (b) by MSD V-plex as per manufacturer’s instructions. (c) Blood was collected from naïve mice

or mice nebulised with LPS and housed in either normoxia (N) or hypoxia (H) for 24 hours. Expression of CD115 was measured by flow cytometry on Ly6C^{hi} monocytes. (d) Blood was collected from mice that were sham treated (veh) or instilled with *S. pneumoniae* (I) that were housed in normoxia (N) or hypoxia (H) for 24

hours. Expression of CD115 was measured on Ly6C^{hi} monocytes by flow cytometry.

-Fig. 7a-l, the experiments were performed in normoxia and hypoxia. No LPS treatment. Instead, Fig. 7k shows differential gene expression in classical blood monocytes from LPS-treated mice (see legend Fig. 7k). Along the manuscript, remains somehow difficult to discern the effects of LPS from the effects of hypoxia.

We apologise for any confusion caused. All the mice in figure 7 received LPS (please see Figure 7a), the PBS control was to match the installation of CSF1-Fc, and we have clarified the figure legend accordingly. We did not administer CSF1 to normoxic mice as the therapeutic aim was to correct the hypoxic-induced monocytopenia and this was not observed in normoxic LPS-treated mice. Baseline data for normoxia vs hypoxia +/- LPS is provided in figure 2 / 3 (24 hours) and figure 6 (day 5).

-The evaluation of erythropoietic factors (i.e.erythropoietin) and MCSF levels in hypoxia is relevant, since it was shown that inhibition of erythropoiesis by TNF is paralleled by overexpression of PU.1, a key transcription factor driving monocytopoiesis (Grigorakaki C, Morceau F, Chateauvieux S, Dicato M, Diederich M.Biochem Pharmacol. 2011

As detailed above, we have now measured EPO, IL-11 and M-CSF levels, observing the predicted hypoxic induction of EPO and IL-11, but not M-CSF (rebuttal Figures 4a-c and Figure 8 a, b, manuscript Figure 5a, b and Figure Sa and Figures S6a, b).

Reviewer #2

(Remarks to the Author)

In this manuscript, the authors describe the impact of hypoxia on lung injury using peripheral blood samples from ARDS patients and also a mouse model that they propose resembles the disease condition in ARDS patients. There are limited treatment options for ARDS patients other than disease management by supportive care. Thus, the study has clinical significance. The rationale for the mouse model they adopted, however, is unclear. Certainly, the patients are hypoxic at the point they are put on a ventilator. However, once intubated, they are under a hyperoxic condition and in fact hyperoxia induces lung injury. Hyperoxic lung injury is a model for ventilator-induced lung injury. It seems the authors are interested in the effect of hypoxia on immune cells, which would be the condition that precedes intubation. Yet, as mentioned in the 2nd line under results, they collected samples after patients were intubated and at 2 time points-at < 48 h after ventilation which they call early ARDS and at > 48 h-7 days, which they refer to as late ARDS. The patients were clearly not hypoxic at any of these time points. There is also insufficient information about the patients from whom samples were obtained.

We thank the reviewer for recognising the clinical significance of our work. In light of the comments provided, we have now undertaken a more detailed characterisation of the oxygenation of the patients studied. All patients were receiving positive pressure ventilation at time of recruitment and in the 24 hours prior to research sampling. Retrospective review of serial arterial blood gas samples taken from the ARDS patient cohort shows these individuals to be persistently hypoxemic despite high inspired oxygen delivery and positive airway support (please see rebuttal Figure 9a, b, manuscript Figure 1a, b). The duration of the hypoxic exposure observed in the human population is in keeping with the acute experimental mouse models we have used, in which combining LPS-induced ALI with reduced inspired oxygen levels resulted in oxygen saturations of 70%, compared with 98% when challenged in normoxia (21% O₂) (manuscript Figure S2a). We propose that the impact of this initial hypoxic insult has long lasting effects within the bone marrow, blood and tissue compartments. This is now evidenced by a series of re-oxygenation experiments in which animals are exposed to 21% O₂ following an initial acute hypoxic insult (please see rebuttal Figure 10, manuscript Figure 3p-s). We demonstrate that re-oxygenation is unable to fully rescue the lung interstitial macrophage expansion following the initial hypoxic insult, both validating the direct regulation of the immune landscape by

systemic hypoxia and providing evidence of long-lasting effects, even when the initial hypoxic challenge is reversed. Additional granularity regarding patient inclusion criteria, ventilation status and research sampling is provided below in response to each of the specific questions raised.

a

b

Figure 9: Lowest arterial oxygenation measured in patients with ARDS demonstrate substantial hypoxaemia in the 24 hours prior to sampling

(a) Retrospective analysis of the partial pressure of oxygen (PaO₂) from clinical arterial blood gas samples taken from individual ARDS patients in our cohort, in the 24 hours preceding research blood sampling and plotted relative to the normal arterial oxygen range (green shaded area). Each point represents an independent clinical sample, with patients sampled for nanostring platform transcriptional analysis highlighted in the red box. (b) Lowest recorded PaO₂ in the 24 hours prior to blood sampling, with red dots representing patients sampled for nanostring platform transcriptional analysis.

Figure 10 Reoxygenation allows partial, but incomplete, recovery of monocyte recruitment to the lung and tMΦ expansion. Mice were left untreated (naïve) or received nebulised LPS and housed either normoxia (N LPS), hypoxia 10% (H LPS) for 48hours or in hypoxia for 24hours, followed by normoxia for the final 24hours (HN LPS). Blood monocyte (a) proportions and (b) numbers lung tMΦ (c) proportion and (d) numbers

and BAL inflammatory monocyte (e) proportions and (f) numbers were evaluated by flow cytometry.

Specific comments:

1. More granular detail in pulmonary vs extrapulmonary causes of ARDS would be relevant. For extrapulmonary ARDS, are the etiologies all infection/sepsis-related or are other etiologies involved? What were the pathogens for pulmonary causes of ARDS in terms of viral vs bacterial vs unknown vs non-infectious? In Fig 1, how were the 5 ARDS patients selected for monocyte DEG expression. Information should be provided about which patients were selected including their clinical characteristics from Table S1.

We thank the reviewer for pointing out this initial omission, and now provide more granularity with respect to the pulmonary and extrapulmonary aetiologies, and associated pathogens. Of the extrapulmonary ARDS aetiologies 8 of the 9 cases were associated with an infection (please see rebuttal table 1, manuscript Table 1). With respect to monocyte DEG expression, as detailed above, we selected consecutive patients recruited to the study from patient 4-8 (please see rebuttal figure 9a). The lowest PaO₂ values and distribution of PaO₂ values recorded for each of these patients is highlighted in red (rebuttal figure 9bb). Patients with both extrapulmonary and pulmonary ARDS were sampled. This has now been clarified in the manuscript text, see page 55, line 3

Timing of Sampling	Early (n=11)	Late (n=11)
Age (Years)	58.8 (±11.5)	56.9 (±13.6)
Proportion Female Sex- Number (%)	6 (55)	5 (45)

Body Mass Index (kg/m ²):	35.8 (±11.6)	31.1 (±4.2)
APACHE2* Score:	20.4 (±7.2)	18.8 (±8.6)
Pulmonary ARDS- Number (%):	9 (82)	7 (64)
Pulmonary Aetiologies†- Number (%):		
Positive Bacterial Culture:	4 (36)	4 (36)
Positive Viral PCR‡:	2 (18)	1 (9)
Positive Mycology:	2 (18)	2 (18)
No Positive Microbiology Samples:	2 (18)	1 (9)
Extra-Pulmonary Aetiologies- Number (%):		
Faecal Peritonitis:	1 (9)	
Mediastinal Soft Tissue Infection:	1 (9)	
Bacteraemia:		1 (9)
Biliary Sepsis:		1 (9)
Retropharyngeal Abscess:		1 (9)
Non-infective:		1 (9)
Index of Tissue Hypoxia:		
Reference PaO ₂ : 11.1-14.4kPa§		
Lowest PaO ₂ in Hospitalisation Preceding Sampling (kPa):	4.38 (±1.72)	6.42 (±2.23)
Lowest PaO ₂ 24 within Hours Before Sampling (kPa):	6.69 (±2.11)	8.15 (±1.90)
Reference FiO ₂ : 21%		
Highest FiO ₂ in 24 Hours Before Sampling (%):		
Reference Arterial Lactate: 0.5-1.6mmol/L§	68.1 (±21.2)	72.3 (±22.3)
Highest Lactate within 24 Hours Before Sampling (mmol/L):	2.45 (±1.28)	1.56 (±0.96)

Receipt of Organ Supportive Therapies- Number (%):		
Invasive Mechanical Ventilation:	7 (64)	7 (64)
Vasopressors:	7 (64)	6 (55)
Renal Replacement Therapy:	1 (9)	1 (9)
Receipt of Additional Medications- Number (%):		
Dexamethasone¶:	0 (0)	0 (0)
Lopinavir or Ritonavir:	1 (9)	0 (0)
Tocilizumab:	0 (0)	0 (0)
Hydroxychloroquine:	1 (9)	1 (9)

Table 1: ARDS patient demographics and clinical characteristics

Plus-minus data values refer to Mean \pm Standard Deviation. *: Acute Physiology and Chronic Health Evaluation

Score 2 (APACHE2); †: One patient in each group returned mixed fungal and bacterial cultures, which could not be causatively differentiated; ‡: Two patients in the “Early” group and one patient in the “Late” group were SARSCoV-2 positive; §: Reference ranges as indicated by local health board (NHS Lothian). ¶ No other corticosteroids were administered.

2. The human data provided do not support that the lung microenvironment found in the mouse experimental data is representative of human ARDS. Why did the authors not sample peripheral blood before each patient was put on a ventilator to study the actual hypoxic condition? The first figure does not have any human data on the lung environment in ARDS. One cannot assume that interstitial macrophages are also low in the human lung during ARDS. It is possible that the authors do not have an approved protocol for lavaging the lungs of ARDS patients although BAL fluid should not contain many interstitial macrophages (IMs). Monocytopenia in the periphery does not allow one to draw conclusions about the lung immune environment.

Timing of blood draws was limited by ethical constraints and the need for consent by proxy. As detailed above, we have now undertaken a detailed retrospective analysis of the

distribution of PaO₂ values recorded across the 24 hour period prior to sampling. For inclusion all patients had, by definition moderate-severe ARDS according to the Berlin criteria and 1. were within 1 week of a known insult or new or worsening respiratory symptoms, 2. had bilateral opacities on chest radiograph, 3. had evidence of respiratory failure 4. PaO₂/FiO₂ < 200 mmHg with PEEP > 5cm H₂O, 5. were less than 7 days from ventilatory support. The distribution of PaO₂ values in the 24 hours leading up to blood sampling verifies that at time of sampling all patients demonstrated significant and sustained hypoxaemia (please see rebuttal Figure 9a, manuscript Figure 1a), despite supplementary oxygen therapy and ventilatory support (please see rebuttal figure 1b, manuscript Figure 1c). With respect to the lung environment, we did not have ethical consent to undertake BAL in patients with ARDS. We would, however, argue the phenotype we observe in murine models lies within the tM \square compartment which would only be accessible by lung biopsy to provide sufficient cell numbers for experimental work. Mechanistic studies to delineate the ontogeny of tM \square populations were therefore conducted in murine models. We acknowledge this is a limitation of our work and have now added text to the discussion clarifying these limitations, please see page 19, line 13 and page 22, line 18.

3. Mechanistically it appears that in the mouse hypoxia caused bone marrow suppression/altered myeloid differentiation and function in the face of LPS-induced ALI. Is this specific to lung injury or is this altered myelopoiesis due to hypoxia in response to any type of systemic inflammation? It is written as if it is specific to ALI/ARDS, but there is no proof of that comparing the results to systemic LPS administration. While one could rationalize that severe hypoxia elicited by LPS+hypoxia induces the peripheral aberrant immune response, the model is not designed to mimic the condition of a patient on a ventilator when they are not hypoxic any longer. It will be interesting to compare hypoxic mice to those under hyperoxia. This would be more representative of a hypoxic patient with lung injury being rescued with high FiO₂ on a ventilator.

We would argue that the hypoxic bone marrow suppression is not specific to LPS-induced ALI, given we replicate our observations in a model of severe streptococcal pneumonia (D39) and in an Influenza A model of virally induced-epithelial injury. Whilst at the dosing

strategy used D39 infection is associated with a bacteraemia, these pathogens were all administered via the intra-tracheal route. To directly address the consequence of systemic inflammation, we have now also undertaken new experimental work, in which we challenge mice with systemic (intraperitoneal) LPS. In keeping with our previous observations, systemic hypoxia results in a failure to expand inflammatory macrophage populations within the peritoneal cavity (please see rebuttal Figure 11). Thus, the phenotype observed is not specific to the site of injury, but is dependent upon the presence of systemic hypoxia, which we observe to be sustained in the ARDS patient cohort, despite oxygen therapy and ventilatory support as detailed above. Evidence of long-lasting consequences of exposure to 24 hours of hypoxia, matching the distribution of patient PaO₂ values, is also now provided in a new series of re-oxygenation experiments, as detailed above (please see rebuttal Figure 10, manuscript Figure 3p-s). In this work, we demonstrate that reoxygenation is unable to fully rescue the tM \square following the initial hypoxic insult. Importantly these animals were returned to normoxia for a 24 hour period prior to sampling, to mirror the recorded patient PaO₂ values, where significant hyperoxia is not observed in the clinical setting (please see rebuttal Figure 9a, manuscript Figure 1b),

peritoneal lavage inflammatory macrophages

Figure 11 Systemic hypoxia hinders inflammatory macrophage number expansion in LPS-induced peritonitis

Mice were inoculated with intraperitoneal LPS and housed in normoxia (N) or hypoxia (H) for 24 hours. Peritoneal lavage fluid was assessed by flow cytometry. Inflammatory macrophages were identified (Live Singles CD45⁺Lin⁻ (CD3/ CD19/ SiglecF/ Ly6G) CD11b⁺Ly6C⁺) and quantified.

4. The clinical data comparing ARDS to healthy controls is making a comparison between a patient with a critical illness syndrome and a healthy patient. The data would be more compelling if ARDS patients were compared to other critically ill patients without ARDS (and by extension, other critically ill patients without hypoxia).

As detailed above, and in response to reviewer 3 below, we propose to clarify that our program of work was designed to explore the importance of tissue hypoxia in reshaping the immune response in the lung. It is incredibly challenging to identify a robust human control group to test the effect of this exposure. Although it is true that critically ill patients without ARDS do not have alveolar injury, and therefore do not have a gas transport issue, they frequently have poor tissue perfusion due to shock (e.g. reduced cardiac output, microembolic phenomena, and interstitial oedema). Therefore, we could not rule out tissue hypoxia (our exposure of interest) in critically ill patients without ARDS. We accept that a healthy-control populations also have limitations, and we acknowledge this as a limitation of our work. Our approach is supported by comments from reviewer 3. We apologise for our lack of clarity with respect to hypoxia and tissue hypoxaemia and the incomplete justification for our program of work. We have now amended the text to reflect the approach taken, please see page 6, line 1, and in our revised results section.

5. The authors have attempted to explain that their findings of monocytopenia in ARDS is at odds with what multiple studies have described in COVID patients overwhelmingly associating monocytes with more severe

disease. I was not able to comprehend their explanation. In non-COVID ARDS as well, a gene signature associated with monocytes identifies a risk score (Jiang et al., JCI insight, 2020).

Preliminary survey of total inflammatory macrophage airspace counts, obtained by BAL, as well as lung monocyte and tM ϕ counts, following re-oxygenation, would suggest an early but partial recovery in numbers (please see rebuttal figure 10, manuscript figure 3p-s). We speculate that this recovery in numbers may be in keeping with the published broncho-alveolar lavage data from patients with severe COVID disease in which the presence of large numbers of inflammatory monocytes is described (Liao et al Nature Medicine 2020). The reduction in circulating monocyte counts that we describe, does match the reduction in the proportion of CD14+ monocytes described in patients with critical COVID-19 disease (Stephenson et al Nature Medicine 2021). We would also, however, suggest that the phenotype of the recruited monocytes is critical to their function within the tissues. Importantly, an impaired blood type I IFN response has been observed in the context of severe and critical patients with COVID-19 disease (Hadjadj et al Science 2020), which we replicate in our model of hypoxic lung injury, providing a mechanism by which systemic hypoxia per se shapes the immune response. In the manuscript from Jiang and colleagues (mentioned by the reviewer), the authors compare the circulating monocyte transcriptome of 4 ventilated sepsis patients and 3 ARDS patients (1 viral, 2 unidentified causes), all of whom also have a diagnosis of pneumonia. As highlighted by reviewer 3, it is difficult to ascertain the difference in tissue hypoxia between these patient groups, making it challenging to directly compare their findings to our work. Importantly, Liao and colleagues have observed enrichment of markers of immaturity (CD14), inflammatory proteins (100A8), and cytokines (CCL3) (Liao et al Nature Medicine 2020) in BAL monocytemacrophages in patients with severe SARS-Cov2 infection. We now provide new data in which we observe CSF1 treatment not only to expand the tM ϕ niche, but also to suppress the expression of these markers, underpinning the importance of phenotype as well as cell number (please see rebuttal figure 12, manuscript figure 7t). We have now added text to better reflect both the current literature in COVID-19 disease, and the observed changes in monocyte / macrophage phenotype, please 21, line 1.

Figure 12 Hypoxia drives a pro-inflammatory signature in MHCII-tM \square that is suppressed in CSF1-induced Lyve1+MHCII+tM \square

Volcano plot denoting differentially regulated genes of interest from Lyve1+MHCII-tM \square from CSF1-treated mice and MHCII-tM \square from PBS-treated mice nebulised with LPS and housed in hypoxia for 5 days that were sorted by flow cytometry for nanostring platform analysis.

Minor points:

Many relevant publications have not been cited. IL-10-expressing regulatory interstitial myeloid cells that help in resolution of lung inflammation in models of pneumonia and lung injury are described in the literature. They should cite previous clinical applications of colony stimulating factors in ARDS in their

discussion if they are proposing it as a potential therapeutic. IV GM-CSF was trialed in ARDS and failed

(<https://www.ncbi.nlm.nih.gov/pmc/articles/PMC3242850/>). Inhaled GM-CSF was tried in a case series (<https://www.atsjournals.org/doi/full/10.1164/rccm.201311-2041LE>).

We thank the reviewer for highlighting this relevant work, and have included reference to it in the revised manuscript, please see page 21, line 24.

Reviewer #3

(Remarks to the Author)

In their manuscript “Hypoxia shapes the Immune Landscape in Lung Injury Promoting Inflammation

Persistence,” Mirchandani and colleagues seek to understand how hypoxia impacts the resolution of inflammation in the injured lung. The investigators begin by showing that fewer monocytes are present in the circulation of patients with ARDS and that monocytes are reduced in the circulation of mice exposed to inflammatory stimuli and low ambient oxygen. Elegant studies of the bone marrow demonstrate lower levels of monocyte precursors in mice subjected to hypoxic conditions and suggest that impaired type 1 interferon signaling is a mechanism. These are followed by experiments in which administration of CSF1-Fc increases lung monocyte numbers while attenuating lung neutrophil accumulation and capillary leak.

The paper focuses on the resolution of lung inflammation in the context of ARDS, and therefore touches on an important area. The paper is well written and the experiments are logically presented. Appropriate controls are used and statistical analysis is sound. Investigation into type 1 interferon signaling

and the role of CSF1 in the resolution are novel and represent strengths of the paper. However, there are several limitations to the manuscript.

The main limitations include: a) conceptual issues with hypoxia versus hypoxemia, b) potential issues with identification of IMs and lack of experiments to show their transcriptional profiles, c) incomplete establishment of links between interferon signaling, hypoxia and CSF1. A detailed description of each of these is described below.

We thank the reviewer for their interest in our work and would like to address each of the specific concerns raised below. In particular, we have improved clarity with respect to hypoxaemia and tissue hypoxia. **a).** A description of the severity of hypoxemia has now been provided by the addition of a detailed retrospective review of the partial pressure of oxygen in clinical arterial blood samples obtained from each ARDS patient in the 24 hour period leading up to research blood sampling, detailed below. We agree that the consequence of hypoxaemia, together with impaired tissue perfusion and lung consolidation, will drive tissue hypoxia. To model this in our murine studies we experimentally limited the fraction of inspired oxygen to which animals were exposed, thus allowing us to explore how tissue hypoxia (bone marrow, blood, lung compartments) shapes the immune landscape in response to lung injury. We have amended the text carefully to address this. **b).** To address the issues regarding the identification of IMs, we have re-named all lung non-AM CD64⁺Siglec F⁻ macrophages “tissue macrophages (tM \square)” and Ly6C⁺IM “monocyte-derived macrophages (MDM)” as helpfully suggested. whilst renaming the remaining Siglec F⁻Ly6C⁻ lung digest macrophages “Ly6C⁻ tissue macrophages (Ly6C⁻ tM \square)”.

All original 24 hour and day 5 lung counts were undertaken after bronchoalveolar lavage, with all original day 5 BAL undergoing flow cytometric analysis allowing differentiation of tissue macrophage populations, from inflammatory airspace monocyte-derived macrophages. We now provide the SSC plots of the lung macrophage populations, and have generated new data to delineate the transcriptional profile of the lung Ly6C⁻ tM \square populations, please see detailed response below. **c).** The hypoxic regulation of circulating and bone marrow levels of type I and type II interferons is now provided, together with IFNAR expression within the bone marrow, blood and lung monocyte and macrophage compartment, detailed below. To more directly explore the link between interferon

signalling and CSF1, we have now treated the *lfnar1* KO mice with CSF1 following LPS mediated acute lung injury, as detailed below.

Major comments:

1. Hypoxia is a main element of the paper. However, there are issues with how the term is used, whether it applies to the human cohort, and whether it is present in the animal models. These issues are described in detail below. Importantly, the human studies are quite nice, and the animal model is elegant.

Careful rewriting of the paper should be able to address the issues.

We thank the reviewer for acknowledging the human studies and murine models and, as helpfully suggested, have undertaken a careful re-write to address the issues raised, in the context of new clinical data detailing the degree and duration of patient hypoxaemia, and in the animal models, measures of hypoxaemia and tissue hypoxia, evidenced by HIF expression.

a. A main concept on which the paper is based is systemic hypoxia. The way this is portrayed in the manuscript tends to cause confusion, since low PaO₂ levels are provided as evidence of hypoxia in the authors' cohort. Importantly, PaO₂ is a measure of hypoxemia (rather than tissue hypoxia). To this point, many patients with ARDS have adequate PaO₂ levels as a result of supplemental oxygen administration (generally with mechanical ventilation). Moreover, when arterial hypoxemia is present it is generally transient. Conversely, hypoxia is common in patients with shock as a result of poor tissue perfusion.

Lactate levels would be a more adequate measure of tissue hypoxia.

We thank the reviewer for this helpful point. We have now undertaken a retrospective analysis of the partial pressure of oxygen (PaO_2) values taken from clinical arterial blood gas samples from individual ARDS patients, in the 24 hours preceding research blood sampling. As detailed above in response to reviewer 2, all patients were receiving positive pressure ventilation at time of recruitment and in the 24 hours prior to research sampling.

Retrospective review of serial arterial blood gas samples taken from the ARDS patient cohort shows these individuals to be persistently hypoxaemic despite high inspired oxygen delivery and positive airway support (please see rebuttal Figure 1a, b, manuscript Figure 1b, d). In 13 of the 22 patients, there is an elevation in circulating lactate levels, indicative of ongoing tissue hypoxia despite fluid resuscitation and the use of vasopressors (please see rebuttal Figure 13, manuscript Figure 1d).

Figure 13: ARDS patients have evidence of raised plasma lactate

The highest recorded arterial plasma lactate recorded in the 24 hours preceding research sampling was recorded. Samples in red denote lactate levels \square upper limit of normal with normal local reference values in shaded green (0.5-1.6mmol/L)

b. **The rationale provided for experiments in Figure 1 is “to determine whether hypoxia is associated with alterations in the blood monocyte compartment.” However, the experimental design doesn’t adequately enable testing of the association. First, subjects with ARDS will have systemic inflammation that will potentially alter monocyte transcriptomes compared to**

controls. Second, while patients with ARDS can have hypoxia, in most cases it is corrected with supplemental oxygen. Third, while a few of the genes presented in the figure are associated with hypoxia, many quintessential monocyte hypoxic genes (i.e. VEGF) are not demonstrated (PMID: 16849508). The experiment itself is fine – the rationale for the experiment and a link to hypoxia is the problem.

We have now modified the text referencing figure 1 to clarify the rationale for the experimental work undertaken. We absolutely agree that the monocyte transcriptome will reflect both the physiological and inflammatory signals to which the patients have been exposed, and will not be a sole consequence of changes in either circulating or regional oxygenation. To more directly phenotype the circulating monocyte populations, we have now undertaken a proteomic survey of classical blood monocytes in which we do identify a hypoxic signature that reflects some of the gene changes described by Bosco et al. following human monocyte culture in in-vitro hypoxia (Bosco et al JI 2006). As highlighted, some archetypal hypoxia responsive genes are unexpectedly downregulated ie VEGF, perhaps reflecting the complexity of human disease and heterogeneity in the etiology of ARDS. We do however observe, as detailed above, that all patients sampled are exposed to significant and sustained hypoxaemia, irrespective of disease aetiology, and it is this observation that has led us to question the importance of hypoxia per se in shaping the immune response following lung injury. We have amended the text accordingly, please see page 5, line 7, page 6, line 14 and page 19, line 3

c. Analysis of monocytes from animal experiments show that genes are differentially expressed. However, it is unclear if hypoxia is the main mechanism. The authors have previously explored HIFresponsive genes (including key metabolic genes). Assessment of these would help support the link between hypoxia and monocyte profiles.

The duration of the hypoxic exposure observed in the human population is in keeping with the acute experimental mouse models we have used, in which combining LPS-induced ALI with reduced inspired oxygen levels resulted in oxygen saturations of 70%, compared with 98% when challenged in normoxia (21% O₂) (please see manuscript Figure S2a). As requested, to more directly explore acute activation of the HIF response in the hypoxic lung

injury model, we have also now undertaken HIF expression profiling of both the bone marrow and lung tissue compartment by immunohistochemistry. We observe increased tissue HIF protein expression following 24 hours of hypoxic exposure (please see rebuttal figure 14, manuscript Figure 3a). To more directly address the regulation of tM \square expansion by hypoxia, we have now undertaken a series of re-oxygenation experiments in which mice are exposed to normoxia (21% O₂) 24 hours following hypoxic LPS challenge. In these experiments we observe a partial recovery of the tM \square compartment upon re-oxygenation, indicative both of oxygen dependence and longlasting effects of hypoxia, even when the initial challenge is reversed (please see rebuttal figure 10, manuscript Figure 3p-s).

Figure 14 HIF protein is expressed in the lung post-LPS and is enhanced in the context of hypoxia Mice were treated with LPS and placed in normoxia (N LPS) and hypoxia (H LPS) for 24 hours. Mice were sacrificed and lungs collected as before. Formaldehyde-fixed lung sections were stained for HIF1 and dapi. The image acquisition and processing steps were performed using the same settings for both sample groups.

2. Identification of interstitial macrophages in the lungs may be flawed based on the scheme shown in Fig 3.

a. The Ly6C hi cells that are reported to be IMs have the characteristics of monocyte-derived macrophages, many of which will be in the airspaces. Examination of the BAL would likely show these populations. Side-scatter might help separate the IMs (PMID: 23672262). However, use of intratracheal antibodies would be more definitive (PMID: 28850249).

We thank the reviewer for raising this important point and, on reflection, can see how this could have created confusion. We want to reassure the reviewer all lung digests were

performed after BAL had been performed. All original day 5 lung counts were undertaken after harvest of airspace cells by bronchoalveolar lavage, allowing differentiation of tissue macrophage populations, from inflammatory airspace monocyte-derived macrophages. Staining with CD11b, CD11c, CD64 and Siglec F was used to delineate alveolar macrophages from BAL and lung tissue (in keeping with PMID: 23672262), rather than SSC profile, as the cells were fixed prior to analysis. SSC and FSC properties are now included below as requested with prior shown gating strategy applied as per manuscript figure 3, (please see rebuttal figure 15a). Furthermore, we have now addressed this point in the manuscript and, in order to avoid confusion, we have re-named all non-AM CD64⁺SiglecF⁻ macrophages “tissue macrophages (tM ϕ)”, whilst renaming the Ly6C⁺IM “monocyte-derived macrophages (MDM)” as helpfully suggested. We have also renamed the remaining non-AM CD64⁺SiglecF⁻Ly6C⁻ lung digest macrophages “Ly6C⁻ tM ϕ ”. We have clarified this nomenclature accordingly in the text and hope the reviewer finds this addresses this important point.

In contrast to the previously published expansion of monocyte derived alveolar macrophages following bleomycin challenge (Misharin J Exp Med 2017), we do not detect this interim population in the LPS model (please see rebuttal figure 15b). This is consistent with recent fate-mapping by the Ginhoux lab using the Ms4a-Cre mouse which showed that monocyte-derived alveolar macrophages do not arise following LPS-induced injury (Liu et al Cell 2019). We have also now undertaken a further series of experiments at 24 hours in which lung counts (neutrophil, inflammatory macrophage and alveolar macrophage) are obtained following bronchoalveolar lavage, and include the paired lung counts (please see rebuttal figure 16).

Figure 15 AM and tM ϕ have distinct SSC, FSC and SiglecF profiles with AM remaining of host origin postLPS

(a) AM (singles Live CD45⁺ Lin⁻CD64⁺SiglecF⁺CD11C⁺) and tM ϕ (singles Live CD45⁺ Lin⁻CD64⁺SiglecF⁻CD11C⁺) have distinct FSC, SSC and SiglecF profiles. (b) The proportion of AM from host origin (CD45.1+CD45.2+) were quantified from lung-protected chimeras left naïve in normoxia (N) or exposed to nebulised LPS and housed in either normoxia (N) or hypoxia (H) and treated with PBS or CSF1.

Figure 16: Hypoxia prevents tM ϕ expansion following LPS without hindering Inflammatory macrophage migration to alveolar space

Mice were left naïve or exposed to nebulised LPS and housed in normoxia (N) or hypoxia (H) for 24 hours. BAL, followed by lung digestion following gentle perfusion was performed and **(a)** BAL Inflammatory/ tissue lung macrophage, **(b)** AM and **(c)** BAL and lung neutrophil numbers were quantified by flow cytometry.

b. The tsne plots shown in figure 7m may help identify IMs. A panel showing non-inflamed mice is essential to provide baseline data. In addition, regions showing resident AMs, monocyte-derived recruited AMs, resident tissue IMs and monocyte derived IMs should be indicated. In the supplement, dot plots with appropriate FMO controls are needed.

We thank the reviewer for raising this point and on reflection we agree that showing the naïve versus LPS-treated dot plot data ensures the key message that CSF1-induces a Lyve1⁺ population within the Ly6C⁻ tM ϕ compartment is much more clearly represented (please see rebuttal Figure 17a, b, manuscript Figure 7n, o). In addition, we provide the gating strategy and appropriate FMO in the supplementary figure (please see Figure S6m, as helpfully suggested (please rebuttal figure 17c, manuscript Figure S6m).

Figure 17: CSF1 induces the expansion of Lyve1+MHCII-tMφ

(a) Representative dot plots of Lyve1 expression in MHCII-tMφ gated on CD64^{bright}SiglecF⁻ Ly6C^{MHC} tMφ from naive or LPS-treated mice housed in normoxia (N) or hypoxia (H) and treated with PBS or CSF1 (CSF1). (b) Proportion of Lyve1⁺ cells within Ly6C^{MHCII} tMφ compartment. (c) Gating strategy for identification of the different monocyte and macrophage populations in the lung, including Lyve1+tMφ, and associated APC FMO control.

c. The authors suggest that Lyve-1+ MHCII negative IMs expand in response to CSF1. These cells should be indicated in the tsne plots in Fig 7m.

In addition, flow plots with gating strategies should be shown to confirm this finding and data from Fig 7n. Side scatter should be included to help distinguish IM and AMs.

As detailed above in response to point 2 b, we now provide dot plots delineating the expansion of the Lyve-1⁺ MHCII⁺ M ϕ in response to CSF1 (please see rebuttal figure 17a, b, manuscript Figure 7n, o), with additional gating strategies and side scatter profiles provided (please see rebuttal figure 17c and Figure 15a, manuscript figure S6m).

d. In Figure 7p many of the F4/80 positive cells appear to be alveolar macrophages rather than IMs. Moreover, AMs are notably absent from the H PBS image. The authors should confirm that exposure times are similar between images. Co-staining with Lyve-1 would help confirm the data from Fig 7n.

We have now confirmed within the manuscript that exposure times were equal between the groups, please figure legend xx and have undertaken co-staining of Lyve-1 with F4/80, showing expansion of co-expressing interstitial macrophages in the CSF1 treated mice exclusively (please see rebuttal figure 18, manuscript Figure 7s).

Figure 18 CSF1 treatment drives Lyve1+ IM expansion in lungs post LPS

Mice were treated with LPS and placed in hypoxia for 5 days during which time they were treated with PBS (H PBS) or CSF1 (H CSF1). Formaldehyde-fixed lung sections were stained for Lyve1, F480 and nuclear dapi. Representative sections shown. The image acquisition and processing steps were performed using the same settings for both sample groups.

3. The link between type 1 interferon signaling and hypoxia are incompletely studied. The investigators show that expression of interferon responsive genes is attenuated in BMDMs cultured in 1% oxygen, and that INFAR expression is reduced in LSK cells isolated from LPS treated mice exposed to 10% FiO₂.

However, attempts to link this to reduced monocyte and macrophage function are missing.

a. Experiments that demonstrate reduced interferon signatures in monocytes from humans and/or mice would provide additional reassurance as would measurement of expression profiles of IMs.

We have worked hard to provide additional data to strengthen the link between type 1 interferons and hypoxia. Firstly, our new data show selective suppression of circulating type I interferons in mice exposed to hypoxia, with preserved levels of type II interferons (please see rebuttal figure 6, manuscript Figure 5c, S5b, c). Secondly, we show that in addition to the suppression of *Ifnar* expression within the LSK compartment, we also observe hypoxia-induced suppression of *Ifnar* expression in the circulating Ly6C⁺ monocyte population (please see rebuttal Figure 19a and manuscript Figure S5d). Of note, an impaired blood type I IFN response has also been observed in the context of severe and critical patients with COVID-19 disease (Hadjadj et al Science 2020) in keeping with our conclusion that hypoxia per se can alter the immune response.

Within the lung compartment, elevated levels of IFNAR are described in the MHCII⁺ interstitial macrophages and Ly6C⁺ interstitial macrophages at 24 hours, but this is not

associated with the induction of a type I Interferon transcriptional signature within the non-AM tissue macrophage compartment at the later timepoint, when circulating IFN levels were equivalent in hypoxia and normoxia (please see rebuttal figure 19).

Figure 19 IFNAR expression changes in response to hypoxia varies between different tissues

IFNAR expression was measured by flow cytometry in (a) classical blood monocytes, (b) lung MHCII+Ly6C- tMφ and (c) lung MDM in mice treated with nebulised LPS and housed in either normoxia (N LPS) or hypoxia (H LPS) for 24 hours.

b. Do the investigators think that type 1 interferons are reduced in states of systemic hypoxemia, or are the downstream IFN signaling mechanisms defective? This could be addressed in the discussion

Please see response to point 3a above, new discussion text added page 20, line 18.

c. If the main detrimental effect of IFNAR deletion is to prevent monocyte production by the bone marrow can this be overcome by administration of CSF1-Fc?

Thank you for raising this interesting question. We have now administered CSF1-Fc to the *Ifnar1*^{-/-} mice and observe that CSF1-Fc can rescue the phenotype by increasing circulating monocyte counts following LPS, with an associated increase in body weight (please see rebuttal figure 20, manuscript figure 7u-x, Figure S6n and Figure S6o).

Figure 20 CSF1 overcomes ifnar deficiency enhancing monocyte lung recruitment, tM \square expansion and accelerating weight recovery in lung injured mice

Ifnar^{-/-} (KO) mice were nebulised with LPS and treated with PBS (KO LPS) or CSF1 (KO LPS+CSF1). Mice were sacrificed on day 5 and (a) blood monocyte, (b) lung Ly6C⁺ monocytes, (c) lung tM \square (d) lung Lyve1+MHCII⁻ tM \square (e) BAL inflammatory macrophage were determined by flow cytometry. (f) Weight change from baseline was also measured.

4. Administration of CSF1-Fc impacts monocyte and macrophage numbers. However, it is unclear what roles these expanded cells play.

a. Do the expanded IMs play a role in dampening inflammation?

Recently Zhou et al demonstrated a key role for IL-10 production from IM as a driver of inflammation resolution in lung injury (Zhou et al Nature Immunology 2020). We observe that CSF1 expands the IL-10-expressing tM \square , which we propose will enhance neutrophil clearance from the lungs in response to CSF1. As detailed in response to reviewer 2, point 5, we have also now undertaken transcriptional analysis of the Ly6C⁺tM \square (further subdivided by their expression of MHCII) populations in the lung of hypoxia LPS-treated mice, to provide further granularity with respect to their role in inflammation suppression in response to treatment with CSF1. In the MHCII⁻ population we observe CSF1 to suppress expression of both pro-fibrotic genes (mmp12, mmp8, Fpr1, Pdgfb and Col14a1) and pro-inflammatory (IL6, IL-1b, Tnf, IL18, S100a8, CCL3 and CXCL1) genes associated with poorer outcomes in ARDS (Meduri et al Chest 1995)(please see rebuttal figure 12, manuscript Figure 7t).

b. Does the CSF-1 affect AM function to dampen inflammatory responses? In other words, is there a salutary effect of CSF-1 that is independent of monocyte recruitment?

This is an interesting question, which will form the basis of a future program of work. We have provided significant new data to strengthen the mechanisms by which CSF1 regulates IL-10 and pro-reparative transcriptional signatures in the IM population, as detailed above.

Preliminary data quantifying AM expression of the integrin receptors (CD11b and CD11c) would suggest that hypoxia inhibits the LPS mediated induction of both CD11b and CD11c, but that CSF1 is unable to overcome this hypoxic suppression (please see rebuttal figure 21).

Figure 21 Hypoxia alters AM surface integrin expression which remains unchanged following CSF1 treatment

Lung digests from naïve mice or mice treated nebulised LPS, that were housed in normoxia (N) or hypoxia (H) for

24 hours were obtained. AM were identified as Live, CD45+Lin-CD64+SiglecF+CD11c+ and their expression of (a) CD11b and (b) CD11c measured by flow cytometry. Lung digest AM (c) CD11b and (d) CD11c surface expression from mice treated with LPS, housed in hypoxia (H) and treated with PBS (H LPS) or CSF1 (H LPS+CSF1) was obtained by flow cytometry.

Minor

Table 1 should be expanded. Measures of tissue hypoxia (such as lactate levels) would be helpful, as mentioned above. In addition, whether or not glucocorticoids were administered should be listed. These are now commonly used in ARDS and have been shown to cause monocyte apoptosis.

We apologise for this oversight, table 1 has been significantly expanded as detailed in response to reviewer 2, point 1. This includes information regarding glucocorticoid administration, a separate graph of lactate levels at time of sampling is provided as detailed in point 1a above. Additional detail has also been provided with respect to the aetiology of ARDS and pathogen detection.

Subjects with ARDS have reduced levels of classical monocytes. In comparison, classical monocytes are increased in hypoxemic mice exposed to inflammatory stimuli. Since classical monocytes are the precursors to recruited macrophages this seems relevant. How do the authors interpret this finding in the context of their model?

This is an interesting question, which may reflect the time point at which patients were sampled, or a divergence in growth factor requirements. We have added discussion text to the manuscript to reflect this point, please see page 20, line 6.

The text describes Figure 6E as showing interstitial neutrophils. Intravascular neutrophils may be included in this assessment. Describing the output as total lung neutrophils (as in the figure) may be more accurate.

Thank you for this helpful suggestion, we have corrected the text to read total lung neutrophils

Raw data for chimerism experiments should be shown. It will be instructive to see how effective engraftment and shielding are for this set of experiments.

Thank you for this helpful suggestion, raw data is now included in Figure S6j.

Decision Letter, first revision:

Subject: Decision on Nature Immunology submission NI-A31918A

Message: 21st Dec 2021

Dear Dr. Mirchandani,

Your Article, "Hypoxia shapes the immune landscape in lung injury promoting inflammation persistence" has now been seen by 3 referees. Some important points were raised. Although we are very interested in the possibility of publishing your study in Nature Immunology, the issues raised by the referee need to be addressed.

We therefore invite you to revise your manuscript taking into account all reviewer comments. At resubmission, please include a "Response to referees" detailing, point-by-point, how you addressed each referee comment. If no action was taken to address a point, you must provide a compelling argument. This response will be sent back to the referees along with the revised manuscript. Please highlight all changes in the manuscript text file in Microsoft Word format.

Please include a revised version of any required reporting checklist. It will be available to referees to aid in their evaluation. The Reporting Summary can be found here: <https://www.nature.com/documents/nr-reporting-summary.pdf>

When submitting the revised version of your manuscript, please pay close attention to our <https://www.nature.com/nature-research/editorial-policies/image-integrity> Digital Image Integrity Guidelines. and to the following points below:

Please use the link below to submit your revised manuscript and related files:
[REDACTED]

We hope to receive your revised manuscript within 3-4 weeks. If you cannot send it within this time, please let us know. We will be happy to consider your revision so long as

nothing similar has been accepted for publication at Nature Immunology or published elsewhere.

Nature Immunology is committed to improving transparency in authorship. As part of our efforts in this direction, we are now requesting that all authors identified as 'corresponding author' on published papers create and link their Open Researcher and Contributor Identifier (ORCID) with their account on the Manuscript Tracking System (MTS), prior to acceptance. ORCID helps the scientific community achieve unambiguous attribution of all scholarly contributions. You can create and link your ORCID from the home page of the MTS by clicking on 'Modify my Springer Nature account'. For more information please visit www.springernature.com/orcid.

Sincerely,

Ioana Visan, Ph.D.
Senior Editor
Nature Immunology

Tel: 212-726-9207
Fax: 212-696-9752
www.nature.com/ni

Reviewers' Comments:

Reviewer #1:
Remarks to the Author:
No further comments

Reviewer #2:
Remarks to the Author:
The authors have addressed many of the concerns raised in the original review of the manuscript. A few issues remain.

One detail in their protocol of characterizing the lung tissue immune cells is missing. As the authors are aware, although vascular perfusion removes leukocytes from the lung vasculature, it does not completely eliminate all cells. Thus, where there is a need to distinguish between tissue and circulating leukocytes, as in this study, anti-CD45 antibodies given intravenously followed by quick harvest of the lungs allows gating out of cells in circulation. Did the authors do that? Please see protocol by Gibbings and Jakubzick in Lung Innate Immunity and Inflammation (Methods and Protocols) edited by Scott Alper and William Janssen. If they did not they should mention this as a limitation and that their

estimates include tissue cells and those in the vasculature.

The description of the experiment and also the labeling of Fig panels in Fig. 3 is misleading. The legend mentions (similar to Fig. 2), the mice were treated with LPS first and then exposed to normoxia or hypoxia (although panel d legend is also incorrectly written). The following text gives the impression it was the opposite-i.e. normoxia or hyperoxia was followed by LPS. "As expected, LPS significantly increased neutrophil numbers in the lung (Figure 3b) in both hypoxia and normoxia, with equivalent numbers of T cells and B cells in the lung (Figure S3a and Figure Sb). However, whilst LPS administration also led to a significant expansion of the macrophage (CD45+ Lineage- CD64bright) compartment in normoxia, this was completely abolished in the setting of hypoxia (Figure 3c and 3d)."

The authors did not use the marker CD43 now routinely used to distinguish between classical and non-classical monocytes, classical Mos being Ly6ChiCD43- and non-classical Mos Ly6Clo/-CD43+. Since a major conclusion of the authors is deficiency of Ly6C+ MDMs, with the markers they have used (Ly6C/CD11b/CD64/SiglecF) can they distinguish between classical Mos and Mo-derived macs under conditions of LPS/H vs LPS/N? Is the defect in recruitment (although a better term would be accumulation) of classical Mos or their differentiation to macs? What are Ly6C+ MDMs (Figs 3h and 3i)? Can they define them better with reference to the literature? This is important to discuss further since a major conclusion is failure to expand monocyte-derived tissue macs.

Reviewer #3:

Remarks to the Author:

Mirchandani and colleagues have improved their manuscript "Hypoxia shapes the immune landscape in lung injury promoting inflammation persistence" by adding additional data and refining their results and discussion. Of particular note, the authors have expanded their description of human subjects and provide compelling data that suggest that the study population suffered from hypoxia. In addition, they provide new data supporting HIF-1alpha driven pathways and have firmed up links involving CSF-1 and IFNAR.

Taken as a whole, the findings of the manuscript are expected to be of high impact and should advance the field. However, there are four items that should be addressed to complete the work. These are discussed in detail below as Major Comments. In brief, the points include:

1. Refinement of terminology for macrophage populations and recognition of monocyte derived macrophages in the airspaces.
2. Clarification of origin of Ly6c negative tissue macrophages.
3. Shoring up data that suggest IL-10 production by tissue macrophage subsets
4. Improved presentation of tissue immunofluorescence

Major Comment 1

In response to the prior review, the authors have refined terminology and classification of macrophages. However, the classification remains confusing, and interpretation of data may be incorrect. The main issue revolves around monocyte derived macrophages that populate the airspaces after LPS exposure (MDM AMs). These cells are most easily seen on Figure 6c, middle panel, in which BAL was performed 5 days after LPS. Since the figure

shows BAL, it is clear that the cells are in the airspaces (i.e. AM). If the gating strategy from Fig 1 was applied to these cells, it would most likely show that the MDM AMs are CD11c low-mid, Siglec-F-, and Ly6C high. The MDM AMs will also be CD64 positive (although it is possible their CD64 could be slightly less than the resident AMs). As a consequence, under the current strategy the MDM AMs are likely to be misidentified defined as tissue macrophages (tM) and would further be classified as MDM tM.

The authors suggest in their rebuttal that: a) lavage can effectively remove all AMs from the lung and b) monocyte-derived AMs do not exist in the LPS model. Neither of these are true. Substantial numbers of AMs remain in the lungs even after aggressive lavage (PMID: 8586127). The presence of MDM AMs in BAL following LPS administration is shown in a number of papers including (PMID 16543608, 33117383, 21471090, 28421818, 21278303). The cite Liu et al (Cell 2019) as evidence that MDM AMs do not exist in the LPS model. However, only Siglec-F + CD11c+ cells were assessed for the Ms4a3 reporter in the paper; MDM AMs were not assessed. While I acknowledge that few and MDM AMs will be present on Day 1 after LPS, MDM AMs will certainly be present on Day 5 (as shown in Fig 6c).

The issue of MDM AMs and the current nomenclature needs to be addressed for two reasons:

- a) The term tissue macrophage (tM) implies that the cells reside in the tissue. However, MDM AMs that would fall under this classification scheme populate the airspace lumen. Thus, the terminology is confusing.
- b) The gating strategies used in the paper will never enable the authors to distinguish MDMs in the airspace (AM) from MDMs in the interstitium (IM)

Experiments to MDM AMs from MDM IMs would most likely require instillation of intratracheal mAb (such as anti-CD45 mAb) or dyes (such as PKH). While this would add to the paper, I don't find it absolutely necessary and would be satisfied if the authors lumped all MDMs together as long as they were described clearly and correctly. Further expansion of extended flow panel showing an overlay with all of the macrophage populations on the various gates would be helpful.

Major Comment 2

The lung-shielded chimeras are a nice tool to aid in lineage tracing. However, I'm somewhat surprised at some of the results. In particular, it is curious that the tissue macrophages (tM) that are suggested to arise from circulating monocytes are negative for Ly6c. Given the short time frame of the experiment, one would expect that monocyte derived tM would be Ly6c positive (as seen by others). To firm up this part of the story, additional information is needed.

- a) In figure S6J, can the please authors show chimerism for both Ly6chi and Ly6clo monocytes?
- b) Please describe exactly how normalization of tM is calculated. This could be in the Methods or elsewhere in the text.
- c) Gating strategies that include Ly6C for tM should be shown in the supplement
- d) EdU or BrdU pulse chase studies are needed to confirm that the Ly6C- tM do not arise from proliferation of a subset of cells already present in the tissue.
- e) If EdU / BrdU pulse chase experiments (as suggested in (d)) show that Ly6C- tMs arise from monocytes, analysis of the lungs at an earlier time point should be performed to assess whether the monocytes that traffic in are initially Ly6c hi or Ly6c low.

Major Comment 3

Studies with IL-10 in the lungs need to be strengthened. The main implication of the story is that production of IL-10 by MDM tMs is a key anti-inflammatory mechanism.

a) The IL-10 staining in the tissues is not convincing. IL-10 staining is not evident in the inset (although this could be due to the tiny size). In addition, negative (isotype) controls are not shown. A panel with larger images and appropriate negative controls should be added to the supplement. This would help provide assurance

b) Figure 7t shows gene expression of the Lyve-1 positive tMs. This dataset should be interrogated for expression of IL-10 and the results reported.

c) The current data don't conclusively prove that MDM tMs are the source of IL-10. I suspect that resident tissue IMs may also produce IL-10 and that many of the Lyve-1 + tMs are not monocyte derived (as touched on in Major Comment 2). The authors already present a robust amount of data in the paper. I would be satisfied if the origin of Lyve-1 tMs is not completely determined.

Major Comment 4

Staining for Lyve-1 is not overly convincing. This could be due to the small size of the figure. Much of the Lyve-1 stain seems to be non-specific. Moreover, the region shown in the H-CSF1 panel is an area around a large airway. While the bronchovascular bundles are often enriched for macrophages, the area of interest in this paper is the alveolar interstitium. A panel with larger images and appropriate negative controls should be added to the supplement.

Minor Comments

Please describe antigen retrieval methods used for immunofluorescence in greater detail in the Methods. This should include the antigen retrieval buffer, and whether or not heat was used.

Page 16 Line 12 states "We found the non-host chimerism of tM ϕ from unchallenged chimeric mice maintained in normoxic conditions to be around ~20% when normalised to that of blood monocytes." This is difficult to read and understand. Is it correct to state that only ~20% of tM ϕ were of donor origin when normalized to blood monocytes.

Author Rebuttal, first revision:

Dear Ioana,

Thank you very much for reviewing our work and for your on-going interest in our study. We note that we had addressed many of the points raised by the reviewers at first revision but that reviewers 2 and 3 have outstanding points that require further clarification. These largely pertain to refinement of terminology for macrophage populations, clarification of the origin of Ly6C negative tissue macrophages, provision of new tissue immunofluorescence data, and changes in the manuscript text to reference monocyte accumulation and conversion, clearer experimental detail and reference to limitations of the work. We have now had opportunity to modify the manuscript text and provide new experimental data to address the points raised by reviewers 2 and 3, as detailed in the point by point response below. With these changes, we very much hope all remaining issues have been resolved

and as a consequence you will consider our manuscript suitable for publication in Nature Immunology.

Reviewers' Comments:

Reviewer #1:

Remarks to the Author:

No further comments

Reviewer #2:

Remarks to the Author:

The authors have addressed many of the concerns raised in the original review of the manuscript. A few issues remain.

We thank the reviewer for acknowledging that we have addressed many of the concerns raised at original review and provide further clarification of the outstanding issues below.

One detail in their protocol of characterizing the lung tissue immune cells is missing. As the authors are aware, although vascular perfusion removes leukocytes from the lung vasculature, it does not completely eliminate all cells. Thus, where there is a need to distinguish between tissue and circulating leukocytes, as in this study, anti-CD45 antibodies given intravenously followed by quick harvest of the lungs allows gating out of cells in circulation. Did the authors do that? Please see protocol by Gibbins and Jakubzick in Lung Innate Immunity and Inflammation (Methods and Protocols) edited by Scott Alper and William Janssen. If they did not they should mention this as a limitation and that their estimates include tissue cells and those in the vasculature.

Unfortunately, we did not add IV anti-CD45 antibodies prior to harvest and have now added this as a limitation in the discussion (page 21 line 3).

The description of the experiment and also the labeling of Fig panels in Fig. 3 is misleading. The legend mentions (similar to Fig. 2), the mice were treated with LPS first and then exposed to normoxia or hypoxia (although panel d legend is also incorrectly written). The following text gives the impression it was the opposite-i.e. normoxia or hyperoxia was followed by LPS. "As expected, LPS significantly increased neutrophil numbers in the lung (Figure 3b) in both hypoxia and normoxia, with equivalent numbers of T cells and B cells in the lung (Figure S3a

and Figure Sb). However, whilst LPS administration also led to a significant expansion of the macrophage (CD45+ Lineage–CD64bright) compartment in normoxia, this was completely abolished in the setting of hypoxia (Figure 3c and 3d).”

Apologies for any confusion caused, we have now clarified in the text that mice were first challenged with LPS and immediately after this, mice were housed in hypoxia or normoxia (page 8 line 15, page 48 line 2).

The authors did not use the marker CD43 now routinely used to distinguish between classical and non-classical monocytes, classical Mos being Ly6ChiCD43- and non-classical Mos Ly6Clo/-CD43+. Since a major conclusion of the authors is deficiency of Ly6C+ MDMs, with the markers they have used (Ly6C/CD11b/CD64/SiglecF) can they distinguish between classical Mos and Mo-derived macs under conditions of LPS/H vs LPS/N? Is the defect in recruitment (although a better term would be accumulation) of classical Mos or their differentiation to macs? What are Ly6C+ MDMs (Figs 3h and 3i)? Can they define them better with reference to the literature? This is important to discuss further since a major conclusion is failure to expand monocyte-derived tissue macs.

We thank the reviewer for raising this important point and agree that nomenclature matters. While the distinction of monocytes and their macrophage progeny is well described in health, this becomes more problematic following an inflammatory insult. In our study, we considered bona fide classical monocytes in the lung to be CD11b+Ly6C^{hi} cells with low expression of CD64 (please see figure 1a, b below and manuscript figure S3c). In contrast, we considered cells with high levels of CD64, as well as Ly6C, to be MDMs (please see manuscript Figure S6m), in keeping with previous studies (see Tamoutounour et al EJI <https://doi.org/10.1002/eji.201242847>). We have amended the text to make this explicitly clear. In addition, we agree with the reviewer that accumulation and conversion are more accurate terms and have edited the text to reflect this (page 2 line 19, page 2 line 23, page 9 line 19, page 19 line 13).

Figure 1: Classical monocytes within the lung

(a) Representative dot plots of gating strategy for classical monocytes in the lung gated on Singles Live CD45⁺Lin⁻ lung cells. (b) Classical monocyte absolute lung numbers in naïve or LPS-treated mice, housed in normoxia (N) or hypoxia (H) for 24hours. Data points represent individual mice from 2 pooled independent experiments. One-way ANOVA with Tukeys post-test.

Reviewer #3:

Remarks to the Author:

Mirchandani and colleagues have improved their manuscript “Hypoxia shapes the immune landscape in lung injury promoting inflammation persistence” by adding additional data and

refining their results and discussion. Of particular note, the authors have expanded their description of human subjects and provide compelling data that suggest that the study population suffered from hypoxia. In addition, they provide new data supporting HIF-1alpha driven pathways and have firmed up links involving CSF-1 and IFNAR.

Taken as a whole, the findings of the manuscript are expected to be of high impact and should advance the field. However, there are four items that should to be addressed to complete the work.

These are discussed in detail below as Major Comments. In brief, the points include:

1. Refinement of terminology for macrophage populations and recognition of monocyte derived macrophages in the airspaces.
2. Clarification of origin of Ly6c negative tissue macrophages.
3. Shoring up data that suggest IL-10 production by tissue macrophage subsets
4. Improved presentation of tissue immunofluorescence

We thank the reviewer for acknowledging the improvements made in the manuscript and the importance of this body of work. With refinement of macrophage terminologies, clarification of cellular origins and the provision of new IHC data, we hope to have addressed the four outstanding points raised.

Major Comment 1

In response to the prior review, the authors have refined terminology and classification of macrophages. However, the classification remains confusing, and interpretation of data may be incorrect. The main issue revolves around monocyte derived macrophages that populate the airspaces after LPS exposure (MDM AMs). These cells are most easily seen on Figure 6c, middle panel, in which BAL was performed 5 days after LPS. Since the figure shows BAL, it is clear that the cells are in the airspaces (i.e. AM). If the gating strategy from Fig 1 was applied to these cells, it would most likely show that the MDM AMs are CD11c low-mid, Siglec-F-, and Ly6C high. The MDM AMs will also be CD64 positive (although it is possible their CD64 could be slightly less than the resident AMs). As a consequence, under the current strategy the MDM AMs are likely to be misidentified defined as tissue macrophages (tM) and would further be classified as MDM tM.

The authors suggest in their rebuttal that: a) lavage can effectively remove all AMs from the lung and b) monocyte-derived AMs do not exist in the LPS model. Neither of these are true. Substantial numbers of AMs remain in the lungs even after aggressive lavage (PMID: 8586127). The presence of MDM AMs in BAL following LPS administration is shown in a number of papers including (PMID 16543608, 33117383, 21471090, 28421818, 21278303). The cite Liu et al (Cell 2019) as evidence that MDM AMs do not exist in the LPS model. However, only Siglec-F + CD11c+ cells were assessed for the Ms4a3 reporter in the paper;

MDM AMs were not assessed. While I acknowledge that few and MDM AMs will be present on Day 1 after LPS, MDM AMs will certainly be present on Day 5 (as shown in Fig 6c).

The issue of MDM AMs and the current nomenclature needs to be addressed for two reasons:

a) The term tissue macrophage (tM) implies that the cells reside in the tissue. However, MDM AMs that would fall under this classification scheme populate the airspace lumen. Thus, the terminology is confusing.

b) The gating strategies used in the paper will never enable the authors to distinguish MDMs in the airspace (AM) from MDMs in the interstitium (IM)

Experiments to MDM AMs from MDM IMs would most likely require instillation of intratracheal mAb

(such as anti-CD45 mAb) or dyes (such as PKH). While this would add to the paper, I don't find it absolutely necessary and would be satisfied if the authors lumped all MDMs together as long as they were described clearly and correctly. Further expansion of extended flow panel showing an overlay with all of the macrophage populations on the various gates would be helpful.

We have found allocating names to the phenotypically identified subsets to be complicated, in part due to the dynamic nature of monocytes and their macrophage progeny in LPS induced lung injury. On reflection, based on the above comments, we agree that the approach we took previously is not clear enough. We also agree that indeed, in our system we cannot distinguish alveolar MDM from tissue MDM when examining cells amongst lung homogenates. To avoid confusion and perceived misinterpretation, we have therefore taken on the helpful advice to pool all MDM together. In addition, we have renamed all macrophages as lung macrophages with further granularity provided by surface marker expression. We hope that together this provides sufficient refinement of the classification of macrophage populations.

Major Comment 2

The lung-shielded chimeras are a nice tool to aid in lineage tracing. However, I'm somewhat surprised at some of the results. In particular, it is curious that the tissue macrophages (tM) that are suggested to arise from circulating monocytes are negative for Ly6c. Given the short time frame of the experiment, one would expect that monocyte derived tM would be Ly6c positive (as seen by others). To firm up this part of the story, additional information is needed.

We thank the reviewer for acknowledging the power of lung-shielded chimeras in tracking the contribution of monocytes to lung macrophage subsets. We apologise for the omission of the data showing the chimerism of Ly6C⁺ MDM in these mice. We have added this to the revised manuscript (Figure S6k and included below for convenience). Given potential precursor-product relationship

between Ly6C⁺ and Ly6C⁻ subsets, we felt it was important to show these data. We too were initially surprised by the high level of turnover of Ly6C⁻ subsets over such a short time frame, however, these data are consistent with preliminary fate mapping using Cx3cr1^{Cre-ERT2} x Rosa26^{LSL-RFP} (data not included in the manuscript). In this complementary system, homeostatic interstitial macrophages can be labelled with RFP through administration of tamoxifen. Then, following a 1 week 'wash out' period to ensure loss of label in blood monocytes, injury can be induced through administration of LPS. Using this system, we show that LPS administration leads to a significant loss of RFP label in lung Ly6C⁻ macrophage subsets consistent with arrival of (RFP⁻) monocytes. While we do not intend to include these data due to their preliminary nature, we have provided these data here to give the reviewer confidence that the turnover of the SiglecF⁻ fraction of CD64^{bright} cells is marked and rapid following LPS induced injury (using complementary systems).

Figure 2: chimerism of lung MDM relative to blood chimerism

C57BL/6J CD45.1⁺CD45.2⁺ mice were reconstituted with congenic cells (CD45.2^{+/+}) 24 hours following lung-shielded radiation and rested for 8 weeks. Subsequently, naïve- or LPS-treated mice were housed in normoxia (N) or hypoxia (H) for 5 days and treated with PBS or CSF1, as per the graph labels. Lung MDM were identified by gating on Singles Live CD45⁺Lin⁻CD64^{bright}SiglecFLY6C⁺ and the proportion of these cells that were of donor origin was quantified. This value was normalised to the monocyte blood chimerism of that individual mouse demonstrating the proportion of these cells that were of blood ontogeny.

a) In figure S6J, can the please authors show chimerism for both Ly6chi and Ly6clo monocytes?

We now include the chimerism of blood monocytes post-LPS for both Ly6chi and lo monocytes as requested (see Figure 3 below and amended manuscript figure S6J).

Figure 3: Blood monocyte chimerism in lung-protected chimeras

C57BL/6J CD45.1⁺CD45.2⁺ mice were reconstituted with congenic cells (CD45.2^{+/+}) 24 hours following lung-shielded radiation and rested for 8 weeks. Subsequently, naïve- or LPS-treated mice were housed in normoxia (N) or hypoxia (H) for 5 days and treated with PBS or CSF1, as per the graph labels. The proportion of cells of donor origin (i.e. chimerism) was measured in classical (Ly6Chi) and non-classical (Ly6Clo) blood monocytes (Live Singles CD45+Lin-CD115+CD11b+) by flow cytometry.

b) Please describe exactly how normalization of tM is calculated. This could be in the Methods or elsewhere in the text.

We apologise for this oversight and now include clear explanation of the normalization method in page 49, line 20. In summary, the chimerism of blood monocytes (proportion of donor cells) was determined by flow cytometry in each individual mouse at day 5 and the chimerism in the lung macrophage populations (as described in the figures) was divided by this reference value, thereby determining the proportion of the cells that were of blood ontogeny.

c) Gating strategies that include Ly6C for tM should be shown in the supplement

The gating strategy for the CD64^{bright} compartment is shown in manuscript Figure 3, with more detail in Figure S6m (see below Figure 4). Figure 3 has now been amended to ensure the gating is clearer. The classical monocyte gating is now also shown in Figure 1a, b (above) and in manuscript Figure S3c for clarity.

Figure 4 Gating strategy for identification of the different monocyte and macrophage populations in the lung.

Representative dot plots of lung digests gated on Live Singles CD45+Ly6G⁻ cells, including Lyve1⁺MHC⁻CD64^{bright}SiglecF⁻MΦ, and associated APC FMO control

d) EdU or BrdU pulse chase studies are needed to confirm that the Ly6C⁻ tM do not arise from proliferation of a subset of cells already present in the tissue.

We thank the reviewer for this suggestion and we agree that we cannot completely exclude a degree of proliferation in the lung as a contributor to the increase in lung macrophage numbers following LPS. We do, however believe that the chimera experiment addresses this question directly since the dominant mechanism for the expansion of the CD64^{bright}SiglecF⁻ compartment is through blood monocyte conversion. Our data demonstrates that following LPS stimulation in normoxia, monocyte recruitment and conversion to lung macrophage leads to around 80% of non-MDM CD64^{bright}SiglecF⁻ to be of blood monocyte origin (please see Figure 7j and Fig 5 below) demonstrating that the dominant mechanism for the expansion of this population is through recruitment. This is significantly reduced in the context of hypoxia, and rescued by treatment of the mice with CSF1. We have amended the text to ensure this important point is clear and, with the inclusion of the above additional changes as helpfully suggested (as described in points 2a, b, c), to ensure this message is accurately conveyed (see page 17 line 3, page 18, line 2).

Figure 5: Ly6C lung macrophage expansion is via blood monocyte recruitment following LPS challenge

The chimerism (proportion of cells of donor origin) was measured in Ly6C⁻ lung macrophages (gated on Singles Live CD45⁺Ly6G⁻CD64^{bright}SiglecF⁻Ly6C⁻) by flow cytometry and compared to the chimerism of blood monocytes for each individual mouse in lung-shielded chimeras challenged with LPS, housed in normoxia (N) or hypoxia (H) for 5 days, and treated with PBS or LPS.

e) If EdU / BrDU pulse chase experiments (as suggested in (d)) show that Ly6c⁻ tMs arise from monocytes, analysis of the lungs at an earlier time point should be performed to assess whether the monocytes that traffic in are initially Ly6c^{hi} or Ly6c^{low}.

Please see response to point 2d above.

Major Comment 3

Studies with IL-10 in the lungs need to be strengthened. The main implication of the story is that production of IL-10 by MDM tMs is a key anti-inflammatory mechanism.

a) The IL-10 staining in the tissues is not convincing. IL-10 staining is not evident in the inset (although this could be due to the tiny size). In addition, negative (isotype) controls are not shown. A panel with larger images and appropriate negative controls should be added to the supplement. This would help provide assurance

We have undertaken additional experimental work and now provide new staining to address this point. Please see figure 7r and Figure S7a

b) Figure 7t shows gene expression of the Lyve-1 positive tMs. This dataset should be interrogated for expression of IL-10 and the results reported.

Many thanks for this helpful suggestion. We have interrogated our nanostring data and now include the graph below (Please see Figure 6 below and manuscript figure 7o and S6o) . Whilst we observe MHCII⁻ lung macrophages from PBS-treated mice versus the MHCII⁻ Lyve1⁺ cells from CSF1-treated mice to have equivalent il10 transcript expression, we also include a graph showing the total number of Lyve1⁺ lung MΦ in our model. Since CSF1 was able to vastly expand the number of IL-10expressing Lyve1⁺ lung macrophages in the lung, this provides a mechanism whereby CSF1- treatment can contribute to increased IL-10 levels in the lung.

Figure 6: il10 gene expression in MHCII-lung MΦ and absolute MHCII-Lyve1+ lung MΦ

(a) il10 expression was measured by NanoString platform analysis in MHC⁻ lung macrophages from LPS-challenged mice housed in hypoxia for 5 days and treated with PBS and compared to Lyve1⁺ MHCII⁻ of LPS-challenged mice, housed in hypoxia and treated with CSF1. (b) Absolute numbers of lung Lyve1⁺ MHCII⁻ lung macrophages in LPS-challenged mice, housed in normoxia (N) or hypoxia (H) for 5 days and treated with either PBS or CSF1 were quantified by flow cytometry. Unpaired t-test applied to data representative of 3 independent experiments.

c) The current data don't conclusively prove that MDM tMs are the source of IL-10. I suspect that resident tissue IMs may also produce IL-10 and that many of the Lyve-1 + tMs are not

monocyte derived (as touched on in Major Comment 2). The authors already present a robust amount of data in the paper. I would be satisfied if the origin of Lyve-1 tMs is not completely determined.

We agree that we have not conclusively shown that MDM are the only source of IL-10 within the lung and agree that the resident lung macrophages are likely sources of this cytokine too, as shown in other models (Bedoret JCI 2016, Zhou Nature Immunology 2020). We have shown in our lungprotected chimera model that approximately 80% of the Lyve1+ lung macrophages post-LPS in normoxia, are of blood origin. This proportion drops to around 50% in the context of hypoxia, with an uplift to 70% in CSF1-treated mice in hypoxia (figure 7p). We agree with the reviewer that this still leaves a proportion of Lyve1+ lung macrophages which are likely derived from resident macrophages and have amended the text to reflect this (see page 17 line 3, page 18, line 2, page 18 line 7, page 18 line 20).

Major Comment 4

Staining for Lyve-1 is not overly convincing. This could be due to the small size of the figure. Much of the Lyve-1 stain seems to be non-specific. Moreover, the region shown in the H-CSF1 panel is an area around a large airway. While the bronchovascular bundles are often enriched for macrophages, the area of interest in this paper is the alveolar interstitium. A panel with larger images and appropriate negative controls should be added to the supplement.

We have undertaken additional experimental work and now provide new Lyve-1 staining to address this point, with appropriate negative controls included in the supplement (Figure 7s, supplemental figure S7b).

Minor Comments

Please describe antigen retrieval methods used for immunofluorescence in greater detail in the Methods. This should include the antigen retrieval buffer, and whether or not heat was used.

Antigen retrieval was performed by microwave heating in citric acid-based antigen unmasking solution (Vector, cat. H-3300-250). We now include these details in the methods section (page 55 line 14 and 21)

Page 16 Line 12 states “We found the non-host chimerism of tM□ from unchallenged chimeric mice maintained in normoxic conditions to be around ~20% when normalised to that of blood monocytes.” This is difficult to read and understand. Is it correct to state that only ~20% of tM□ were of donor origin when normalized to blood monocytes.

We thank the reviewer for raising this point. We have now addressed this result in the text (page 16, line 18) and it now reads “In unchallenged chimeric mice, the chimerism observed in

CD64^{bright}SiglecF^{bright}Ly6C⁻ M \square was ~20%, relative to the blood. This demonstrates that, during normal homeostatic conditions, replenishment of this population is dependent on blood monocytes, albeit at low levels. Given that most of these will represent interstitial macrophages in the healthy lung, our work is consistent with previous studies. Administration of LPS in normoxia led to a dramatic increase in chimerism (~80%), in keeping with enhanced recruitment of bone marrow-derived cells into the non-AM CD64^{bright}SiglecF^{bright}Ly6C⁻ M \square compartment demonstrating that recruitment of blood monocytes is the dominant mechanism driving the expansion of this population." We hope this clarifies the reporting of this result.

Decision Letter, second revision:

Subject: Your manuscript, NI-A31918B

Message: Our ref: NI-A31918B

28th Mar 2022

Dear Dr. Mirchandani,

Thank you for your patience as we've prepared the guidelines for final submission of your Nature Immunology manuscript, "Hypoxia shapes the immune landscape in lung injury promoting inflammation persistence" (NI-A31918B). Please carefully follow the step-by-step instructions provided in the attached file, and add a response in each row of the table to indicate the changes that you have made. Please also check and comment on any additional marked-up edits we have proposed within the text. Ensuring that each point is addressed will help to ensure that your revised manuscript can be swiftly handed over to our production team.

We would like to start working on your revised paper, with all of the requested files and forms, as soon as possible (preferably by April 4th). Please get in contact with us if you anticipate delays.

When you upload your final materials, please include a point-by-point response to any remaining reviewer comments and please make sure to upload your checklist.

In recognition of the time and expertise our reviewers provide to Nature Immunology's editorial process, we would like to formally acknowledge their contribution to the external peer review of your manuscript entitled "Hypoxia shapes the immune landscape in lung injury promoting inflammation persistence". For those reviewers who give their assent, we will be publishing their names alongside the published article.

Nature Immunology offers a Transparent Peer Review option for new original research manuscripts submitted after December 1st, 2019. As part of this initiative, we encourage our authors to support increased transparency into the peer review process by agreeing to

have the reviewer comments, author rebuttal letters, and editorial decision letters published as a Supplementary item. When you submit your final files please clearly state in your cover letter whether or not you would like to participate in this initiative. Please note that failure to state your preference will result in delays in accepting your manuscript for publication.

Cover suggestions

As you prepare your final files we encourage you to consider whether you have any images or illustrations that may be appropriate for use on the cover of Nature Immunology.

Nature Immunology has now transitioned to a unified Rights Collection system which will allow our Author Services team to quickly and easily collect the rights and permissions required to publish your work. Approximately 10 days after your paper is formally accepted, you will receive an email in providing you with a link to complete the grant of rights. If your paper is eligible for Open Access, our Author Services team will also be in touch regarding any additional information that may be required to arrange payment for your article.

Please note that *Nature Immunology* is a Transformative Journal (TJ). Authors may publish their research with us through the traditional subscription access route or make their paper immediately open access through payment of an article-processing charge (APC). Authors will not be required to make a final decision about access to their article until it has been accepted. [Find out more about Transformative Journals](https://www.springernature.com/gp/open-research/transformative-journals).

If you have any questions about costs, Open Access requirements, or our legal forms, please contact ASJournals@springernature.com.

Authors may need to take specific actions to achieve [a](https://www.springernature.com/gp/open-research/funding/policy-)

compliance-faqs"> compliance with funder and institutional open access mandates. If your research is supported by a funder that requires immediate open access (e.g. according to [Plan S principles](https://www.springernature.com/gp/open-research/plan-s-compliance)) then you should select the gold OA route, and we will direct you to the compliant route where possible. For authors selecting the subscription publication route, the journal's standard licensing terms will need to be accepted, including [self-archiving policies](https://www.springernature.com/gp/open-research/policies/journal-policies). Those licensing terms will supersede any other terms that the author or any third party may assert apply to any version of the manuscript.

Please use the following link for uploading these materials: [REDACTED]

Best regards,

Elle Morris
Senior Editorial Assistant
Nature Immunology
Phone: 212 726 9207
Fax: 212 696 9752
E-mail: immunology@us.nature.com

On behalf of

Ioana Visan, Ph.D.
Senior Editor
Nature Immunology

Tel: 212-726-9207
Fax: 212-696-9752
www.nature.com/ni

Reviewer #3:

Remarks to the Author:

The authors have adequately addressed concerns from prior reviews. I have no additional concerns. This manuscript will add to the field.

Final Decision Letter:

Subject: Decision on Nature Immunology submission NI-A31918C

Message: In reply please quote: NI-A31918C

Dear Dr. Mirchandani,

I am delighted to accept your manuscript entitled "Hypoxia shapes the immune landscape

in lung injury promoting inflammation persistence" for publication in an upcoming issue of Nature Immunology.

Over the next few weeks, your paper will be copyedited to ensure that it conforms to Nature Immunology style. Once your paper is typeset, you will receive an email with a link to choose the appropriate publishing options for your paper and our Author Services team will be in touch regarding any additional information that may be required.

Please note that *Nature Immunology* is a Transformative Journal (TJ). Authors may publish their research with us through the traditional subscription access route or make their paper immediately open access through payment of an article-processing charge (APC). Authors will not be required to make a final decision about access to their article until it has been accepted. [Find out more about Transformative Journals](https://www.springernature.com/gp/open-research/transformative-journals).

Your paper will be published online soon after we receive your corrections and will appear in print in the next available issue. Content is published online weekly on Mondays and Thursdays, and the embargo is set at 16:00 London time (GMT)/11:00 am US Eastern time (EST) on the day of publication. Now is the time to inform your Public Relations or Press Office about your paper, as they might be interested in promoting its publication. This will allow them time to prepare an accurate and satisfactory press release. Include your manuscript tracking number (NI-A31918C) and the name of the journal, which they will need when they contact our office.

About one week before your paper is published online, we shall be distributing a press release to news organizations worldwide, which may very well include details of your work. We are happy for your institution or funding agency to prepare its own press release, but it must mention the embargo date and Nature Immunology. Our Press Office will contact you closer to the time of publication, but if you or your Press Office have any enquiries in the meantime, please contact press@nature.com.

Also, if you have any spectacular or outstanding figures or graphics associated with your manuscript - though not necessarily included with your submission - we'd be delighted to consider them as candidates for our cover. Simply send an electronic version (accompanied by a hard copy) to us with a possible cover caption enclosed.

Please note that we encourage the authors to self-archive their manuscript (the accepted version before copy editing) in their institutional repository, and in their funders' archives, six months after publication. Nature Research recognizes the efforts of funding bodies to increase access of the research they fund, and strongly encourages authors to participate in such efforts. For information about our editorial policy, including license agreement and

author copyright, please visit www.nature.com/ni/about/ed_policies/index.html

Sincerely,

Ioana Visan, Ph.D.
Senior Editor
Nature Immunology

Tel: 212-726-9207
Fax: 212-696-9752
www.nature.com/ni